

# Biomass burning events measured by lidars in EARLINET. Part II. Results and discussions.

Mariana Adam[1], Doina Nicolae[1], Livio Belegante[1], Iwona S. Stachlewska[2], Lucja Janicka[2], Dominika Szczepanik[2], Maria Mylonaki[3], Christiana Anna Papanikolaou[3], Nikos Siomos[4], Kalliopi Artemis Voudouri[4], Luca Alados-Arboledas[5], Juan Antonio Bravo-Aranda[5], Arnoud Apituley[6], Nikolaos Papagiannopoulos[7], Lucia Mona[7], Ina Mattis[8], Anatoli Chaikovsky[9], Michaël Sicard[10,11], Constantino Muñoz-Porcar[10], Aleksander Pietruczuk[12], Daniele Bortoli[13], Holger Baars[14], Ivan Grigorov[15], Zahary Peshev[15]

[1]National Institute for R&D in Optoelectronics, Magurele, 077225, Romania
[2]Faculty of Physics, University of Warsaw, 02-093, Poland
[3]National Technical University of Athens, Department of Physics, Athens, 15780, Greece
[4]Laboratory of Atmospheric Physics, Aristotle University of Thessaloniki, Thessaloniki, 54124, Greece
[5]Andalusian Institute for Earth System Research, Department of Applied Physics, University of Granada, Granada, 18071, Spain
[6]KNMI – Royal Netherlands Meteorological Institute, De Bilt, 3731, the Netherlands
[7]Consiglio Nazionale delle Ricerche - Istituto di Metodologie per l'Analisi Ambientale (CNR-IMAA), C.da S.Loja. Tito Scalo (PZ), Italy
[8]Deutscher Wetterdienst, Meteorologisches Observatorium Hohenpeißenberg, Hohenpeißenberg, 82383 Germany
[9]Institute of Physics, NAS of Belarus, Minsk, 220072, Belarus
[10]Remote Sensing Laboratory/CommSensLab, Universitat Politecnica de Catalunya, Barcelona, 08034, Spain
[11]Ciencies i Tecnologies de l'Espai - Centre de Recerca de l'Aeronautica i de l'Espai/Institut d'Estudis Espacials de Catalunya (CTE-CRAE/IEEC), Universitat Politecnica de Catalunya, Barcelona, 08034, Spain
[12]Institute of Geophysics, Polish Academy of Sciences, Warsaw, 01-452, Poland
[13]Earth Sciences Institute, Physics Department, University of Évora, Évora, 7000, Portugal
[14]Leibniz Institute for Tropospheric Research, Leipzig, 04318, Germany
[15]Institute of Electronics, Bulgarian Academy of Sciences, 1784, Sofia

*Correspondence to*: Mariana Adam (mariana.adam@inoe.ro)

**Abstract.** Biomass burning events are analysed using the European Aerosol Research Lidar Network database for atmospheric profiling of aerosols by lidars. Atmospheric profiles containing forest fires layers were identified in data collected by fourteen stations during 2008–2017. The data ranged from complete data sets (particle backscatter coefficient, extinction coefficient and linear depolarization ratio) to single profiles (particle backscatter coefficient). The data analysis methodology was described in Part I (*Biomass burning events measured by lidars in EARLINET. Part I. Data analysis methodology*, under discussions to ACP, the EARLINET special issue). The results are analysed by means of intensive parameters in the following directions: I) long range transport of smoke particles from North America (here, we divided the events into 'pure North America' and 'mixed'-North America and local) smoke groups, and II) analysis of smoke particles over four geographical regions (SE Europe, NE Europe, Central Europe and SW Europe). 24 events were determined for case I). A statistical analysis over the four geographical regions considered revealed that smoke originated from different regions. The smoke detected in



the Central Europe region (Cabauw, Leipzig, and Hohenpeißenberg) was mostly brought over from North America (87 % of the fires), by long range transport. The smoke in the South West region (Barcelona, Evora, and Granada) came mostly from the Iberian Peninsula and North Africa, the long-range transport from North America accounting for only 9 % here. The smoke in the North Europe region (Belsk, Minsk, and Warsaw) originated mostly in East Europe (Ukraine and Russia), and had a 31

% contribution from smoke by long-range transport from North America. For the South East region (Athens, Bucharest, Potenza, Sofia, Thessaloniki) the origin of the smoke was mostly located in SE Europe (only 3 % from North America). Specific features for the lidar-derived intensive parameters based on smoke continental origin were determined for each region. Based on the whole dataset, the following signatures were observed: i) the colour ratio of the lidar ratio and the backscatter Ångström exponent increase with travel time, while the extinction Ångström exponent and the colour ratio of the particle

depolarization ratio decrease; ii) an increase of the colour ratio of the particle depolarization ratio corresponds to both a decrease of the colour ratio of the lidar ratios and an increase of the extinction Ångström exponent; iii) the measured smoke originating from all continental regions is characterized in average as aged smoke, except for a few cases; iv) in general, the local smoke shows a smaller lidar ratio while the long range transported smoke shows a higher lidar ratio; and v) the depolarization is smaller for long range transported smoke. A complete characterization of the smoke particles type (either

fresh or aged) is presented for each of the four geographical regions versus different continental source regions.

## 1 Introduction

The biomass burning (BB) context was presented in Adam et al., 2020, where the BB was reviewed, and its importance and role on radiative transfer, air quality and human health, were highlighted. An overview of the fire monitoring perspective was discussed therein as well. There is a direct link between climate change and forest wildfires. The European Union reports

(http://effis.jrc.ec.europa.eu/reports-and-publications/annual-fire-reports/) of fires occurrence over Europe indicate that the climate change induces an increase in the number of fires. Flannigan et al. (2000) modelled the climate change impact, demonstrating an increase of forest wildfire activity. Carvalho et al. (2011) modelled the impact of forest fires in a changing climate on air quality (a case study on Portugal) showing a big impact on ozone and PM10 (particulate matter with dimension up to 10 μm). One of the current challenges is evaluating accurately the role of BB in climate change. Keywood et al. (2013)

report that the inverse effect of BB impact on climate is well recognized but not fully understood. The authors state that, based on the BB impact on air pollution, climate, poverty, security, food supply and biodiversity, a more effective control of the fires is needed, along with continuous and improved monitoring. EARLINET (European Aerosol Research Lidar Network; e.g. Papalardo et al., 2014) provides high temporal and spatial resolution ground-based measurements of the transported smoke, and represents a valuable tool for smoke monitoring. EARLINET is part of the Aerosol Cloud and Trace Gases Research

Infrastructure (ACTRIS) (https://actris.nilu.no/, last access: 20200505). The current study proves that the EARLINET database is an appropriate source of the information necessary to characterize the BB smoke at various locations throughout Europe. There are numerous studies describing various BB events over Europe, most of them focusing on the smoke optical properties



of either fresh/local aerosol (e.g. Alados-Arboledas et al., 2011; Sicard et al., 2012; Balis et al., 2003; Nicolae et al., 2013; Stachlewska et al. 2017a,b; Osborne et al., 2019) or aged/long range transported aerosol (Wandinger et al., 2002; Mattis et al., 2003; Müller et al., 2005; Ancellet et al., 2016; Ortiz-Amezcua et al., 2017; Haarig et a., 2018; Hu et al., 2019; Stachlewska et al., 2018; Vaughan et al., 2018). The findings reported in these studies, as well as in studies over other regions outside

Europe, are considered when interpreting the results presented in the current paper.

This paper presents Part II of a study on the biomass burning as measured by EARLINET, and it focuses on results interpretation. Part I (Adam et al, 2020) described in detail the methodology used to analyse lidar data. However, a short overview of the methodology is given in Section 2. In Section 3 we analyse the results for long range transport (LRT) of smoke from North America. In Section 4, we focus on results from four European geographical regions, with different continental

smoke origin. Finally, in Section 5 we provide the summary and conclusions. A list of acronyms used in the current work is given in the Supplement (Table S2).

## 2 Review of methodology

The methodology steps are shown in Fig. S1 (Fig. 2 in Adam et al., 2020). The input for the analysis is the backscatter (b) and extinction (e) files providing the profiles of particle backscatter coefficient, particle extinction coefficient, and particle linear

depolarization ratio. Most of these files are allocated to the Forest Fire category in the EARLINET/ACTRIS database, and their quality is ensured by EARLINET Quality Check (QC) procedures. However, additional files were directly provided by several stations (not in the database as for March 2018, nor checked by EARLINET QC procedures). Note that all the data are in EARLINET database as for 20200620. Therefore, for the further analysis carried out in this work, other specific QCs were applied to selected datasets (as described in Part I). The smoke layers were identified following an in-house developed method.

For each layer, the backtrajectory was computed for ten days using the Hybrid Single-Particle Lagrangian Integrated Trajectory model (HYSPLIT) (Stein et al., 2015; Rolph et al., 2017). The meteorological model applied is the Global Data Assimilation System (GDAS), with 0.5° resolution. The identification of the smoke layers was assessed based on the hypothesis of an existing fire within 100 km and ± 1 h from the time and location of the airmass, respectively. The location of the fires was provided by the Fire Information for Resource Management System (FIRMS) (https://firms.modaps.eosdis.nasa.gov/, last

access 20191126) from satellite observations with the Moderate Resolution Imaging Spectroradiometer (MODIS) sensor airborne by the Aqua and Terra satellites (Davies et al., 2009). The mean optical properties within the layers were calculated following a few criteria; only the optical properties for which SNR ≥ 2 were selected. The number of layers and optical properties for each time stamp are shown for all stations in Fig. S2 and the mean optical properties in the smoke layers are shown in Fig. S3 (in the Supplement). The last criterion for selecting the QC intensive parameters consisted on the rejection

of outliers. The number of layers and intensive optical parameters (IPs) for each time stamp are shown for all stations in Fig. S4 while the corresponding IPs values for each station are shown in Fig. S5 (in the Supplement). All the IP values (including the outliers) are shown, along with the extreme values considered as acceptable (marked by lines).



Although when taking into consideration the satellites (Terra and Aqua) overpasses (four observations a day) and their footprints we may have missed some fires, if they occurred within short periods between two consecutive overpasses, one can assume that such short-lived fires did not significantly contribute to the amount of transported smoke. Further, some fires might have not been detected due to clouds.

The mean, median, minimum and maximum values of the intensive parameters for all of the stations providing at least one parameter (all stations but Sofia) are shown in Table 1. The number of available values for each variable is shown in Table 1 (# lines). As mentioned in Part I, there was a small number of IPs dismissed (outliers) based on predefined ranges of acceptable values (3.7 % overall). Regarding LR@355 obtained for the Thessaloniki station, we observed lower values than the ones reported by Amiridis et al. (2009) and Siomos et al. (2018a). The current mean value (over 20 cases) lies within the standard

deviation reported by Amiridis et al. (2009) but it is lower than the Free Troposphere minimum and maximum values of $61 \pm 5$ sr and $71 \pm 7$ sr, respectively (monthly averages over 15 years with filtered extreme values), reported by Siomos et al (2018a). Note that the current dataset for Thessaloniki was processed with SCC (version 4), while the dataset used by Siomos et al. (2018a) was processed with a station algorithm. Siomos et al. (2018b) showed that the AOD values (based on the particle extinction coefficient retrieved from the Raman channel) computed with the SCC algorithm are underestimated, as compared

with those processed with the inhouse algorithm.

### 3 Long range transport biomass burning events

24 events of LRT from North America were identified, for which at least one intensive parameter was retrieved. The events occurred during 2009 and 2012–2017. The same graphics as in Section 3 are used to describe the fires' location, the backtrajectories and the IPs retrieved at stations. As mentioned in Part I, eight events represented measurements of smoke

coming solely from North America ('pure North America'), while the others represented 'mixed' smoke (mixture of North American and local smoke). "Local smoke" refers to smoke originating in European locations, in general. In a few cases, the smoke came from North Africa or Middle East. The number of fires as well as the number of their detections (a fire can be detected more than once) are quantified.

In the Part I paper we presented the event recorded on 13 July 2017 in Athens. Three layers (centred at 2942 m, 2102 m and

3872 m) were identified as LRT smoke. The first two layers were identified as mixed, while the third layer as 'pure North America' smoke. The first two layers revealed a much larger contribution from the local fires than the third layer (6/8 and 18/19 fires for the two events). For the third layer, values of CRLR > 1 and EAE ~ 1 suggested the presence of relatively large, aged particles. For the second layer CRLR was < 1, suggesting the presence of fresh smoke, in agreement with the large contribution of the local fires. No CRLR was derived for the first layer. The BAE@532/1064 and BAE@355/532 had,

respectively, the largest and lowest values for the 'pure North America' smoke. The CRBAE showed a larger value for 'pure North America' smoke.



### 3.1. Smoke event recorded during 20130708–20130710

The most interesting event was recorded during 8–10 July 2013 at three stations: Belsk, Cabauw and Warsaw. The Hysplit backtrajectories along with locations of the fires are shown in Fig. S6 in the Supplement. Even without having a fire radiative power (FRP) detailed analysis, one can notice that the fires detected in North America (especially over Canada) were very

strong, with FRP values around 2000 MW or even higher (see colour coded fires marks), in most of the cases examined. The location of the fires and the calculated IPs are shown in Fig. 1, a–d and e–i, respectively. Panels a–d show all the fires contributing to the measurements on 8–10 July 2013, most of them located in North America (panels a–b). The coordinates of the fires versus the fires' occurrence and smoke measurement times are shown in Figs. 4c, d, respectively. There is also a fire located in Europe (Sweden, 17.988 E, 59.175 N) that have been detected eleven times by Belsk and Warsaw (see blue and

yellow marks over East longitudes), and contributed to smoke mixing. Panels c)-d) show that the fires occurred during 30/06 – 05/07 period for North America and on 07/07 for the local fire while the smoke was recorded during 08/07 – 10/07 period. Due to the presence of many fires, it is hard to mark the individual fires on c) and d) (similar with Figs. 12 in Part I). Panels e–i show the smoke layers and corresponding IPs determined for the 8–10 July period. However, only some of them contain smoke originating from North America (according to our criteria) as marked by squares on panel e) and shown further on

panels j–n (the mixed cases are marked by diamonds).

As a first remark, we observed a very large number of fires occurring in North America (hundreds) from 30 June to 5 July 2013. Specifically, the smoke from 961 fires, detected 1664 times was measured by Belsk, 855 fires were detected 1241 times by Cabauw, and 646 fires were detected 1065 times by Warsaw. This amounts to a total of 2462 fires, detected 3970 times. Most of these fires were quite strong, as shown by the colour and size of the markers (Figs. S6). The local fire in Sweden

occurring on 7th of July contributed to the measurements taken on the 8th (four layers in Warsaw, one layer in Belsk) and 9th (one layer in Belsk). In Table 2 are listed the IPs measurements for all stations shown in Fig. 1, j–n, along with contributing fires information (location and occurrence time). The detection time of the fires represents the time interval during which they were detected, i.e. the furthest and closest times. Similarly, the coordinates represent the area in which the fires were detected, i.e. the farthest and nearest latitude and longitude. We show in red the contributing local fire. The only measurement with all

IPs available is highlighted (second measurement on 8th of July). Based on EAE and $CR_{LR}$, we were able to label the smoke as aged only for this measurement. At the first glance, the values of the intensive parameters for 'mixed' and 'pure North America' smoke are quite similar. Hence, the local fire contribution to the smoke mixtures was minimal. Indeed, the FRP for the local fire in Sweden was 9.9 MW, while the FRP for North America ranged from a few dozens to hundreds of MW (over 800 MW). Thus, we may assume the local fire was not strong enough to significantly influence the mixture.

This event was partly discussed by Ortiz-Amezcua et al. (2017) and Janicka et al. (2017). The study by Ortiz-Amezcua et al. describes BB measurements at three stations (Granada, Leipzig and Warsaw). However, the measurements analysed were taken on different days (14/07, 17/07 and 09/07 for the Granada, Leipzig and Warsaw, respectively) and, thus, most probably the fire source was different. Strong fires over North America were reported over two weeks (1–15 July). Our closest



measurement to that reported for the Warsaw station (00:00–01:00 UTC on 9th of July) was at 20:25, on 8th of July. Ortiz-Amezcua et al. reported the following values for Warsaw: layer altitude 2280 ± 100 m, LR@355 = 34 ± 6 sr, LR@532 = 58 ± 10 sr, EAE = 1 ± 0.1, BAE@355/532 = 1.9 ± 0.2. Unfortunately, our criteria did not provide any retrievals for LR and EAE. The only common IP was BAE@355/532 = 1.9 ± 0.04, where the mean altitude was ~ 2000 m (remarkably close to their

retrievals). Janicka et al. (2017) analysed measurements taken during 10th of July over Warsaw, and reported values of EAE of 0 ± 0.3, LR@355= 50 ± 10 sr, LR@532=50 ± 10 sr, BAE@355/532 =0 ± 0.1, BAE@532/1064 = 1 ± 0.2 (layer estimate of 2900–3200 m a.s.l.) for the 18:20–19:40 time interval. Our only available measurement was taken between 18:30–19:30 (for b files). Thus, our BAE values (layer estimated at 2664–4210 m a.s.l.) were 0.5 ± 0.05 and 0.9 ± 0.03 for 355/532 and 532/1064, respectively. Based on Hysplit backtrajectory (GDAS1), a Saharan dust origin is observed for the layer at 3000 m a.g.l. (see

their Fig. 3). Our Hysplit backtrajectory (GDAS0.5) for 3437 m (mean layer altitude) indicated a BB origin in North America. Moreover, the ensemble backtrajectory for 3000 m a.g.l. at 19:00 using GDAS 0.5 showed more backtrajectories towards North America. Note that small differences are seen if one uses as input altitude a.g.l. (above ground level) or altitude a.s.l. For further insights related to the differences between the backtrajectories please see Su et al. (2015), where backtrajectories are compared using two GDAS datasets. In conclusion, based on the different approach to estimate the smoke layer and

different input to Hysplit (GDAS, altitude a.s.l. or a.g.l.), the results are not in agreement.

In our study, the mean and standard deviation (STD) for all IPs (except LR@532 = 91 ± 3 sr and EAE = 0.3 ± 0.1, where we have only single values) are as following: LR@355 = 38 ± 6 sr (average over nine cases), BAE@355/532 = 1.4 ± 0.5 (average over 13 cases), BAE@532/1064 = 1.1 ± 0.3 (average over 31 cases), PDR@355 = 2.5 ± 0.4 % (average over nine cases), PDR@532 = 2.4 ± 0.9 % (average over eight cases). IPs STD represents ~ 15, 38, 27, 17 and 37 % of the mean, respectively,

and thus we claim a relatively small variability over the whole three days of measurement. The low depolarization indicates the presence of particles with better sphericity, while the low LR at 355 nm (< 50 sr) may indicate low absorption. The BAE values indicate larger backscatter at smaller wavelengths (355 nm), i.e. 10 out of 13 measurements where both BAE were estimated have BAE@355/532 larger than BAE@532/1064. We speculate that an increase in LR indicates a stronger absorption, as previously reported (e.g. Tesche et al., 2011; Kolgotin et al., 2018; Veselovskii et al., 2020; Ohneiser et al.,

2020). In general, the change in LR can be linked to a change in particle size and/or a change in the light absorption capability of the particles (Müller et al., 2007).

For another event (not shown), recorded on 14/07 2013 in Granada over 01:31–02:00 UTC, the IPs retrieved in the layer with a mean altitude of 6080 m (520 m thickness) were BAE@355/532 = 1.8 ± 0.1 and BAE@532/1064 = 2.3 ± 0.03. Ortiz-Amezcua et al. (2017) reported for Granada, for the 14/07 average over 02:00–03:00 UTC and the layer @ 5200 ± 100 m, a

BAE@355/532 of 1.2 ± 0.5. Based on the backscatter profile at 1064 nm, we estimated three layers centred at 4840, 5460 and 6080 m. However, based on Hysplit and FIRMS, we found that only the uppermost one was a smoke layer. BAE@355/532 was reported as increasing with height, for the same flight path (Janicka et al., 2019). The lesson learned here is that the layers can be defined based on different criteria and, thus, a straightforward comparison between different reports is a delicate



endeavour. On the other hand, slightly small changes in the Hysplit input may give different results for certain atmospheric situations.

Note that during August 2017–January 2018, a stratospheric event of LRT from North America (Canada) was reported by many lidar stations. Layers between 10 and 20 km altitude were recorded at many EARLINET lidar stations in Europe
(Ansmann et al., 2018; Haarig et al., 2018; Hu et al., 2019; Sicard et al., 2019; Haarig et al., 2019; Wang and Stachlewska, 2020). Baars et al. (2019) presented measurements by 23 lidar stations in Europe and West Asia. At stratospheric altitudes, the most reliable measurements are the ones for the aerosol backscatter coefficient and, sometimes, the lidar depolarization ratio (e.g. Haarig et al., 2019). The aerosol optical depth (AOD) estimates in the layers are performed using an a priori LR (e.g. Baars et al., 2019; Wang and Stachlewska, 2020).

In the present study, two stations (Observatory Hohenpeißenberg and Evora) reported in EARLINET the August 2017 wildfires in Canada (Peterson et al. 2018). The Observatory Hohenpeißenberg measured two layers between 10 and 12 km on 17 August 2017, and only EAE could be retrieved (0.78 and 0.89). For the Evora measurements on 31 August 2017, the backtrajectories for layers at ~ 17 km altitude show an end point in Europe recorded 10 days later. However, for the lower tropospheric smoke altitudes with North American origin, we retrieved the following values:

- @ 00:45 5520 m, 36 fires detected 52 times: BAE@532/1064 = 0.8 ± 0.4. Fires were located in North USA and South Canada, and the smoke travel time was 130–190 h.
         - @ 04:45 5820 m, 3 fires detected 6 times: BAE@532/1064 = 1 ± 0.2. Fires were located in central North Canada, and the travel time was 180 h.
         - @ 05:45 5780 m, 3 fires detected 6 times: LR@532 = 36 ± 9 sr, BAE@532/1064 = 1.5 ± 0.1. Fires were located in
20         central East Canada and the travel time was 96 h.
         - @ 20:45 1860 m, 15 fires detected 15 times: BAE@355/532 = 2.3 ± 0.04, BAE@532/1064 = 1.3 ± 0.1. Fires were located in SW Canada and travel time was 140 h.

The backtrajectories are similar for the second and the third case, but the fires location and their travel times are different. The first backtrajectory is similar to the second and third case over Atlantic and East Canada, but differ in Canada and USA. The
fourth case has a different pathway, at lower latitudes. For these measurements, the BAE@532/1064 decreases with travel time.

For measurements recorded in 4 September 2017 in Evora, Sicard et al. (2019) reported the following values: BAE@532/1064 = 1.3 ± 0.1, LR@532=55 ± 14 sr, EAE=1.5 ± 0.8 for Mid Troposphere (average over ~ 2.5–8 km).

To better characterize the LRT, $CR_{LR}$ and EAE are essential. The LRT smoke measured might not be strong enough and thus
not all the backscatter signals have a good SNR, required for estimating a greater range of IPs (especially EAE).



### 3.2. Statistics on LRT

We encountered 168 measurements over the 24 LRT periods, as follows: one period in 2009 and 2011, two periods in 2012, 2014, three periods in 2015, and five periods in 2013, 2016 and 2017. From these measurements, 77 have a North American origin and 91 have a different BB origin (local).

All these periods lasted one day, except for the period 8–10 July 2013 (previously discussed). The following stations performed LRT measurements (individually): Athens (20140210, 20140320, 20160711, 20170713, 20171012), Barcelona (20120724, 20160523), Bucharest (20150416, 20160530 ), Evora (20170831), Granada (20120911, 20130714, 20130819), Leipzig (20130824), Minsk (20110927, 20150430), Observatory Hohenpeißenberg (20170817), Thessaloniki (20090831, 20150828, 20160328), and Warsaw (20160702, 20170619).

In Fig. 2 are shown the measurements with intensive parameters and layers altitudes taken during the LRT periods (smoke originating in North America is shown in black and mixed smoke in blue). At a first glance we do not see major differences between the two cases. The mean, minimum and maximum values from literature are displayed in red (values and corresponding references are presented in Table 3). Compared to the values found in the limited existing literature for LRT from North America, we noted several IP values (especially for BAE@355/532) that fall outside of the range reported. The

large value for the mixed case EAE may be due to the contribution of the local, fresh smoke. The large value reported for the 'pure North America' smoke was not investigated. We observe the following features. All intensive parameters for 'pure North America' except LR@355 and EAE cases are close to the mean values reported in literature (see references shown in Table 3). However, we do not know if the values from references are 'pure North America' or 'mixed', as such an examination was not reported. We also observe that our mean PDR values are smaller compared with the mean over the reported values.

However, the current mean values are within the extreme values found in literature for LRT from North America (see Fig. 2 and Table 3). The minimum mean value reported for PDR@355 was $2.1 \pm 4$ % (Haarig et al., 2018). Similarly, PDR@532 means of 2.9 % and 3 % were reported by Haarig et al. (2018) and Müller et al. (2011), respectively. An EAE extreme value of -0.3 was reported by Haarig et al. (2018) for the stratospheric smoke. For the current dataset, we consider that the relative differences between 'mixed' and 'pure North America' cases are not drastic, except for LR@532 and PDR@532: - 24 %

(LR@355), - 43 % (LR@532), 21 % (EAE), - 11 % (BAE@355/532), 13 % (BAE@532/1064), 4 % (PDR@355) and 34 % (PDR@532).

Overall, based on the mean values, we observed a moderate absorption at 355 nm and a high absorption at 532 nm ($CR_{LR} >$ 1), with low depolarization at both wavelengths, low EAE (big particles), slightly larger BAE@355/532 than BAE@532/1064. $CR_{LR}$ and EAE suggest the presence of aged particles, while BAE shows more backscatter for smaller wavelengths.



## 4 Analyses of biomass burning over geographical regions

### 4.1. Geographical regions

The locations of the fires whos' smoke was detected by the stations located in SE, SW, NE and CE Europe, and their histogram are shown in Fig. S7. For a straightforward comparison, we reproduce the figure for the SE region from Part I. Note that the grid size is 1° x 1° longitude and latitude, respectively. First remarks for each region are as follows.

For the SE region, we distinguished a number of 321 fires located in North America (4.3%) and 7127 elsewhere, most of them located in East Europe. Most of the fires were located over [20°E 30°E] and [37°N 46°N]. This corresponds to the Balkan region, covering parts of Romania, Bulgaria, North Macedonia and Greece. Most of the measurements were taken at Bucharest, Athens and Thessaloniki.

For the SW region, we identified a number of 197 fires in North America (8.7%) and 2066 elsewhere, most of the latter being located in the Iberian Peninsula and North Africa. Most of the fires occurred in the region [0° 10°W] x [35°N 43°N]. This corresponds to mostly to the Iberian Peninsula. Other fires were located over [0° 20°E] x [30°N 40°N], corresponding to North Africa (mostly North Algeria) and Sicily in South Italy. Most of the measurements were taken at Granada.

For the CE Europe region, we have found 1420 fires originating in North America (86.9 %) and 214 elsewhere, most of the latter located in East Europe. Most of the fires occurred over [80°W 75°W] x [51°N 53°N] region, which corresponds to North America (East Canada). As mentioned above, Cabauw focused almost entirely on the LRT smoke, contributing to the histogram peak indicating North American locations. Stations Observatory Hohenpeißenberg and Leipzig contained a ~ 24 % and ~79 % LRT contribution as well, but their number of cases is much smaller than that of the Cabauw.

For the NE region, 2761 of fires identified were located in North America (30.7 %), and 6228 elsewhere, most of the latter being located in East Europe (Ukraine and West Russia). Two peaks of the histogram indicate locations from North America ([75°W 78°W] x [51°N 53°N] and 95°W x [57°N 59°N]), which correspond to measurements taken at Belsk (solely LRT) and Warsaw. Most of the measurements in East Europe belong to the grids delimited by [20°E 40°E] x [46°N 53°N] (East Europe, mostly in Ukraine).

The main fire sources recorded are located in: East Europe (especially Ukraine and West Russia), South Europe (Iberian Peninsula, Italy, Balkan region) and North America. Wildfires in the west Russian regions and Ukraine occur each year from March to October. Events of small particles (PM1) transport, in the boundary layer, from these regions to the Nord-West (Belarus, Poland, Germany, Nordic countries and European Arctic) are regularly recorded (Lund Myhre et al., 2007). Such transport of biomass burning aerosol can be extremely fast and affect relative humidity within the boundary layer (Stachlewska et al. 2017b)

The histogram of the backtrajectories (not shown) revealed some preferential air circulation patterns for three of the regions (CE, SW and NE), with one common pattern being circulation over Atlantic. For the SW region, we identified a vortex type circulation over North Africa as the main air pathway. For the NE region we observed other patterns as well: a circulation from



Iberian Peninsula, a circulation from East Europe (Caspian Sea), and a circulation over North Europe (Scandinavian Peninsula and West Russia).

## 4.2. Intensive parameters by geographical regions

A statistical investigation of the intensive parameters was performed, based on their continental fire source origin. As
mentioned in Part I, the following continental source origins were considered: Europe (EU), Africa (AF), Asia (AS), North America (NA), and combinations of two or more of these (EUAF=EU+AF, EUAS=EU+AS, EUNA=EU+NA, etc). The statistical analysis was performed over all the available cases. In Part I, the results for SE were analysed based on the scatter plots between various IPs. To thoroughly assess the aerosol type, the scatter plots of EAE and $CR_{LR}$ were used (for the same measurements), as discussed later.

Here we present the results for the NE region. In Fig. 3 are shown the scatter plots between the two LR, the two PDR and the two BAE. The other combinations are shown in Supplement (Fig. S8). As specified in Part I, the number of pair points available for each combination is different. Overall, we observed a well-defined linear correlation between the two LR and between the two PDR, as previously reported in literature (e.g. Nicolae et al., 2018; Janicka et al., 2019). Our main remarks on these scatter plots are as follows (the mean values are investigated). Large LR values (~ 70–80 sr) observed for EU and EUAS source

regions suggest a large absorption of the smoke particles (larger in comparison with SE region). A mean value around 0.95 (1.2) is observed for $CR_{LR}$ for EU (EUAS) source region, corresponding to fresh (aged smoke). In average, the values of EAE for EU and EUAS regions are ~1.4 and ~1.2 respectively, suggesting a mixture of fresh and aged smoke for the EU source region and aged smoke for the EUAS source region. All PDR values correspond to low depolarization (< 8 %). Based on the scatter plot between the two PDR, we obtained the largest PDR values for the EUAF region, suggesting that the presence of

particles from Africa increases the depolarization. Mixing with mineral particles might contribute to the increase the depolarization value. Smaller values of PDR are observed for the NA and EUNA source regions. $CR_{PDR}$ (colour ratio of the PDRs) ranges from 0.96 (EUNA) to 2.18 (EUAF). These values correspond to a PDR@532 decrease with time (e.g. Nisantzi et al., 2014).

For the scatter plot between the two BAE, all BAE@355/532 are larger than BAE@532/1064 ($CR_{BAE}$ < 1), signifying more
backscatter at 355 nm. The BAE@532/1064 corresponding to EU and EUAF source regions has the lowest values, suggesting it has a higher contribution to backscatter from large particles, compared with other source regions. Large values are observed for both BAE for the AS source region, suggesting more backscattering from small, rather than medium, size particles originating from Asia. The similarity between NA and EUNA source regions suggests a major contribution from NA to the EUNA mixture. The standard deviation for all BAE cases is large and there is a large overlap among individual values. In

conclusion, based on the current (limited) dataset, there is a signature for the mean values but there is also a large overlap for individual cases. For $CR_{BAE}$ the values are similar, in the range 0.4 (EUAF) to 0.75 (AS) showing the larger backscatter at smaller wavelengths.





The main features for the CE and SW regions, based on the scatter plots (Fig. S9) are the following. For the CE region, we observe low absorption, low depolarization and EAE ~ 1.5 (fresh smoke) for the EU source region. Based on the scatter plot between the two BAE, we observe that BAE@355/532 (BAE@532/1064) increases (decreases) from the EU to the NA source regions. For the SW region, we observe high absorption at both wavelengths and EAE < 1 (aged smoke), while there is a direct proportionality between the two BAEs. BAE@355/532 is the largest for the NA and EUNA source regions, while BAE@532/1064 is the largest for the AF source region.

### 4.3 Statistical analysis over the all regions

The analysis based on the mean IP values can be performed in various ways. Here we chose to analyse the function of continental source region. One can look at the mean values computed as the average over all available measurements for each IP. Recall that the number of events for each IP may vary for different measurements. Thus, the synergetic interpretation based on all IPs is challenging. Alternately, one may consider analysing the scatter plots between the different CRs and EAE, where, for each scatter plot, the mean values correspond to the same measurements. However, different scatter plots can be based on slightly different sets of measurements. The latter approach was chosen for our investigations.

As a general statement, we consider that the cases where we have only one or two measurements are not statistically significant, a good confidence being given by at least five measurements available. Therefore, the results discussed below should be regarded with care in such situations.

### 4.3.1 Signature based on scatter plots

The general observations based on the scatter plots between CR or EAE (Fig. 4) reveal the following. First, note that each point on the graph represents the average for one measurement region (SE, SW, CE and NE) and one continental source (EU, AF, AS, NA, EUAF, EUAS, EUNA). All available data are used for averages. We added for comparison the mean values (red circles, Fig. 4) found in literature (Table S1, Part I). Additional mean values defined as the average of the minimum and maximum values are added as well (blue circles). The later values slightly deviate from the others (Fig. 4a–c). For the NE region, the PDRs provided by the Warsaw station allowed for a complete set of values availabilities. For a better visualization of the mean CR (Fig. 4) and the corresponding IPs, in Fig. 5 are shown the CR and IP values versus continental source regions (i.e. each of the panels a–f of Fig. 5 corresponds to one of the a–f scatter plots of Fig. 4). The right-hand side axis shows the number of available measurements for the scatter plots.

For increasing $CR_{PDR}$ we found an increase of the EAE (Fig. 4b), while the $CR_{LR}$ decreased (Fig. 4c). The correlation coefficient (R) was 1, indicating fresh smoke with higher depolarization, respectively aged smoke with lower depolarization, at 532 nm. A slight decrease of the $CR_{PDR}$ with travel time was observed, while the $CR_{BAE}$ maintained similar values for all the source regions. An increase of EAE versus decreasing $CR_{LR}$ (Fig. 4d), evident especially for the NE region (Fig. 5), was reported also by Samaras et al. (2015) and Janicka et al. (2019). The correlation coefficient was R = 0.52.



No clear relationship between $CR_{BAE}$ and $CR_{LR}$ (Fig. 4e), $CR_{BAE}$ and EAE (Fig. 4f) and $CR_{BAE}$ and $CR_{PDR}$ (Fig. 4a) was noticed ($R < 0.5$). Regarding the relationship between EAE and BAE, Veselovskii et al. (2015) showed that while EAE depends mainly on the particle size, BAE depends both on the particle size and complex refractive index. Thus, the relationship between BAE (and, further, $CR_{BAE}$) and EAE is not straightforward. They reported an increase of BAE@532/1064 and a decrease of

BAE@355/532 (and thus an increase of $CR_{BAE}$) with decreasing EAE, for an EAE ranging between 0.5 and 1.5 (scenario I). As shown in their simulations, this corresponds to a change in effective radius and real part of the refractive index, while the imaginary part of refractive index is constant (Figs. 20 and 22, Veselovskii et al. 2015). They obtained a different behaviour when varying the effective radius while keeping the refractive index and the fine: course mode particle ratio constant (see their Fig. 19). Thus, an increase of both BAEs with increasing EAE is also possible when the fine mode is predominant (scenario

II).

As seen in Fig. 5, different signatures are observed for different measurement regions. Based on the EAE–$CR_{BAE}$ scatter plot, for the source regions EUAF, EUAS and EUNA, we observed the following. For the NE region, EAE decreases (from EUAF towards EUAS and EUNA) while both BAE increase, but $CR_{BAE}$ is similar. For the SW region, EAE, both BAE and $CR_{BAE}$ increase from EUAF to EUNA source regions. For the SE region, EAE increases while no correlation is found for BAE and

$CR_{BAE}$. The CE region provides data for the EU and NA source regions. Here, EAE, both BAE and $CR_{BAE}$ decrease from EU to NA source regions. Considering the findings of Veselovskii et al. (2015), we conclude that, for the CE region, the fine particle mode is predominant for the EU source region (as compared with the NA source region), result which is expected. For the SW region we find a larger amount of fine particles for EUNA source region as compared with EUAS and EUAF source regions (scenario II). This implies a large contribution of the EU source region to the mixture. The NE measurements resemble

partly the scenario I. Here, we also find an increase for BAE@355/532 while based on the LR and $CR_{LR}$ signature, the absorption at 532 nm increases from EUAF towards EUAS and EUNA (thus, the imaginary part of the refractive index is not constant). The relationship between BAE and EAE was analysed from the relative humidity (RH) perspective by Su et al. (2008) and Wang et al. (2019). They showed that the relationship of BAE and EAE depends on the RH values and, thus, one can find correlated and anti-correlated behaviours. We did not investigate the RH dependence here.

**4.3.2 Continental source regions**

As mentioned above, as one can see in Fig. 4d–f, there is no clear relationship between $CR_{BAE}$ and EAE or $CR_{LR}$. A slight decrease of EAE versus increasing $CR_{LR}$ is observed ($R = 0.52$).

Based on the results shown in Figs. 7 and 8, an evaluation for each continental source region can be done, and assess how distinct are the characteristics of the smoke coming from various continental source regions as observed by different

geographical regions. The mean values are shown in Table S1 for each of the d)–f) scatter plots presented in Fig. 4. The smoke type (fresh versus aged) is assessed based on the values of the $CR_{LR}$ and EAE. Information about the smoke absorption and depolarization (where available) is provided.





Except for one case which will be discussed later, for all cases illustrated in Fig. 4 we obtained positive values for BAE (and $CR_{BAE}$), which indicates more backscattering towards smaller wavelengths. All PDR are below 10 % (low depolarization). These features will not be repeated unless there is something very specific. Except for two extremes (-1.6 and 3.2), all $CR_{BAE}$ values range between 0.18 and 1.6. The $CR_{PDR}$ (as for the NE region only) has the largest value for the EUAF source region, followed by EU and EUAS. The lowest $CR_{PDR}$ and EAE values were found for the EUNA source region, characterized also by the highest $CR_{LR}$ (aged smoke; less depolarizing and more absorbing at 532 nm). The high EAE values of the smoke mixtures are likely due to the large EU contribution (EAE value for the NE region with EUAF origin is 1.46, for the SE region with EUAS origin is 1.5, and for the SE region with EUNA origin is 1.9). The following assessment, based on the continental source region, is summarized in Table 4.

EU source region

For the SE region, based on the values of $CR_{LR}$ (1.2) and EAE (1.4), aged smoke might have been measured. The LR corresponds to medium absorption (59/48 sr at 532/355 nm). For the SW region, $CR_{LR}$ (0.82) and EAE (1) suggest contradictory features (fresh versus aged). The absorption is found to be high at 355 nm (~ 78 sr) and relatively high at 532 (~ 64 sr). According to Veselovskii et al. (2020), $CR_{LR}$ <1 can occur when RH is high and the imaginary part of the refractive index for 355 nm is higher (see their Figs. 18–19). The RH corresponding to the three measurements for the SW region (according to Hysplit) was 67.8, 69.8 and 70 %. Thus, we can assume that the smoke is aged. For the CE region, $CR_{LR}$ (1.2) and EAE (1.1) suggest the presence of low absorbing (36/30 sr at 532/355 nm), aged smoke. For the NE region, $CR_{LR}$ (0.95) and EAE (1.4) suggest a mixture of fresh and aged smoke; particles are highly absorbing (74/77 sr at 532/355 nm) and there is more depolarization for medium size particles (532 nm).

For the EU source region, as clear in the scatter plots, the $CR_{BAE}$ values obtained for the four regions are similar for the SE and CE (~ 0.8–0.9), smaller for NE region (~0.4), and larger for SW (1.4). This means that, in SW region, the backscatter at 532 nm is slightly larger compared with the other regions.

AF source region

For the SW measurement region, $CR_{LR}$=0.8 and EAE=1 (based on one case). Similar to the case for the EU source region, we can assume that we detected aged smoke (RH = 73 %). High absorption is seen at both wavelengths (70/84 sr at 532/355 nm). Note that the case discussed here occurred one hour before two of the three cases with the EU source origin. This might explain the similar values. However, the backtrajectories are different. Based on the $CR_{LR}$–$CR_{BAE}$ scatter plot for the SE region, $CR_{LR}$ value was 1.6, i.e. indicating aged smoke (moderately/low absorbing at 532/355 nm).

NA source region

Based on the $CR_{LR}$–$CR_{BAE}$ scatter plot, we obtained for the SE region a $CR_{LR}$ of 2.4, corresponding to aged smoke. Particles are high /low absorbing at 532/355 nm (72/32 sr). Based on the EAE–$CR_{BAE}$ plot (one value available) for the CE region, we obtained an EAE of 0.9, corresponding also to aged smoke. The BAE@532/1064 of - 0.2 and BAE@355/532 of 0.1 indicate more backscattering at 1064 nm ($CR_{BAE}$ of - 1.6).

EUAF source region





For the NE measurement region (one measurement available), the $CR_{LR}$ of 0.9 and EAE of 1.5 suggest fresh smoke, due to the large EU contribution to the mixture. Based on the PDR scatter plots, the depolarization at 532 nm is larger. For the SE and SW regions, both $CR_{LR}$ and EAE values indicate aged smoke. The smoke has a relatively high absorption for all regions (around 65 sr), except for NE at 532 nm, where its absorption is medium. A large $CR_{BAE}$ (5.4) is observed for the SE region, indicating a large contribution to the backscatter from medium size particles (532 nm), besides the small size particles (355 nm) contribution.

EUAS region

For the NE measurement region (two measurements were available), based on both $CR_{LR}$ (1.1) and EAE (1.1), we determined that the smoke measured was aged and very high absorbing at 532 nm (87 sr) and 355 nm (70 sr). Based on the PDR scatter plot, the depolarization is larger at 532 nm. For the SE region, based on the EAE value of 1.5 we have identified fresh smoke, and based on the $CR_{LR}$ value of 1.1, aged smoke. Thus, a mixture of fresh and aged smoke is possible. Smoke is moderately absorbing at both wavelengths (around 50 sr). $CR_{BAE}$ is smaller but positive (0.4) for the SW region, suggesting more scattering towards smaller wavelengths.

EUNA source region

For the NE measurement region (one measurement available), $CR_{LR}$ (2) and EAE (0.3) suggest very highly absorbing / medium absorbing (91/46 sr) at 532/355 nm aged smoke. Based on the PDR scatter plots, there is only slightly more depolarization at 355 nm. For the SW region (four measurements available), according to $CR_{LR}$ (1.3) and EAE (0.9) values, a very highly absorbing / highly absorbing (90/73 sr) at 532/355 nm aged smoke was identified. For the SE measurement region, we have $CR_{LR}$ of 0.9 and EAE of 1.9, suggesting fresh smoke (implying a large smoke contribution from EU source region). Smoke is medium absorbing (50/54 sr at 532/355 nm). Positive but below unity $CR_{BAE}$ for the SW and NE suggest that small particles have a larger contribution to the backscatter.

The converse analysis, i.e. considering firstly the region of measurement and then discussing the source regions, is synthetized below:

SE measurement region:

EU source region: aged smoke, medium absorbing (slightly more absorbing at 532 nm).

NA source region: aged smoke (based on $CR_{LR}$ only), highly/low absorbing at 532/355 nm.

EUAF source region: aged smoke, relatively high absorbing.

EUAS source region: fresh/aged smoke, medium absorbing; fresh smoke due to the contribution of EU source region to the mixture.

EUNA source region: fresh smoke; medium absorbing; fresh smoke due to the large contribution of EU source region to the mixture.

SW measurement region:

EU source region: aged smoke, high/relatively high absorption at 532/355 nm. RH between 68–70 %.

AF source region: aged smoke, high absorption. RH of 73 %.



EUAF source region: aged smoke, relatively high absorption.

EUNA source region: aged smoke, very high/high absorption at 532/355 nm.

$CR_{LR}$ and EAE values for the EU and AF source regions are similar, which suggests the same 'type of contamination' even though the backtrajectories are different.

CE measurement region:

EU source region: aged smoke, low absorbing at both wavelengths.

NA source region: aged smoke (based on EAE only) with more backscatter for large size particles (1064 nm).

NE measurement region:

EU source region: fresh/aged smoke, highly absorbing at both wavelengths, more depolarization at 532 nm.

EUAF source region: fresh smoke, medium/relatively high absorption at 532/355 nm, more depolarisation at 532 nm; the fresh smoke detected is due to the high contribution of the EU source region in the mixture.

EUAS source region: aged smoke, very high/high absorption at 532/355 nm, more depolarization at 532 nm.

EUNA source region: aged smoke, very high/medium absorption at 532/355 nm, slightly more depolarization at 355 nm.

Based on the summary above, the main features are the following. In the SE region it was measured, in general, aged smoke

sourced from the EU, NA and EUAF regions. However, in the SE region a mixture of fresh and aged smoke from the EUAS source region and fresh smoke from EUNA source region was found, due to the local (EU) contribution. The EU source region provided medium absorbing particles, while the AF and NA provided high absorbing particles. In the SW region, aged, highly absorbing smoke particles from all source regions were measured. For the EU and AF source regions (which have similar EAE and LR values) we assume we measure aged smoke, based on the high RH (where $CR_{LR} < 1$ and EAE $< 1$). In the CE region

aged smoke from EU and NA source regions was measured, displaying a low absorption for the EU source region and more backscatter at 1064 nm for the NA source region. The NE region displayed aged smoke from the EUAS and EUNA source regions (highly absorbing), fresh smoke from the EUAF region (due to EU contributions), medium/relatively high absorbing and mixed fresh and aged from the EU source region (highly absorbing). Higher/lower depolarization at 355/532 nm was seen for the LRT (as for the NE region). Based on a single continental source, in all regions but the NE was measured aged smoke

(in the NE was measured a mixture of fresh and aged smoke for the EU source region). Based on two continental sources (mixtures), the regions can measure either aged, fresh or mixed aged and fresh smoke, depending on the lower or higher contribution of the local source.

## 5 Summary and conclusions

The present study shows results based on biomass burning events as recorded by EARLINET stations over the 2008–2017

period, according to a methodology described in Part I (Adam et al., 2020). The main features of the methodology are: aerosol layers were labelled as smoke layers based on their Hysplit backtrajectory and the fire locations (provided by FIRMS), along the airmass backtrajectory according to established criteria. The smoke is labelled as 'mixed' if multiple fires contributed to


the smoke measured. For LRT smoke from North America events, the smoke is labelled as 'pure North America' or 'mixed' (with contribution from both North America and Europe fires). Based on the methodology presented in Part I, we demonstrated that in most of the cases we record mixed smoke. Moreover, the number of fires and detections contributing to a smoke measurement was quantified.

The LRT event described here was recorded at three stations (Belsk, Cabauw and Warsaw) in July 2013, and captured some of the strongest fires that occurred in North America in June–July 2013. Our analysis revealed the presence of only one local fire (Sweden), which was weak compared to the North America ones and, thus, did not significantly contributed to the mixture. The 2462 fires identified in North America were detected 3970 times. The IPs values for 'mixed' and 'pure North America' cases are very similar (weak fire in Europe). The particles depolarization was low and small particles had low absorption.

The statistics over all the LRT events from North America revealed the following. The mean values of all IPs, except the LR@355 and EAE for 'pure North America' smoke, are closer to the mean values reported in literature for LRT smoke coming from North America. However, the relative differences between 'pure North America' and 'mixed' cases are not significant. For the LRT smoke, a moderate absorption at 355 nm (46 sr overall) and a high absorption at 532 nm (71 sr overall) were observed. The mean $CR_{LR}$ and EAE suggest aged smoke, while the PDR values indicate a low depolarization and the BAE 15 reveal more backscatter at smaller wavelengths.

The trajectory analysis based on four geographical regions revealed specific features. The histogram of the fires detected by each region along with the histogram of the backtrajectories revealed the following: the Central Europe stations detected mainly LRT smoke from North America, while the SW Europe region mostly smoke from fires occurring in the Iberian Peninsula and North Africa; in the NE and SE regions was measured mostly smoke from fires occurring in East Europe (especially Ukraine 20 and West Russia). However, sporadic measurements were taken during the presence of smoke coming by LRT from North America. For each region, the IPs and, further, the colour ratio (CR) of various IPs are analysed based on their continental source origin. Most of the measurements were confined locally, within the SE, SW and NE Europe regions. The present methodology results revealed that North American fires contributed by 87 % to the smoke detected in Central Europe, 31 % to the smoke in the NE region, 9 % in the SW region and 4 % in the SE region.

The signature analysis of the scatter plots revealed the following features for the current dataset. $CR_{LR}$ increases while EAE and $CR_{PDR}$ decreases with distance (travel time). EAE increases with increasing $CR_{PDR}$, while $CR_{LR}$ decreases with increasing $CR_{PDR}$. We also noticed that the $CR_{BAE}$ decreases with increasing $CR_{PDR}$, and EAE decreases with increasing $CR_{LR}$. For the current dataset, the variability of the mean values (STD) is large in general and, thus, the individual values for different source regions overlap. Based on data from Warsaw (NE region), the depolarization at 532 nm decreases for LRT (while $CR_{PDR} < 1$).

For a single continental source, we noticed that the smoke is aged for all regions except NE, when the source is located in Europe (where we have a mixture of fresh and aged smoke). Based on two continental sources (mixtures), the regions can measure either aged, fresh or a mixture of aged and fresh smoke, based on the smaller or higher contribution of the European (local) sources. Thus, in the SE measurement region it was measured fresh smoke for the EUNA source regions and a mixture of fresh and aged smoke originating from the EUAS. In the NE region fresh smoke originating from EUAF was measured.





For the SW region with European or African source regions we obtained a $CR_{LR}$ of 0.8 and an EAE of 1. We decided that the smoke measured was aged based on the high RH (in agreement with Veselovskii et al., 2020).

The lowest absorption was determined for the CE region (LRs < 36 sr). The SW region displayed a highly absorbing smoke (61 sr < LR@355 < 79 sr and 64 < LR@532 < 91 sr). The SE region displayed smoke with a medium/relatively high absorption

at 532 nm (50–72 sr) and a low/medium absorption at 355 nm (31–48 sr). The smoke measured in the NE region has a medium to very high absorption at 532 nm (57–91 sr) and a medium to high absorption at 355 nm (46–78 sr). The quite diverse absorption determined for the different measurement's regions, even for smoke from the same continental source region, may be related with different RH conditions (e.g. Veselovskii et al, 2020). We did not investigate the RH field.

The current study showed (in line with previous studies) that BAE and further $CR_{BAE}$ do not show specific values based on

sources and no trend is observed. Thus, they cannot be used to identify the smoke type.  In order to easily quantify the aerosol type, information about LR ($CR_{LR}$) and EAE is essential. Based on the implementation of ACTRIS RI in the next few years, the presented methodology will be applied on a larger dataset (more automatic lidar systems expected) providing more and more complete 3 backscatter + 2 extinction + depolarization datasets with enhanced quality control procedures.

One of the most important features observed on this study is that most of the smoke represents a mixture of several fires, which

can be located very far from each other, and have (most probably) different characteristics. The quantification (based on number of fires and detections) of the contributing fires to the mixture explains the various values obtained for the intensive parameters and colour ratios.

The present methodology used to analyse the biomass burning events shows new approaches for smoke characterization (smoke type along with information about absorption and depolarization in the context of different continental sources) and

can provide valuable information for various scientific communities (modelling, satellites).

For further investigations we envisage a more detailed analysis on grouping the sources' locations using cluster analysis, where a larger number of clusters should be chosen, to identify more homogeneous regions with similar vegetation type. Thus, a more accurate correlation between the source type and the measurements is envisaged. Moreover, the smoke time travel will be integrated. The challenge that remains is the quantification of the contribution of different fires in the mixed smoke (besides

their number and detections).

*Author contributions*. MA developed the methodology, analysed results and wrote the paper. All authors, except MA, contributed by conducting measurements, ensuring data quality, and performing data evaluation and data provision to the EARLINET Data Base. NP, ISS, NS, KAV, LAA, LM, AA, MS, DB, IM, AC, contributed with revisions of the paper. All

authors read the paper and agreed with its content.

*Competing interests*. The authors declare that they have no conflict of interest.



*Special issue statement*. This article is part of the special issue "EARLINET aerosol profiling: contributions to atmospheric and climate research". It is not associated with any conference.

Acknowledgements:

*We acknowledge the use of data and imagery from LANCE FIRMS operated by the NASA/GSFC/Earth Science Data and Information System (ESDIS) with funding provided by NASA/HQ. The authors gratefully acknowledge the NOAA Air Resources Laboratory (ARL) for the provision of the HYSPLIT transport and dispersion model and/or the READY website (http://www.ready.noaa.gov) used in this publication. The authors acknowledge the EARLINET-ACTRIS community for provision of the aerosol lidar profiles used in this study, in particular those who performed measurements, evaluated lidar*

*data and provided profiles to the Forest Fire category in the EARLINET-ACTRIS database. We acknowledge Wojciech Kumala, Krzysztof Markowicz, and Rafal Fortuna (University of Warsaw) for technical support at the ACTRIS site in Warsaw, and Cristi Radu, Dragos Ene and Alexandru Dandocsi (INOE 2000) for technical support at the ACTRIS site in Magurele.*

Funding: *The research leading to these results has received funding from the European Union Seventh Framework Programme (FP7/2007-2013) under grant agreement n° 262254, as well as the H2020 ACTRIS-2 grant n° 654109. It was*

*also supported with following national funding: the Romanian National contracts 18N/08.02.2019 and 19PFE/17.10.2018 as well as with the European Space Agency (ESA-ESTEC) funding: The Technical assistance for Polish Radar and Lidar Mobile Observation System (POLIMOS 4000119961/16/NL/FF/mg). The work is co-funded by the European Union through the European Regional Development Fund, included in the COMPETE 2020 (Operational Program Competitiveness and Internationalization) through the ICT project (UIDB/04683/2020) with the reference POCI-01-0145- FEDER-007690 and*

*also through TOMAQAPA (PTDC/CTAMET/ 29678/2017)*

Data access: *The aerosol lidar profiles used in this study are available upon registration from EARLINET webpage* https://data.earlinet.org/earlinet/login.zul*, last access: 20191126). The FIRMS data used in the study is available upon request from https://firms.modaps.eosdis.nasa.gov/, last access: 20191126).*

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





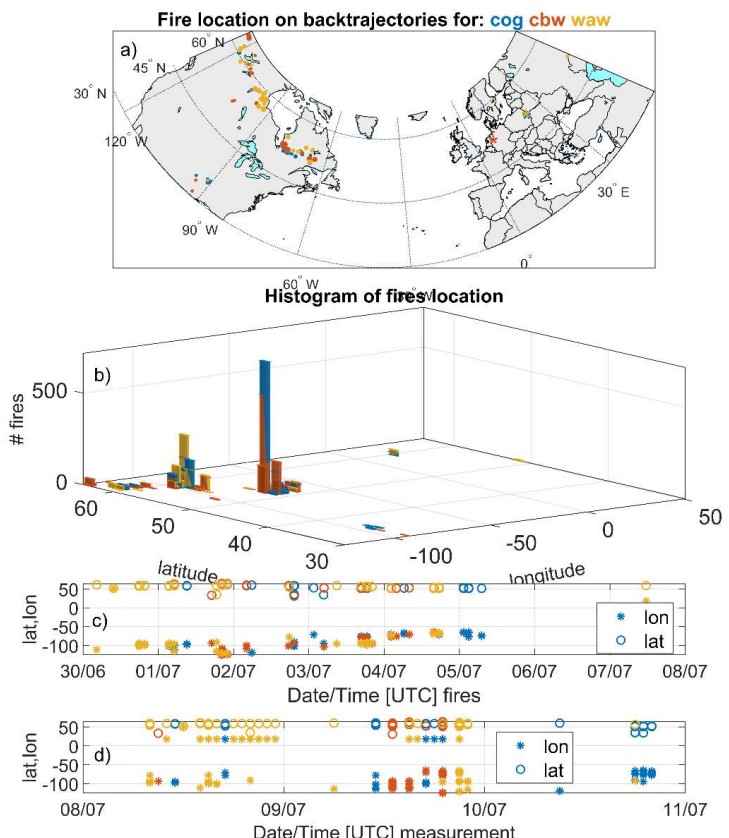





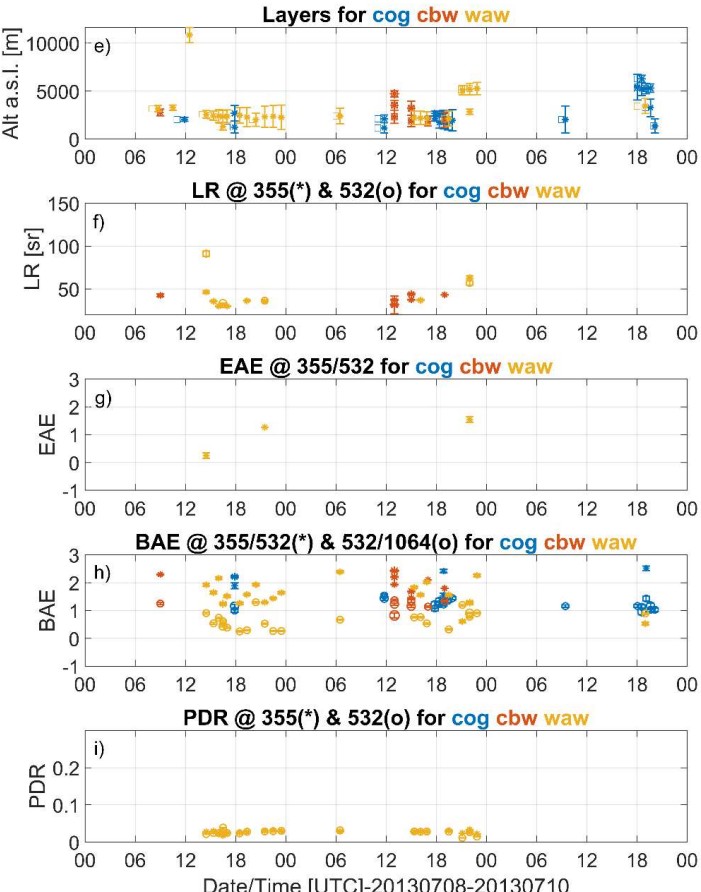



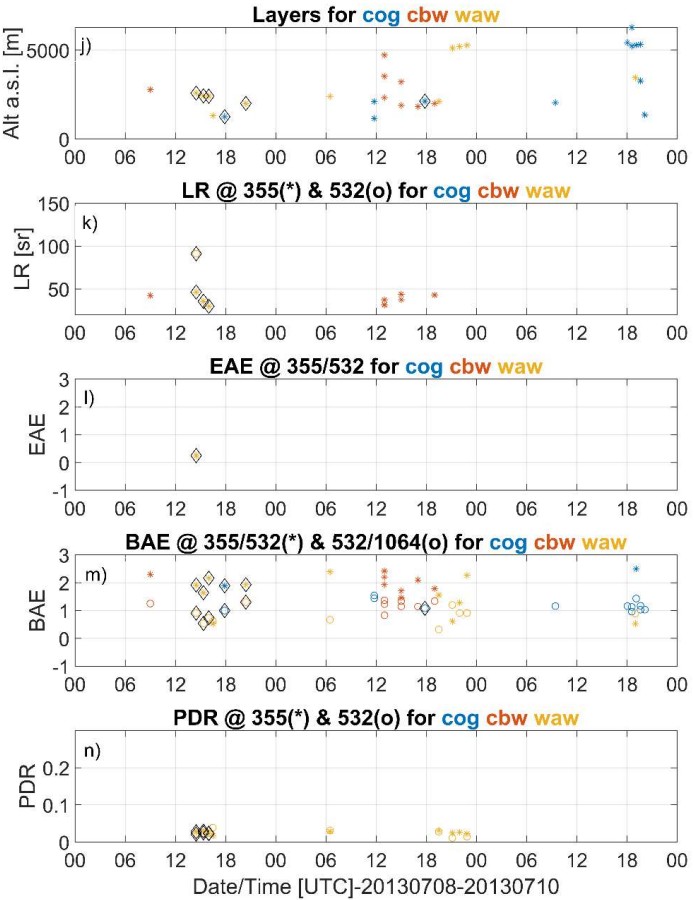

**Figure 1. LRT as measured at Belsk ("cog"), Cabauw ("cbw") and Warsaw ("waw") during 7–10 July 2013. The fires contributing to Belsk, Cabauw and Warsaw measurements are shown in blue, red and yellow respectively (a-d). The measurements in Belsk, Cabauw and Warsaw are shown in blue, red and yellow respectively (e-n). a) Location of the fires; b) Histograms of the fires for each station; c) Fires' coordinates versus fires' occurrence time; d) Fires' coordinates versus smoke measurements time; e) Location of the layers; the smoke layers with North America origin are marked by a square; f)–i) Intensive parameters; j)–n) same as e)–i) for the smoke layers with smoke originating in North America. The layers marked by diamonds represent mixed smoke (North America and local).**



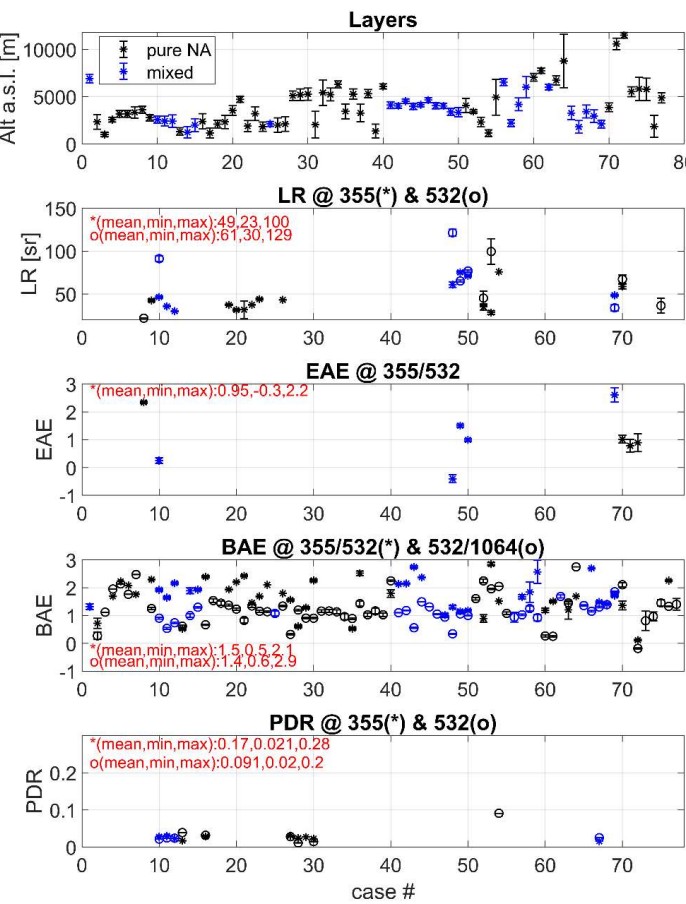

**Figure 2. All of the 77 measurements recorded during LRT from North America. Measurements are divided into North America origin ('pure NA') and 'mixed' (North America and local) origin. Along with the intensive parameters, the layers altitude and thickness (marked as error bar) are shown on the upper plot. Mean, minimum and maximum values from literature are shown in red. 'pure NA' stands for 'pure North America'.**





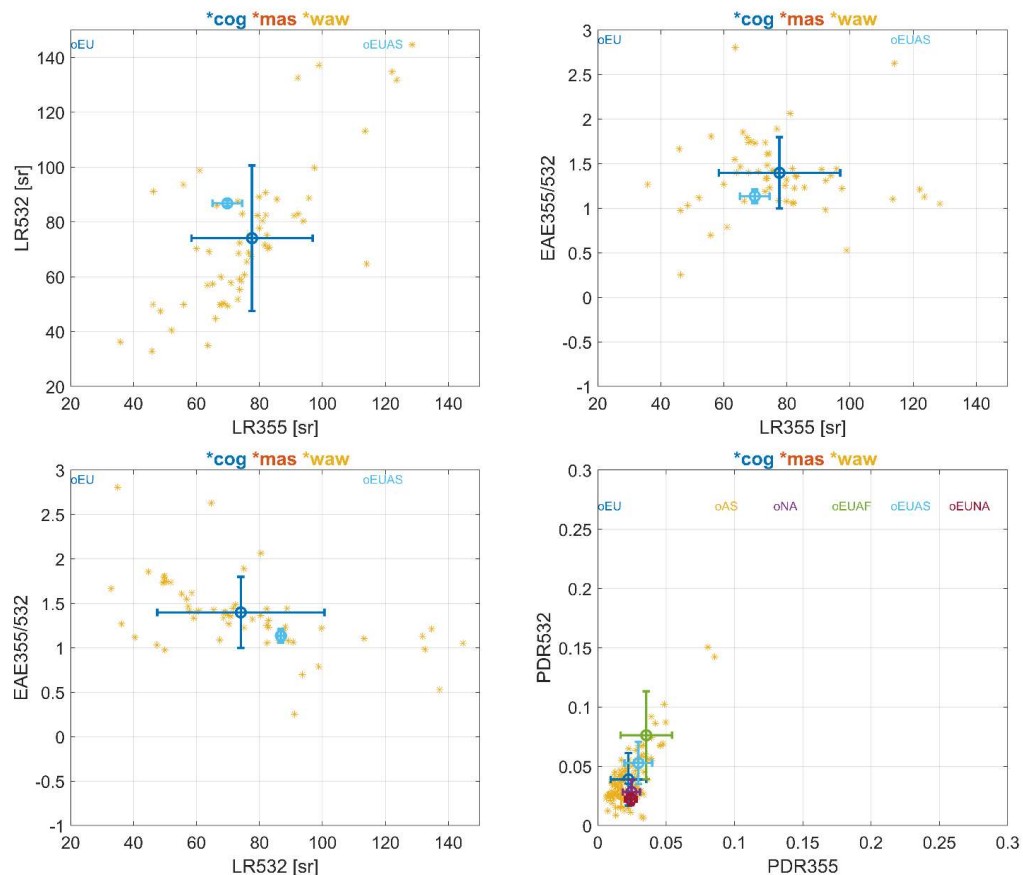





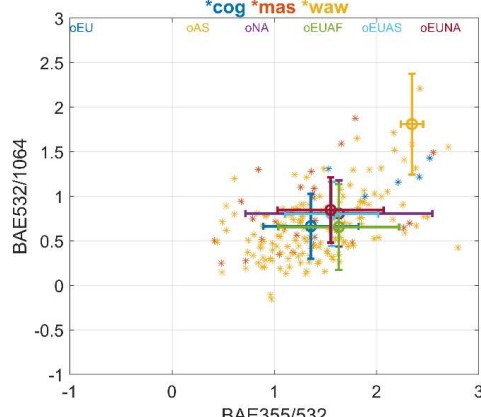

Figure 3. Scatter plots between various two intensive parameters for NE region. The colour code of the asterisks is station related (as labelled in the title). The colour code for the mean (circle) and STD values is related with the source origin (stated as text on the plots).





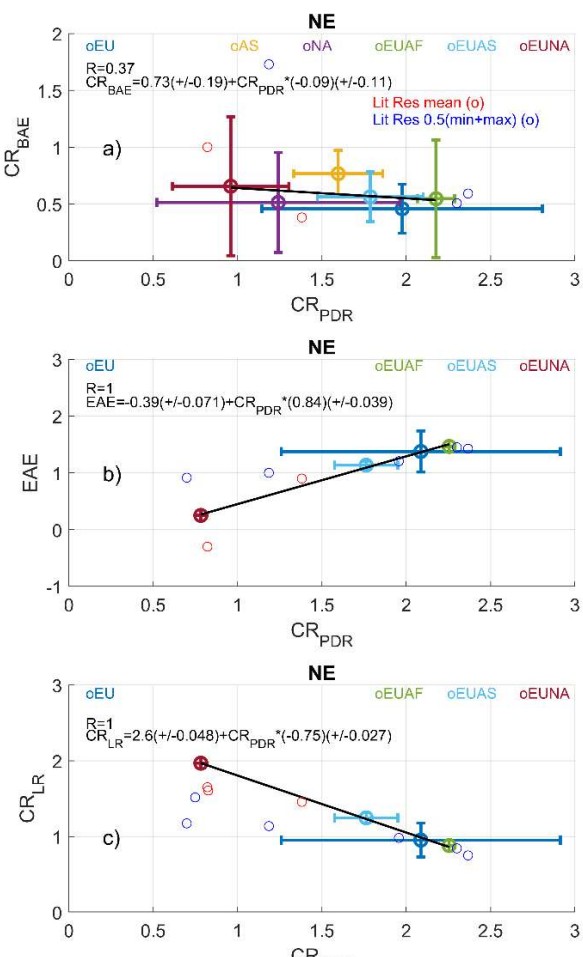





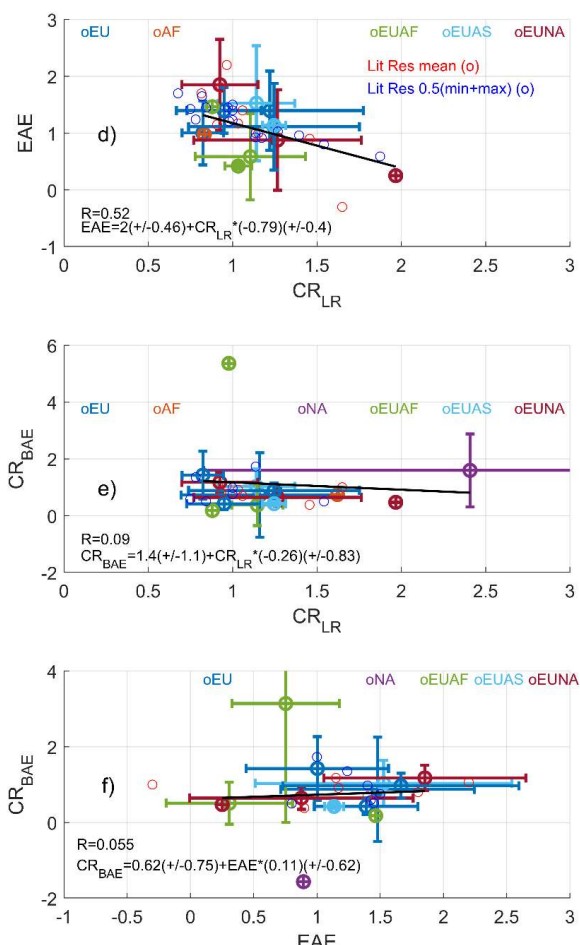

**Figure 4. Scatter plots between CR_BAE and CR_PDR (a), EAE and CR_PDR (b), CR_LR and CR_PDR (c), EAE and CR_LR (d), CR_BAE and CR_LR (e), CR_BAE and EAE (f). The mean values found in literature are added. a)-c) plots are obtained only for NE region where two PDR are available (Warsaw). The regression equation and coefficient of correlation are shown for each plot.**



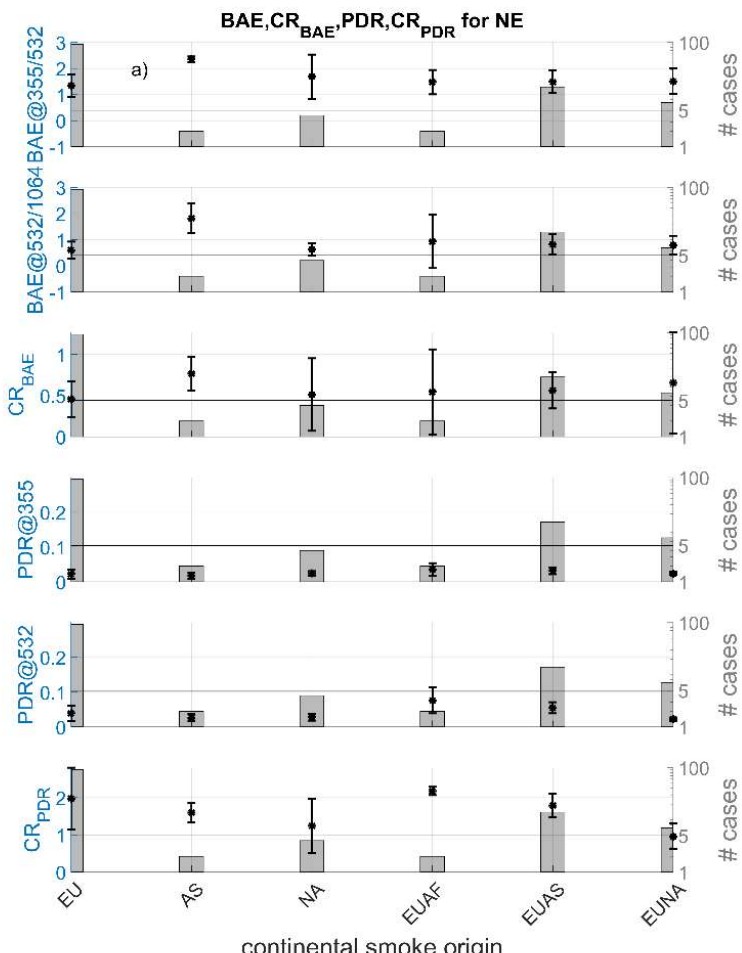



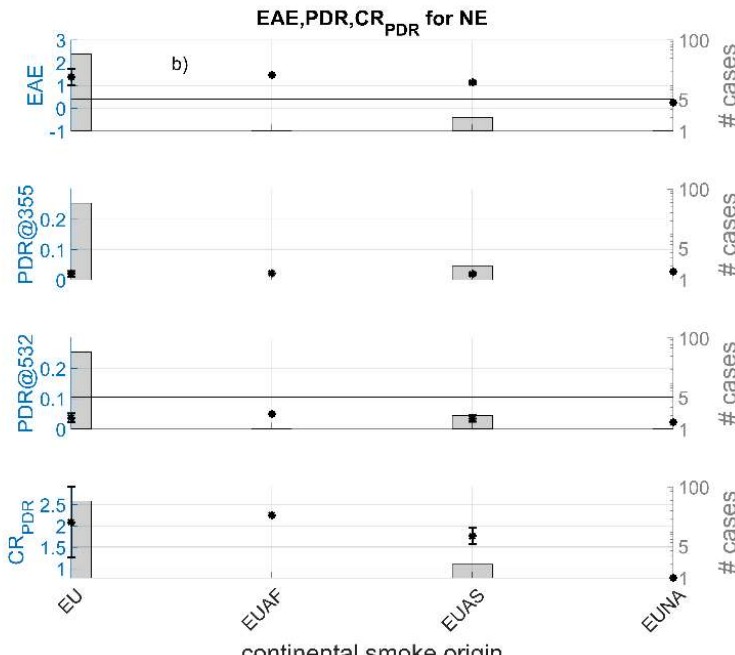





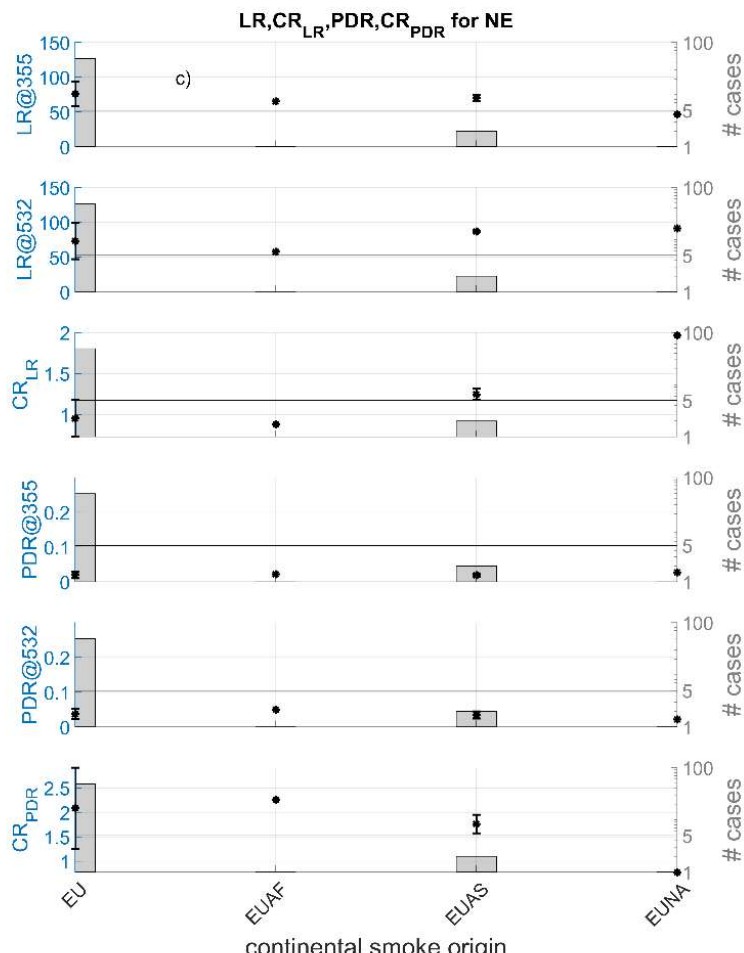





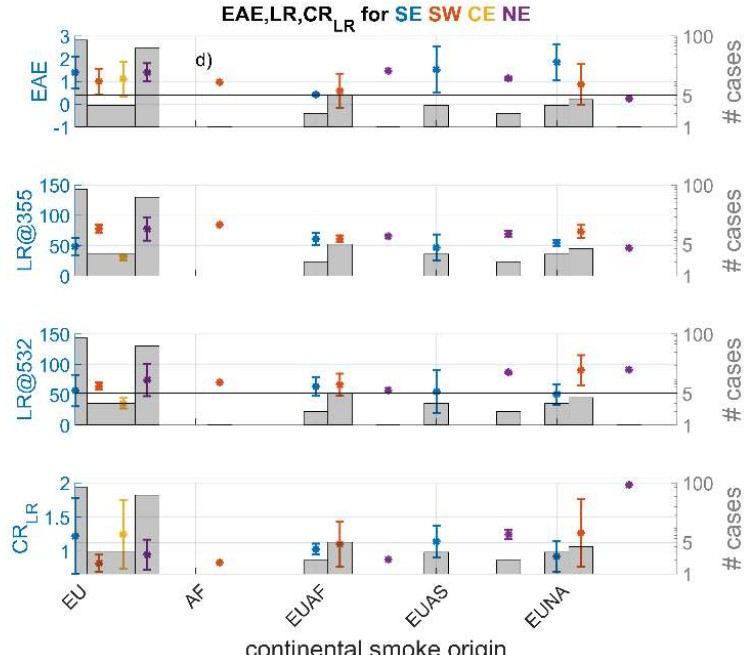





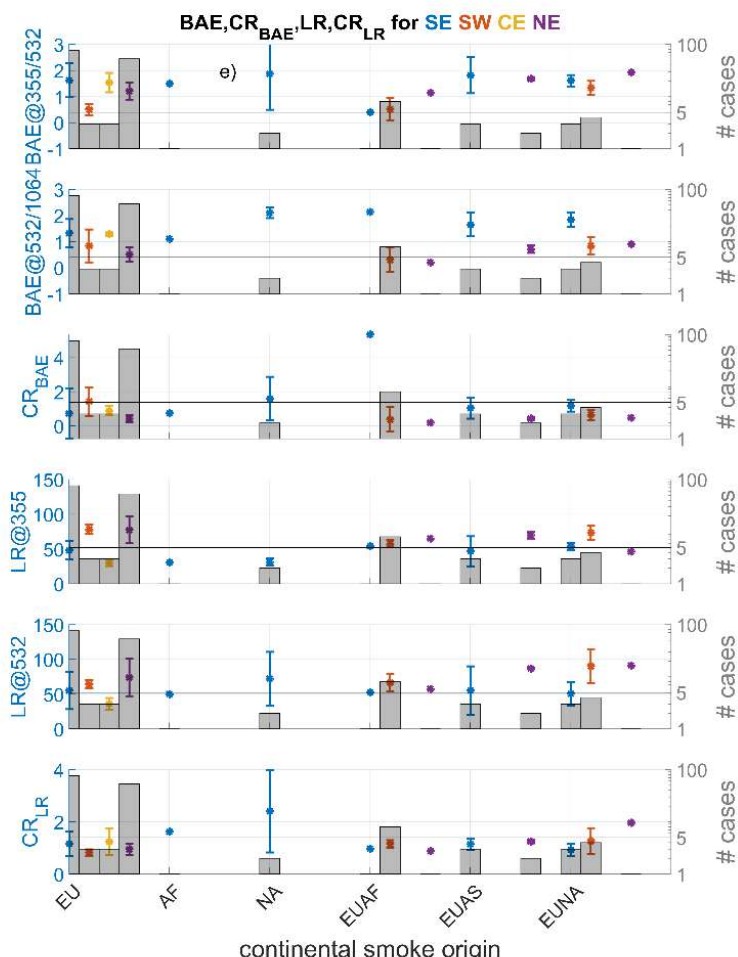





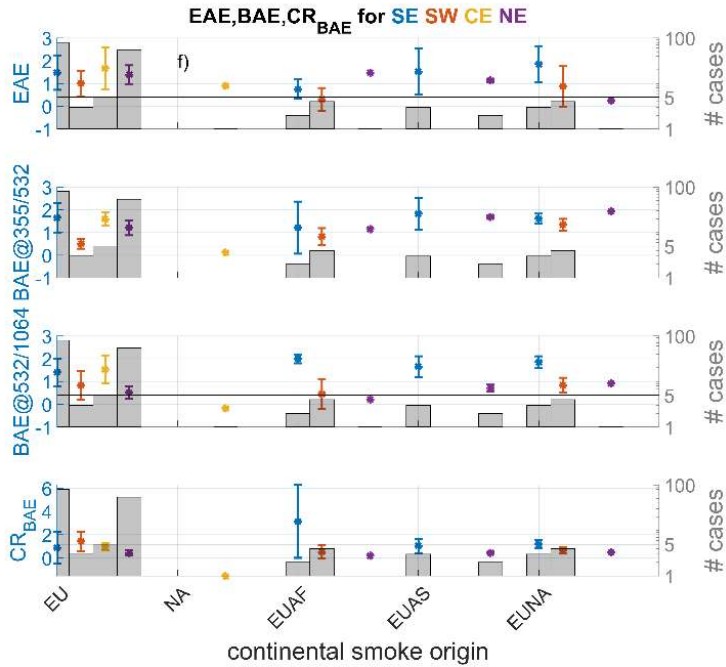

Figure 5. Intensive parameters and corresponding CR based on Fig. 4. Plots a)–f) correspond to plots a)–f) in Fig. 4.



**Table 1. Main features (mean, median, minimum, maximum values and associated uncertainties) of the intensive parameters.**

[1] 355/532; [2] 532/1064; [3] number of total values for a specific parameter. Mean values are highlighted. See stations' acronyms in Table S2.

| | | atz | brc | cog | ino | cbw | evo | gra | lei | mas | hpb | pot | the | waw |
|---|---|---|---|---|---|---|---|---|---|---|---|---|---|---|
| LR355 | #[3] | 83 | 0 | 0 | 52 | 9 | 0 | 19 | 0 | 0 | 2 | 3 | 20 | 71 |
| | Mean | 47 ±2 | | | 51 ±2 | 35 ±3 | | 65 ±2 | | | 33 ±1 | 48 ±1 | 36 ±0.4 | 73 ±2 |
| | Med | 42 ±3 | | | 52 ±1 | 37 ±1 | | 61 ±1 | | | 33 ±1 | 51 ±1 | 32 ±0.5 | 74 ±1 |
| | Min | 20 ±1 | | | 25 ±1 | 24 ±1 | | 31 ±1 | | | 32 ±1 | 43 ±1 | 21 ±1 | 30 ±1 |
| | Max | 126 ±6 | | | 90 ±11 | 44 ±1 | | 86 ±7 | | | 33 ±1 | 51 ±1 | 65 ±1 | 128 ±2 |
| LR532 | #[3] | 54 | 0 | 0 | 34 | 1 | 1 | 22 | 0 | 0 | 2 | 2 | 34 | 78 |
| | Mean | 57 ±5 | | | 57 ±1 | 46 ±2 | 36 ±9 | 68 ±3 | | | 31 ±1 | 55 ±3 | 58 ±1 | 77 ±2 |
| | Med | 50 ±1 | | | 53 ±1 | 46 ±2 | 36 ±9 | 66 ±1 | | | 31 ±1 | 55 ±1 | 50 ±1 | 72 ±1 |
| | Min | 21 ±3 | | | 29 ±1 | 46 ±2 | 36 ±9 | 40 ±4 | | | 30 ±1 | 27 ±2 | 20 ±1 | 29 ±1 |
| | Max | 142 ±8 | | | 115 ±1 | 46 ±2 | 36 ±9 | 121 ±4 | | | 33 ±1 | 82 ±3 | 133 ±3 | 146 ±2 |
| EAE[1] | #[3] | 64 | 0 | 0 | 32 | 2 | 0 | 13 | 0 | 0 | 7 | 2 | 46 | 59 |
| | Mean | 1.4 ±0.2 | | | 1.3 ±0.02 | 1.3 ±0.3 | | 0.8 ±0.1 | | | 1.7 ±0.3 | 1.4 ±0.1 | 1.6 ±0.05 | 1.4 ±0.05 |
| | Med | 1.4 ±0.5 | | | 1.2 ±0.1 | 1.3 ±0.4 | | 1 ±0.04 | | | 1.7 ±0.1 | 1.4 ±0.2 | 1.8 ±0.1 | 1.3 ±0.02 |
| | Min | -0.8 ±0.1 | | | -0.6 ±0.01 | 0.3 ±0.1 | | -0.4 ±0.1 | | | 0.8 ±0.2 | 0.6 ±0.1 | -0.9 ±0.1 | 0.3 ±0.1 |
| | Max | 2.6 ±0.3 | | | 2.6 ±0.01 | 2.3 ±0.4 | | 1.7 ±0.3 | | | 2.9 ±0.2 | 2.3 ±0.1 | 2.7 ±0.2 | 2.8 ±0.2 |
| BAE[1] | #[3] | 113 | 5 | 5 | 113 | 14 | 2 | 77 | 0 | 37 | 5 | 4 | 78 | 150 |
| | Mean | 1.6 ±0.1 | 1.4 ±0.2 | 2.1 ±0.1 | 1.4 ±0.03 | 1.8 ±0.01 | 2 ±0.03 | 1.2 ±0.04 | | 1.3 ±0.2 | 1.2 ±0.02 | 1 ±0.05 | 1.5 ±0.1 | 1.4 ±0.05 |
| | Med | 1.6 ±0.1 | 1.2 ±0.3 | 2.2 ±0.05 | 1.5 ±0.006 | 1.8 ±0.1 | 2 ±0.1 | 1.3 ±0.001 | | 1.2 ±0.2 | 1.4 ±0.01 | 0.8 ±0.1 | 1.5 ±0.1 | 1.3 ±0.15 |
| | Min | -1 ±0.01 | 1.2 ±0.03 | 1.5 ±0.1 | -0.1 ±0.0004 | 0.5 ±0.01 | 1.7 ±0.03 | -0.2 ±0.02 | | 0.4 ±0.1 | 0.1 ±0.03 | 0.7 ±0.03 | -0.8 ±0.03 | 0.4 ±0.1 |
| | Max | 2.9 ±0.01 | 1.8 ±0.02 | 2.5 ±0.1 | 2.8 ±0.1 | 2.4 ±0.03 | 2.3 ±0.04 | 2.9 ±0.03 | | 2.6 ±0.1 | 1.8 ±0.02 | 1.7 ±0.1 | 3 ±0.01 | 2.8 ±0.1 |
| BAE[2] | #[3] | 110 | 14 | 20 | 119 | 14 | 8 | 98 | 6 | 35 | 6 | 3 | 76 | 176 |
| | Mean | 1.4 ±0.04 | 1.2 ±0.04 | 1.2 ±0.1 | 1.4 ±0.02 | 1.1 ±0.05 | 1.3 ±0.1 | 1 ±0.01 | 0.9 ±0.1 | 0.8 ±0.1 | 1.2 ±0.02 | 1.3 ±0.01 | 1.3 ±0.03 | 0.7 ±0.03 |
| | Med | 1.3 ±0.1 | 1.1 ±0.4 | 1.2 ±0.2 | 1.3 ±0.01 | 1.2 ±0.1 | 1.3 ±0.2 | 1.1 ±0.04 | 1.1 ±0.1 | 0.7 ±0.03 | 1.3 ±0.1 | 1.3 ±0.01 | 1.2 ±0.2 | 0.7 ±0.11 |
| | Min | 0.8 ±0.02 | 0.7 ±0.01 | 1 ±0.1 | 0.1 ±0.003 | 0.6 ±0.04 | 0.8 ±0.3 | -0.7 ±0.04 | -0.6 ±0.1 | -0.9 ±0.1 | -0.2 ±0.02 | 1.3 ±0.02 | 0.1 ±0.002 | -0.2 ±0.03 |
| | Max | 2.8 ±0.05 | 1.8 ±0.01 | 1.5 ±.1 | 2.8 ±0.01 | 1.4 ±0.03 | 1.6 ±0.01 | 2.9 ±0.01 | 1.6 ±0.05 | 1.9 ±0.1 | 2.6 ±0.04 | 1.3 ±0.01 | 3 ±0.01 | 2.2 ±0.03 |
| P | #[3] | 0 | 0 | 0 | 0 | 0 | 0 | 0 | 0 | 0 | 0 | 0 | 0 | 132 |
| | Mean | | | | | | | | | | | | | 2.4 |

...





| | | | | | | | | | | | | | | |
|---|---|---|---|---|---|---|---|---|---|---|---|---|---|---|
| | | | | | | | | | | | | | | ±0.02 |
| | Med | | | | | | | | | | | | | 2.2 ±1 |
| | Min | | | | | | | | | | | | | 0.2 ±0.001 |
| | Max | | | | | | | | | | | | | 8.6 ±0.04 |
| PDR532 (%) | #³ | 0 | 0 | 0 | 64 | 0 | 0 | 0 | 0 | 0 | 10 | 5 | 0 | 160 |
| | Mean | | | | 6.6 ±0.3 | | | | | | 3.3 ±0.1 | 4.5 ±0.1 | | 3.9 ±0.04 |
| | Med | | | | 4.9 ±0.6 | | | | | | 2.4 ±1.3 | 4.9 ±0.1 | | 3.4 ±1.2 |
| | Min | | | | 0.04 ±0.001 | | | | | | 1.2 ±0.03 | 2.3 ±0.1 | | 0.6 ±0.01 |
| | Max | | | | 27.5 ±1.5 | | | | | | 8.1 ±0.1 | 6.1 ±0.1 | | 15.1 ±0.1 |



**Table 2. Intensive parameters for 8–10 July 2013 at Belsk ("cog"), Cabauw ("cbw") and Warsaw ("waw") stations along with information about contributing fires from North America. The measurement with all IPs available is highlighted.**

| station | t measurement | Alt [m] | LR 355 | LR 532 | EAE | BAE 355/532 | BAE 532/1064 | PDR 355 [%] | PDR 532 [%] | # fires | # fires' detections | t fires' detection dd/mm hh:mm (range) | Fires' coordinates longitude x latitude (N) (range) | Fire type* |
|---|---|---|---|---|---|---|---|---|---|---|---|---|---|---|
| | | | | | | | | 20130708 | | | | | | |
| cbw | 09:00 | 2768 | 42±2 | | | 2.3±0.01 | 1.3±0.04 | | | 1 | 1 | 17:01 01/07 | [94 93] x [33 34] | 0 |
| waw | 14:29 | 2578 | 46±1 | 91±3 | 0.3±0.1 | 1.9±0.1 | 0.9±0.03 | 2.8±0.03 | 2.2±0.02 | 2 / 1 | 4 / 2 | 04:23 30/06 / 11:31 07/07 | [-112 -111] x [61 62] / 17.988 x 59.175 | 1 |
| waw | 15:22 | 2424 | 36±1 | | | 1.6±0.1 | 0.5±0.03 | 2.9±0.03 | 2.5±0.02 | 136 / 1 | 261 / 2 | 17:49 30/06 05:04 01/07 / 11:31 07/07 | [-101 -94] x [57 59] / 17.988 x 59.175 | 1 |
| waw | 15:59 | 2406 | 30±1 | | | 2.2±0.04 | 0.7±0.03 | 2.3±0.02 | 2.4±0.02 | 1 / 1 | 2 / 2 | 17:47 02/07 / 11:31 07/07 | [-78 -77] x [54 55] / 17.988 x 59.175 | 1 |
| waw | 16:29 | 1315 | | | | 0.5±0.1 | 0.6±0.03 | 1.7±0.02 | 3.9±0.04 | 37 | 45 | 17:50 30/06 03:26 01/07 | [-102 -99] x [58 59] | 0 |
| cog | 17:53 | 1245 | | | | 1.9±0.1 | 1±0.1 | | | 1 / 1 / 1 | 1 / 1 / 1 | 01:34 03/07 / 17:47 02/07 / 11:31 07/07 | [-72 -71] x [52 53] / [-78 -77] x [54 55] / 17.988 x 59.175 | 1 |
| waw | 20:25 | 1999 | | | | 1.9±0.04 | 1.3±0.02 | | | 1 / 1 | 1 / 2 | 18:38 01/07 / 11:31 07/07 | [-92 -91] x [35 36] / 17.988 x 59.175 | 1 |
| | | | | | | | | 20130709 | | | | | | |
| waw | 6:29 | 2391 | | | | | 0.7±0.02 | 2.8±0.02 | 3.2±0.02 | 9 | 18 | 05:05 01/07 | [-114 -113] x [60 61] | 0 |
| cog | 11:46 | 1163 | | | | | 1.5±0.1 | | | 1 | 1 | 17:47 02/07 | [-78 -77] x [54 55] | 0 |
| cog | 11:46 | 2100 | | | | | 1.4±0.1 | | | 62 | 96 | 03:26 01/07 19:27 02/07 | [-114 -95] x [57 61] | 0 |
| cbw | 13:00 | 2325 | 37±1 | | | 1.9±0.01 | 1.4±0.03 | | | 48 | 62 | 20:23 01/07 17:37 02/07 | [-114 -100] x [58 61] | 0 |
| cbw | 13:00 | 3525 | 31±1 | | | 2.2±0.01 | 1.2±0.04 | | | 23 | 30 | 04:50 03/07 20:09 03/07 | -104 -95] x [ 52 57] | 0 |



| | | | | | | | | | | | | | | |
|---|---|---|---|---|---|---|---|---|---|---|---|---|---|---|
| cbw | 13:00 | 4714 | 32±10 | | | 2.4±0.03 | 0.8±0.1 | | | 1 | 1 | 19:20 02/07 | [-93 -92] x [30 31] | 0 |
| cbw | 15:00 | 1883 | 38±0.4 | | | 1.4±0.01 | 1.3±0.05 | | | 27 | 29 | 05:06 01/07 04:09 02/07 | [-115 -99] x [58 64] | 0 |
| cbw | 15:00 | 3206 | 44±1 | | | 1.7±0.01 | 1.2±0.04 | | | 37 | 74 | 20:23 01/07 18:31 03/07 | [-110 -92] x [56 61] | 0 |
| cbw | 17:00 | 1826 | | | | | 1.1±0.03 | | | 41 | 41 | 08:00 04/07 15:56 04/07 | [-70 -64] x [52 53] | 0 |
| cog | 17:50 | 2123 | | | | | 1.1±0.1 | | | 21 1 | 40 2 | 03:26 01/07 09:08 01/07 11:31 07/07 | [-102 -95] x [58 59] 17.988 x 59.175 | 1 |
| cbw | 19:00 | 1999 | 43±0.2 | | | 1.8±0.004 | 1.3±0.02 | | | 677 | 1003 | 20:10 01/07 17:34 04/07 | [-126 -65] x [51 64] | 0 |
| waw | 19:29 | 2114 | | | | 1.6±0.04 | 0.3±0.02 | 3.1±0.02 | 2.8±0.03 | 1 | 2 | 08:55 03/07 | [-96 -95] x [58 59] | 0 |
| waw | 21:07 | 5103 | | | | 0.6±0.05 | 1.2±0.03 | 2.4±0.02 | 1.1±0.01 | 68 | 97 | 16:42 03/07 15:47 04/07 | [-96 -66] x [54 57] | 0 |
| waw | 21:59 | 5200 | | | | 1.3±0.04 | 0.9±0.03 | 2.7±0.02 | | 92 | 147 | 20:10 01/07 17:34 04/07 | [-122 -65] x [51 62] | 0 |
| waw | 22:52 | 5267 | | | | 2.3±0.04 | 0.9±0.02 | 2.2±0.02 | 1.5±0.01 | 16 | 17 | 18:33 01/07 20:10 03/07 | [-116 -92] x [55 60] | 0 |
| 20130710 | | | | | | | | | | | | | | |
| cog | 9:26 | 2040 | | | | | 1.2±0.04 | | | 36 | 47 | 05:48 02/07 | [-120 -118] x [60 61] | 0 |
| cog | 18:03 | 5415 | | | | | 1.2±0.05 | | | 7 | 14 | 17:34 04/07 | [-72 -71] x [51 52] | 0 |
| cog | 18:34 | 5220 | | | | | 1.1±0.1 | | | 27 | 40 | 02:59 05/07 | [-66 -65] x [51 52] | 0 |
| cog | 18:34 | 6270 | | | | | 1±0.1 | | | 10 | 13 | 19:20 02/07 18:29 03/07 | [-92 -74] x [34 52] | 0 |
| waw | 18:59 | 3437 | | | | 0.5±0.05 | 0.9±0.03 | | | 3 | 6 | 16:42 03/07 | [-93 -92] x [56 57] | 0 |
| cog | 19:05 | 5273 | | | | 2.5±0.1 | 1.4±0.1 | | | 9 | 10 | 02:59 05/07 07:06 05/07 | [-77 -73] x [51 52] | 0 |
| cog | 19:37 | 3263 | | | | | 1±0.1 | | | 29 | 38 | 01:21 05/07 02:59 05/07 | [-67 -64] x [52 53] | 0 |





| cog | | | | | | | 1.2± | | | 3 | 6 | 04:45 03/07 | [-95 -94] x [34 35] | |
|-----|-------|------|--|--|--|--|------|--|--|-----|-----|-------------|---------------------|---|
| | 19:37 | 5310 | | | | | 0.1 | | | | | | | 0 |
| cog | | | | | | | | | | 565 | 940 | 16:43 03/07 | [-78 -66] x [52 50] | |
| | 20:08 | 1358 | | | | | 1±0.1 | | | | | 06:23 04/07 | | 0 |

*0 = 'pure North America' smoke, 1 = 'mixed' smoke (North America and local)



Table 3. Long range transport events with fire sources in North America. Mean and STD of the intensive parameters. The number of cases available is given in parenthesis (# cases).

| Intensive parameter | LR 355 mean ± STD [sr] | LR 532 mean ± STD [sr] | EAE 355/532 mean ± STD | BAE 355/532 mean ± STD | BAE 532/1064 mean ± STD | PDR 355 mean ± STD [%] | PDR 532 mean ± STD [%] |
|---|---|---|---|---|---|---|---|
| All | 46 ± 16 | 66 ± 32 | 1.1 ± 0.9 | 1.7 ± 0.6 | 1.2 ± 0.5 | 2.4 ± 0.5 | 3.1 ± 2.2 |
| (# cases) | (18) | (10) | (9) | (53) | (74) | (10) | (10) |
| Pure NA | 42 ± 14 | **54** ± 30 | 1.3 ± 0.7 | **1.6** ± 0.7 | **1.3** ± 0.6 | **2.5** ± 0.5 | **3.6** ± 2.9 |
| (# cases) | (11) | (5) | (4) | (32) | (48) | (6) | (6) |
| Mixed | **52** ± 17 | 78 ± 32 | **1** ± 1.2 | 1.8 ± 0.5 | 1.1 ± 0.3 | 2.4 ± 0.6 | 2.4 ± 0.2 |
| (# cases) | (7) | (5) | (5) | (21) | (26) | (4) | (4) |
| Lit res* | 49 ± 9.2 | 61 ± 5 | 0.9 ± 0.5 | 1.45 ± 0.2 | 1.4 ± 0.1 | 17 ± 2.3 | 9.1 ± 1.8 |
| (# cases) | (15) | (18) | (8) | (8) | (10) | (6) | (13) |
| min, max | 23, 100 | 30, 129 | -0.3, 2.2) | 0.5, 2.1) | 0.6, 2.85 | 2.1. 28) | 2, 20 |

*Literature research: according to references 4,8,12,13,14,16,17,19,20,22,23,26,27,32,39,40,42** from Table S2, Part I (Adam et al., 2020). Present mean values are the averages over the values reported (see text). Minimum and maximum values are reported as well.

** 4.    Ancellet, G., Pelon, J., Totems, J., Chazette, P., Bazureau, A., Sicard, M., Di Iorio, T., Dulac, F., and Mallet, M.: Long-range transport and mixing of aerosol sources during the 2013 North American biomass burning episode: analysis of multiple lidar observations in the western Mediterranean basin, Atmos. Chem. Phys., 16, 4725–4742, doi:10.5194/acp-16-4725-2016, 2016.

8.    Burton, S. P., Ferrare, R. A., Hostetler, C. A., Hair, J. W., Rogers, R. R., Obland, M. D., Butler, C. F., Cook, A. L., Harper, D. B., and Froyd, K. D.: Aerosol classification using airborne High Spectral Resolution Lidar measurements – methodology and
examples, Atmos. Meas. Tech., 5, 73–98 , doi:10.5194/amt-5-73-2012, 2012.

12.    Giannakaki, E., van Zyl, P. G., Müller, D., Balis, D., and Komppula, M.: Optical and microphysical characterization of aerosol layers over South Africa by means of multi-wavelength depolarization and Raman lidar measurements, Atmos. Chem. Phys., 16, 8109–8123, doi:10.5194/acp-16-8109-2016, 2016.

13.    Groß, S., Esselborn, M., Weinzierl, B., Wirth, M., Fix, A, and Petzold, A.: Aerosol classification by airborne high spectral
resolution lidar Observations, Atmos. Chem. Phys., 13, 2487–2505, doi:10.5194/acp-13-2487-2013, 2013.

14.    Haarig, M., Ansmann, A., Baars, H., Jimenez, C., Veselovskii, I., Engelmann, R., and Althausen, D.: Depolarization and lidar ratios at 355, 532, and 1064 nm and microphysical properties of aged tropospheric and stratospheric Canadian wildfire smoke, Atmos. Chem. Phys., 18, 11847-11861, https://doi.org/10.5194/acp-18-11847-2018, 2018.

16.    Heese, B., and Wiegner, M.: Vertical aerosol profiles from Raman polarization lidar observations during the dry season
AMMA field campaign, J. Geophys. Res., 113, D00C11, doi:10.1029/2007JD009487, 2008.

17.    Hu, Q., Goloub, P., Veselovskii, I., Bravo-Aranda, J.-A., Popovici, I., Podvin, T., Haeffelin, M., Lopatin, A., Pietras, C., Huang, X., Torres, B., and Chen, C.: A study of long-range transported smoke aerosols in the Upper Troposphere/Lower Stratosphere, Atmos. Chem. Phys., Atmos. Chem. Phys., 19, 1173–1193, https://doi.org/10.5194/acp-19-1173-2019, 2019.

19.    Janicka, L., Bockmann, C., Wang, D., Stachlewska, I. S.: Lidar derived fine scale resolution properties of tropospheric
aerosol mixtures, ILRC29, S2-122, Hefei, China, 2019.





20. Janicka, L., Stachlewska, I. S., Veselovskii, I., Baars, H.: Temporal variations in optical and microphysical properties of mineral dust and biomass burning aerosol derived from daytime Raman lidar observations over Warsaw, Poland, Atmos. Environ., 169, 162-174, http://dx.doi.org/10.1016/j.atmosenv.2017.09.022, 2017.

22. Mattis, I., Ansmann, A., Wandinger, U., and Müller, D.: Unexpectedly high aerosol load in the free troposphere over central Europe in spring//summer 2003, G.R.L., 30, 2178, doi:10.1029/2003GL018442, 2003.

23. Mattis, I., Müller, D., Ansmann, A., Wandinger, U., Preißler, J., Seifert, P., and Tesche, M.: Ten years of multiwavelength Raman lidar observations of free-tropospheric aerosol layers over central Europe: Geometrical properties and annual cycle, J. Geophys. Res., 113, D20202, doi:10.1029/2007JD009636, 2008.

26. Müller, D., Kolgotin, A., Mattis, I., Petzold, A., and Stohl, A.: Vertical profiles of microphysical particle properties derived from inversion with two-dimensional regularization of multiwavelength Raman lidar data: experiment, Appl. Opt., 50, 2069-2079, 2011.

27. Murayama, T., Müller, D., Wada, K., Shimizu, A., Sekiguchi, M., and Tsukamoto, T.: Characterization of Asian dust and Siberian smoke with multiwavelength Raman lidar over Tokyo, Japan in spring 2003, Geophys. Res. Lett., 31, L23103, doi:10.1029/2004GL021105, 2004.

32. Mylonaki, M., Papayannis, A., Mamouri, R., Argyrouli, A., Kokkalis, P., Tsaknakis, G., and Soupiona, O.: Aerosol optical properties variability during biomass burning events observed by the EOLE-AIAS depolarization lidars over Athens, Greece (2007-2016), 28th ILRC, Bucharest, Romania, 2017.

39. Stachlewska, I. S., Zawadzka, O., and Engelmann, R.: Effect of HeatWave Conditions on Aerosol Optical Properties Derived from Satellite and Ground-Based Remote Sensing over Poland, Remote Sens., 9, 1199; doi:10.3390/rs9111199 www.mdpi.com/journal/remotesensing, 2017.

40. Stachlewska, I. S., Samson, M., Zawadzka, O., Harenda, K. M., Janicka, L., Poczta, P., Szczepanik, D., Heese, B., Wang, D., Borek, K., Tetoni, E., Proestakis, E., Siomos, N., Nemuc, A., Chojnicki, B. H., Markowicz, K. M., Pietruczuk, A., Szkop, A., Althausen, D., Stebel, K., Schuettemeyer, D., and Zehner, C.: Modification of Local Urban Aerosol Properties by Long-Range Transport of Biomass Burning Aerosol, Remote Sens., 10, 412; doi:10.3390/rs10030412, 2018.

42. Tesche, M., Müller, D., Gross, S., Ansmann, A., Althausen, D., Freundenthaler, V., Weinzierl, B., Veira, A. and Petzold, A.: Optical and microphysical properties of smoke over Cape Verde inferred from multiwavelength lidar measurements, Tellus, 63B, 677-694, DOI: 10.1111/j.1600-0889.2011.00549.x, 2011.





**Table 4. Smoke characteristics based on CR$_{LR}$ and EAE for each measurement region (SE, SW, NE and CE) and each source region (EU, AF, AS, NA, EUAF, EUAS, EUNA) based on scatter plots in Fig. 4.**

| | CR$_{LR}$ | EAE | LR532 (sr)* | LR355 (sr)* | CR$_{BAE}$** | Comments | Smoke type based on CR$_{LR}$ and EAE*** | Absorption at 532nm and 355nm**** |
|---|---|---|---|---|---|---|---|---|
| **EU** | | | | | | | | |
| SE | 1.2 | 1.4 | 57 | 48 | 0.8 | 81 meas. CR$_{LR}$ and EAE | aged | Medium |
| SW | 0.8 | 1 | 64 | 78 | 1.4 | 3 meas. CR$_{LR}$ and EAE RH=68–70%. | aged | Rel. high at 532nm High at 355nm |
| CE | 1.2 | 1.1 | 36 | 30 | 0.9 | 3 meas. CR$_{LR}$ and EAE | aged | Low |
| NE | 1 | 1.4 | 74 | 78 | 0.4 | 54 meas. CR$_{LR}$ and EAE CR$_{PDR}$ > 1 | fresh/aged | High |
| **AF** | | | | | | | | |
| SE | 1.6 | | 50 | 31 | 0.74 | 1 meas. CR$_{LR}$ and CR$_{BAE}$ | aged | Medium at 532nm Low at 355nm |
| SW | 0.8 | 1 | 70 | 84 | | 1 meas. CR$_{LR}$ and EAE. RH=73%. | aged | High |
| **NA** | | | | | | | | |
| SE | 2.4 | | 72 | 32 | 1.6 | 2 meas. CR$_{LR}$ based on CR$_{LR}$ versus CR$_{BAE}$ | aged | High at 532nm Low at 355nm |
| CE | | 0.9 | | | -1.6 (-0.2 / 0.1) | 1 meas. EAE and CR$_{BAE}$. More backscattering at 1064nm. | aged | |
| **EUAF** | | | | | | | | |
| SE | 1 | 0.4 | 64 | 61 | 3.2 | 2 meas. CR$_{LR}$ and EAE | aged | Rel. high |
| SW | 1.1 | 0.6 | 67 | 61 | 0.5 | 5 meas. CR$_{LR}$ and EAE | aged | Rel. high |
| NE | 0.9 | 1.5 | 57 | 65 | 0.2 | 1 meas. CR$_{LR}$ and EAE larger EU contribution CR$_{PDR}$ > 1 | fresh | Medium at 532nm Rel. high at 355nm |
| **EUAS** | | | | | | | | |
| SE | 1.1 | 1.5 | 55 | 47 | 1 | 3 meas. CR$_{LR}$ and EAE larger EU contribution | fresh/aged | Medium |
| NE | 1.1 | 1.1 | 87 | 70 | 0.4 | 2 meas. CR$_{LR}$ and EAE CR$_{PDR}$ > 1 | aged | Very high at 532nm High at 355nm |
| **EUNA** | | | | | | | | |
| SE | 0.9 | 1.9 | 51 | 54 | 1.2 | 3 meas. CR$_{LR}$ and EAE larger EU contribution | fresh | Medium |
| SW | 1.3 | 0.9 | 90 | 73 | 0.6 | 4 meas. CR$_{LR}$ and EAE | aged | Very high at 532nm High at 355nm |
| NE | 2 | 0.3 | 91 | 46 | 0.5 | 1 meas. CR$_{LR}$ and EAE CR$_{PDR}$ < 1 | aged | Very high at 532nm Medium at 355nm |

5  * corresponding to CR$_{LR}$; ** based on CR$_{BAE}$ versus CR$_{LR}$ and/or EAE; *** Based on scatter plot between EAE and CR$_{LR}$ where available (Fig. 4, upper right-hand side); **** LR is considered low for [30,40]sr, medium for [40,60]sr, relatively high (Rel. high) for [60,70]sr, high for [70.80]sr, very high for >80sr.