# Peer review of "Methodology for data analyses"

_Atmospheric Chemistry and Physics, 2020_

## Referee Comment (RC1) · Anonymous Referee #1 · 5 Oct 2020

This paper presents observations of biomass burning observations, specifically aerosol intensive parameters, from a regional lidar network, EARLINET. The paper then discusses their origins as determined by back trajectory analysis. This paper is a companion to an earlier paper (acp-2020-320) which described the methodology for QC on this network of instruments of disparate data quality. This is certainly an important topic; however, I found the paper to be difficult to follow and review, first because the paper seems to have several different opinions about what its topic is, and second because the limited data available makes broader interpretation challenging. I would recommend the authors consider what exactly they want to convey from this paper, and make sure that the paper accomplishes this.

I have some further concerns about the analysis presented in the paper, especially

much of Section 4 (4.2 and 4.3), again largely due to the limited data available for analysis. Of course this doesn't mean analysis can't be performed, or that the results are meaningless, but only that they must be more carefully considered, and I would like to see much more to that effect (maybe some of this is in there; if so, I couldn't easily follow it). NB: I have not reviewed the companion paper, but I have skimmed it to try to see whether I've missed some information which would address these concerns, but I wasn't able to find much that was relevant. The topic presented here is interesting, but the paper as written does not address it in a satisfactory manner. At the very least I think major revisions are in order before it can be considered for publication.

—

Specific comments, structure and focus:

–The title and the abstract seem to be describing two different papers. The title talks about biomass burning events as a whole, but the abstract focuses heavily on BB due to long-range transport, and specifically LRT from North America. Why this framing? And then according to the first sentence of Section 3.2 (and the first page of 4.1), N America isn't even half of the LRT cases? Although this sentence says "We encountered 168 measurements over the 24 LRT periods. . . from these measurements, 77 have a North American origin and 91 have different BB origin (local)." Are the 168 measurements then all long-range, or also local? In other parts of the paper it seems that this "local" transport is also included under the "LRT" category. I would have thought these would be two distinct classifications. There's also a discussion of transport from Asian and African sources, so why is the abstract solely focused on N Am sources?

–The abstract also doesn't make any mention of the case study [edit: studies] which is discussed in Section 3.1. Why not? Isn't this a big part of this paper?

– I found it very difficult to follow all the multiple sets of acronyms used (site names, groups of sites, plume origins, parameter names. . .). A list of the acronyms is provided, but it's in the last page of a separate supplement file, rather than in an appendix or even

in the main text. This really inhibited my ability to understand the paper. I would suggest at the very least moving the acronym list to an appendix within the paper, if not to the main text itself. I think the authors rely on these abbreviations enough that it would belong in the main text.

– Does Figure 3 use one color to indicate two completely different things (e.g. am I reading this right that WAW and AS are both marigold, but the marigold asterisks are not what's used to determine the marigold circle-whisker?)? This is unnecessarily confusing. Pick different colors or consider e.g. using shape distinctions for one or the other. (this comment refers to the final pattern of Fig 3... I guess the previous page is also Fig 3).

– On that subject, a lot of the figures are simply too large, and the captions are insufficient to describe them. There are also a whole lot of supplementary figures too. Consider splitting them into smaller sets of figures, or reassess whether the information really needs to be presented this way. Figures that span 3 pages with a single (small) caption are very confusing to follow. Especially e.g. cases such as Figure 4: you conclude there is no correlation between the variables presented in panels d-f. Maybe this isn't necessary to show at all, then?

—

Specific comments, scientific analysis:

– perhaps this was addressed in Part I, although I found no mention of it: it seems this classification is entirely determined based on one run of the HYSPLIT trajectory model run over ten days. What have you done to assess whether these trajectories are robust? Did you run it in ensemble mode to verify whether the trajectories were consistent? Or did you run the model forward from a fire location to see if the airmasses ended up in roughly the same location? Perhaps the authors have done this, but I would like to see much more information regarding how they addressed the uncertainties inherent in HYSPLIT or any trajectory model. Especially since many of the conclusions

presented here are based on very few (sometimes just one) points, I think this is a necessary exercise.

Related, p.6 Line 11: "Moreover, the ensemble backtrajectory for 3000 m a.g.l. at 19:00 using GDAS 0.5 showed more backtrajectories towards North America." More appropriate would probably be something like "the ensemble back trajectory corroborated that the air mass originated in N America," if that's what the ensemble trajectory showed. Consistent ensemble paths back to NA would give extra confidence to the result that this air mass originated there, but doesn't necessarily mean that more air came from there. Conversely, if the ensemble shows e.g., half from NA and half from Asia, that would indicate the HYSPLIT trajectory is not particularly robust in this case and we may need to reassess that confidence. I don't see much other text indicating whether these types of tests have been performed. I do notice that p.6L14-15 seems to say that "the results are not in agreement" [for two different configurations of HYSPLIT trajectories]? Is that to what this sentence is referring? Then how can you believe them? Maybe I'm misunderstanding this section. . .

– I don't follow the logic behind Section 4.3. Examining means can be a decent way to analyze two different populations, but in this case (with very small sample sizes and large ranges between them; e.g. the error/range points/whiskers presented in Figs 3, 4, 5) I'm not convinced studying just the means is meaningful. For that to be the case, you'd want to have populations where the properties were known to be 1) normally distributed and 2) fairly uniform, but as the authors themselves state on p. 17 (Lines 14-17), these episodes are a mixture of several fires already. So this doesn't seem a good approach to me. Perhaps medians/percentiles, or simply present the ranges (rather than stdev) especially when the points are so few?

Plus, if you're trying to draw conclusions based on 2 points of one case and 1 of another, this doesn't seem statistically robust or generalizable. That's not to say that minimal data can't be useful, but its limitations need to be acknowledged. I think the data presented in Figure 5 are actually quite useful to that end, but again, I found

the presentation quite confusing (log-y axis on one side, lin-y axis on the other, with multiple colors and a not very informative caption).

Perhaps focusing on additional case studies would be a clearer presentation for the cases outside EU-origin.

– in Figure 4 you present values of R=1. This is because you are fitting through only 3 (2?) points. This R is meaningless in this context, especially since one has very large uncertainty bars and the other has only one point.

– Section 4.2: the authors draw conclusions that e.g., EAE values of 1.4 indicates a mixture of fresh and aged smoke, and 1.2 is only aged smoke. With such a small difference between the two, I'm not convinced this is robust; e.g., even the STD values in Table 3 are uniformly greater than this. What are the ranges?

– Further, (and assuming the above issue can be resolved) in Table 4 all except for three categories are classified as "aged," with two more "fresh/aged" and one "fresh." This is one of the conclusions in the abstract as well, but I'm not seeing any definition of what "aged" means in this case. And if this is truly the case, the classification of "aged" corresponding to different regions and origins is not very useful at all. Aerosol aging especially of BB can mean different things on different timescales (e.g. Haywood et al. https://doi.org/10.1029/2002JD002226 defined "fresh" as only a few minutes after emission), and a few days old likely won't be the same as a week old, so what exactly is meant by this? It may be more instructive to only focus on the cases which are *not* aged and examine what distinguishes them from the other cases. Or, conversely, if you're using back trajectories, can these determine exactly how "aged" (= time from emission) each population is? That might actually be more instructive than classifying observations as "aged" based on (as I understand it based on p.5L26) the EAE and CR, and could potentially allow for discussion of the property of BB different ages, rather than the regional Europe division.

– In the abstract, findings i) and ii) are saying basically the same thing, but inverted.

And (related to the previous comment), "travel time" might more appropriately be called "aerosol age" (="time from emission"). But, I don't really see any discussion of this beyond the case studies in Section 3 (and maybe a statement p.17L1-2, that it was aged based on high RH. . . I don't know that this is always the case, e.g. African biomass burning in particular can see high humidity very near emission). And regarding the conclusion in the abstract and in Section 4.3.1, "A slight decrease of the CRPDR with travel time was observed, while the CRBAE maintained similar values for all the source regions," I'm not clear whether this can be concluded from the data as presented. Was the same plume observed from multiple stations along its trajectory? Otherwise it's hard to say whether the initial aerosol properties were the same, or whether they were different to begin with. I'd like to see more clarification as to how this conclusion was reached.

–p.6,L19-20: "IPs STD represents $\sim$ 15, 38, 27, 17 and 37 % of the mean, respectively, and thus we claim a relatively small variability over the whole three days of measurement." I don't follow this. First it will depend what the mean value and what dynamic range is typically expected from a particular parameter, and second, with only 8-13 measurements for several of these parameters, I'm not sure stdev is the best metric for variability. What's the range, or maybe (for the 31-case BAE) percentiles?

–p.8,L10+: As above, what is the confidence on the back trajectories from NAm vs elsewhere? Further, I think this paragraph ("we noted several IP values. . . outside the range reported") refers to both NA and NA+local mixed aerosol? What fraction of each is included under "mixed" conditions? Is it relatively constant, or does the percentage vary for different cases? Was there a threshold (e.g., only 5% local) below which EUNA->just NA? I didn't see this in the paper. . . and while certainly it's possible to report values outside what's been previously reported, it seems prudent to discuss how much mixing and how much confidence goes into the present estimates, where they disagree. (Relatedly, what are the asterisk vs circle for the literature numbers in the Fig 2 panels? I don't see this described anywhere.)

–conclusions, p. 16,L23-25: with the limited amount of data available, I think this is too strong of a statement.

–another couple points of clarification: when the authors say "fire" is identified by being within 100km,+/-1h from the trajectory (p.3L23), were altitude thresholds applied or not; i.e., if HYSPLIT trajectories were at 8km, was it still considered to be a smoke source even for surface-confined, non-pyrocb fires? Also, are the "total fires" on p.5L~18 distinct fires, or just MODIS fire detections at a given time? (may be semantics, but surely a single fire will often be detected multiple times during multiple overpasses; are these considered distinct fires in these summaries?

––

Other comments:

– Q: What is red in Table 2? A: local fire; this is buried in the text, add it to the caption.

– p.4L18: "same graphics as in Section 3" but this is Section 3?

– p.6,L21: "better sphericity" => "greater sphericity"?

– p.6,L23: "increase in LR" relative to what? Not clear.

– p.6,L27: If this event occurred on 14July, why is it within a section labeled 0708-0710? Expand the 3.1 title or make a new subsection. Same for the following paragraph, now you're talking about the 2017-2018 event in this same section?

–p.7L11: suggest move the Peterson reference up to L4.

–p.11,L8-13: this paragraph in particular was pretty incomprehensible to me. There are some other phrasings throughout which are difficult to parse as well; I haven't listed them all.

---

## Referee Comment (RC2) · Anonymous Referee #2 · 5 Oct 2020

The manuscript is the second part of a broader series where long range transport and local biomass burning events are detected and characterized through EARLINET - ACTRIS lidar network observations in Europe.

Despite the importance of the subject under discussion, the paper is not introducing anything new at this stage compared with the other manuscript already published. Biomass burning events have been extensively characterized by lidar observations over the past two decades. This manuscript, at present, reads as a dull and sometimes hard-to-follow laundry list of individual biomass burning events distinguished by some ambiguous set of common characteristics. Instrument networks are of fundamental importance to monitoring aerosol optical, geometrical and microphysical characteristics, and thus measurements and results cannot be reduced to such trivialization. The

paper is further missing compulsory context, as in who is going to benefit from these observations and how the article improves our knowledge on the subject? Taken as a whole, the paper is more of a technical report that important contribution to the literature. The paper does not, therefore, clear the bar for advocacy of publication and need major revisions before publication.

In the manuscript, it is often cited that the increase in lidar ratio is linked to a higher absorption of the aerosols. The authors cannot assume that the size distribution is unchanged? It would be very interesting to pair lidar data with AERONET observations for a case study. The synergy among the two instruments could help to better characterize the microphysical elements in these events.

The manuscript even if "Part II", should be able to stand alone. The majority of the acronyms are not defined and left to reader interpretation.

Specific comments are found in the attached file.

Please also note the supplement to this comment:
https://acp.copernicus.org/preprints/acp-2020-647/acp-2020-647-RC2-supplement.pdf

[Figure]

**Biomass burning events measured by lidars in EARLINET. Part II. Results and discussions.**

Mariana Adam[1], Doina Nicolae[1], Livio Belegante[1], Iwona S. Stachlewska[2], Lucja Janicka[2], Dominika Szczepanik[2], Maria Mylonaki[3], Christiana Anna Papanikolaou[3], Nikos Siomos[4], Kalliopi Artemis Voudouri[4], Luca Alados-Arboledas[5], Juan Antonio Bravo-Aranda[5], Arnoud Apituley[6], Nikolaos Papagiannopoulos[7], Lucia Mona[7], Ina Mattis[8], Anatoli Chaikovsky[9], Michaël Sicard[10,11], Constantino Muñoz-Porcar[10], Aleksander Pietruczuk[12], Daniele Bortoli[13], Holger Baars[14], Ivan Grigorov[15], Zahary Peshev[15]

[1]National Institute for R&D in Optoelectronics, Magurele, 077225, Romania
[2]Faculty of Physics, University of Warsaw, 02-093, Poland
[3]National Technical University of Athens, Department of Physics, Athens, 15780, Greece
[4]Laboratory of Atmospheric Physics, Aristotle University of Thessaloniki, Thessaloniki, 54124, Greece
[5]Andalusian Institute for Earth System Research, Department of Applied Physics, University of Granada, Granada, 18071, Spain
[6]KNMI – Royal Netherlands Meteorological Institute, De Bilt, 3731, the Netherlands
[7]Consiglio Nazionale delle Ricerche - Istituto di Metodologie per l'Analisi Ambientale (CNR-IMAA), C.da S.Loja. Tito Scalo (PZ), Italy
[8]Deutscher Wetterdienst, Meteorologisches Observatorium Hohenpeißenberg, Hohenpeißenberg, 82383 Germany
[9]Institute of Physics, NAS of Belarus, Minsk, 220072, Belarus
[10]Remote Sensing Laboratory/CommSensLab, Universitat Politecnica de Catalunya, Barcelona, 08034, Spain
[11]Ciencies i Tecnologies de l'Espai - Centre de Recerca de l'Aeronautica i de l'Espai/Institut d'Estudis Espacials de Catalunya (CTE-CRAE/IEEC), Universitat Politecnica de Catalunya, Barcelona, 08034, Spain
[12]Institute of Geophysics, Polish Academy of Sciences, Warsaw, 01-452, Poland
[13]Earth Sciences Institute, Physics Department, University of Évora, Évora, 7000, Portugal
[14]Leibniz Institute for Tropospheric Research, Leipzig, 04318, Germany
[15]Institute of Electronics, Bulgarian Academy of Sciences, 1784, Sofia

*Correspondence to*: Mariana Adam (mariana.adam@inoe.ro)

**Abstract.** Biomass burning events are analysed using the European Aerosol Research Lidar Network database for atmospheric profiling of aerosols by lidars. Atmospheric profiles containing forest fires layers were identified in data collected by fourteen stations during 2008–2017. The data ranged from complete data sets (particle backscatter coefficient, extinction coefficient and linear depolarization ratio) to single profiles (particle backscatter coefficient). The data analysis methodology was described in Part I (*Biomass burning events measured by lidars in EARLINET. Part I. Data analysis methodology*, under discussions to ACP, the EARLINET special issue). The results are analysed by means of intensive parameters in the following directions: I) long range transport of smoke particles from North America (here, we divided the events into 'pure North America' and 'mixed'-North America and local) smoke groups, and II) analysis of smoke particles over four geographical regions (SE Europe, NE Europe, Central Europe and SW Europe). 24 events were determined for case I). A statistical analysis over the four geographical regions considered revealed that smoke originated from different regions. The smoke detected in

**Fig. 1.**

**Supplement:**

**Biomass burning events measured by lidars in EARLINET. Part II. Results and discussions.**

Mariana Adam1, Doina Nicolae1, Livio Belegante1, Iwona S. Stachlewska2, Lucja Janicka2, Dominika Szczepanik2, Maria Mylonaki3, Christiana Anna Papanikolaou3, Nikos Siomos4, Kalliopi Artemis

- 5 Voudouri4, Luca Alados-Arboledas5, Juan Antonio Bravo-Aranda5, Arnoud Apituley6, Nikolaos Papagiannopoulos7, Lucia Mona7, Ina Mattis8, Anatoli Chaikovsky9, Michaël Sicard10,11, Constantino Muñoz-Porcar10, Aleksander Pietruczuk12, Daniele Bortoli13, Holger Baars14, Ivan Grigorov15, Zahary Peshev15

[revised manuscript text omitted]

---

## Referee Comment (RC3) · Anonymous Referee #3 · 7 Oct 2020

Review of ACP-2020-647, Anonymous

The paper by Adam et al. is part of a 2-part set of papers describing EARLINET observations of biomass burning transport to EARLINET sites in Europe. This second paper specifically covers the attribution of smoke events to longe range transport and regional smoke emissions, and provides an analysis of smoke property changes during transport. The attribution is specifically supported through HYSPLIT trajectory analyses and MODIS fire detections.

The subject matter is highly relevant to ACP and the evaluation of smoke transport dynamics is a significant contribution to the field. Arguably, the vertically resolved information on smoke properties after long range transport is one of the most important contributions by EARLINET as a whole. The quality of the EARLINET data set has

been the subject of a long list of previous publications and as such is beyond reproach. The validity of the analysis and attribution methods using the trajectory and satellite fire detection algorithms is less obviously appropriate as indicated by my specific comments and questions below. The presentation quality in the form of figures and text is not at the required high level for ACP – specific suggestions for improvements are made below.

Overall, I would suggest that the paper is not acceptable for publication in its current form. However, because of the importance of the subject matter, the authors should be encouraged to resubmit or address all general and detailed comments included below.

General comments:

1) There is no section of text that describes how the analysis in this paper relates to the contents of paper 1. The authors seem to frequently assume that the reader must have read part 1 prior to reading part 2. This is always the challenge with multi-part papers. While it is appropriate to leave technical details to the other paper, it is not appropriate to require the reader to read the first paper before this one. Please add a section that describes the connectivity of the two papers and why they were divided the way they were.

2) Regarding the focus of the paper, there really is none. It is a confusing mix of single event discussions, source-receptor links, and particle property evolution during long range transport. The Abstract and Conclusion sections are accordingly confusing regarding the paper's main motivation.

3) Pertaining to the quality of the analysis, I fundamentally question the notion that a fire affects an airmass, just because the back-trajectory is located within a certain horizontal distance (100km) and time (1hr) of a satellite fire detection. It seems necessary to detect whether (i) the trajectory is low enough in the atmosphere to be affected or (ii) the fire injection height is likely to have reached the trajectory altitude. This could be done on the basis of simple stability arguments from a reanalysis model. Without
such an analysis, the attribution of smoke layers to given fires is highly questionable.

4) Except for stations explicitly mentioned in the text, there is no way to translate the station acronyms into geographic locations. Please eliminate all use of station acronyms, and provide a map and table with station information.

5) A long list of acronyms are not defined upon first use or not at all: LR, CRLR, SCC.

6) Quality of figures and figure captions is low for many figures – there are many instances of text on top of text (e.g., Fig 1, Fig S6), figure titles that are difficult to understand (e.g., in FigS2 - "evo:# times=13,# times with layers and optical properties=13. # total layers=16"), legend text is too small and/or blurry to read at 100% magnification; these occurrences are too frequent to list in detail. Some figures appear to run on for several pages (e.g., Fig S3), have lots of white space, no identification/letters for subplots, etc. Other figures have different aspect ratios for different subplots (e.g., Fig S4) because the number of events are different for the different stations, but that gives the reader the impression that the stations with fewer events are more important than others, because the former are taller. As a matter of fact, the list of required improvements to the figures is so long and the text is so difficult to follow, that I am breaking off the review in section 4.2. The authors should go through a general effort of making figures mode legible and titles, legends and captions more understandable, rather than relying on this version of figures that was clearly intended for QA purposes, first and foremost.

7) Predominantly in the Abstract and Introduction, there is a confusing lack of distinction in language between receptor regions and smoke origin regions. This makes it difficult to understand the primary purpose of the paper at hand. No map is provided that shows the four receptor regions and the 14 stations therein. Specifically, understanding the meaning of the sentence in the abstract "The results are analysed by means of intensive parameters in the following directions: I) long range transport of smoke particles from North America (here, we divided the events into 'pure North America' and 'mixed'-North America and local) smoke groups, and II) analysis of smoke particles over four geographical regions (SE Europe, NE Europe, Central Europe and SW Europe)." is very difficult. Given the focus on transport, the word "directions" is highly unfortunate. In fact, considering the entire manuscript, this is a rather poor description of the approach. A simpler description would have been that you analyzed the identified smoke occurrences in four European regions and separated the smoke events into source regions based on trajectory analyses (long-range transport from NA, mixed NA, and local).

8) There is significant room for improvement in the use of English grammar. In particular, the use of definite and indefinite articles and the use of proper verb forms needs to be improved throughout the entire manuscript.

Detailed comments:

1) Page 1, line 36: Define what constitutes an "event" .

2) Abstract contains no statistics of the number of analyzed layers at all stations or in the stations grouped into geographic regions. Please add.

3) Page 2, line 24: I am not familiar with the Keywood study, but based on a google search and my own understanding I can attest that an "inverse effect of BB impact on climate" is not commonly used terminology nor that such an impact is well recognized. Please expand.

4) Page 3, line 22: "The identification of the smoke layers was assessed based on the hypothesis of an existing fire within 100 km and ïĆś 1 h from the time and location of the airmass, respectively". Was the altitude of the back-trajectory near the identified location of fires or an estimated Plume Injection Height not considered in the identification of likely smoke layers? (see general comment above).

5) Page 3, line 26: Davies reference is missing.

6) Page 3: "The mean optical properties within the layers were calculated following a few criteria;..." This statement contains no information. Either add the criteria or

remove.

7) Page 3, line 32: Explain why there are often fewer numbers of parameters retrieved than layers detected in Fig S5.

8) Page 4, line 1: Terra/Aqua do not have four observations per day everywhere. Specify latitude range for which this is true or revise statement to properly reflect frequency of observations relevant to this investigation.

9) Page 4, line 4: provide reference for MODIS fire detection.

10) Page 4, line 18: "same graphics as in Section 3" – this is section 3. Do you mean section 3 of the part 1 paper?

11) Page 4, lines 24-31: I do not understand the purpose of describing an event here that was described in detail in part 1. Without figure, this discussion is difficult to follow. Please describe the relevance for bringing this event up here.

12) Page 5, line 6 and Fig 1: I find panels c and d superfluous – they do not add to my understanding, as this info is contained in the map. The locations of Warsaw and Belsk on the map do not look right – the markers are too close to the border with Belarus.

13) Page 5, line 8: is this supposed to say Figs. 1c, d?

14) Page 5, line 16: "As a first remark...". By virtue of placement in the text, this is not a first remark, but it should be. Please move general description to the front of this section.

15) Page 5 and onward: Section 3.1 is a dense and unacceptably difficult to follow enumeration of facts and statistics that are not linked by any common thought or thread.

16) Page 9, line 10: Figure S7, after significant improvement needs to go into the introduction section and frame the entire paper.

17) Figure 3 caption: explain the meaning of the markers with large error bar symbols.

---

## Author Comment (AC1) · 30 Nov 2020

We would like to thank the reviewers for the thorough revision of the manuscript and for suggestions! We hope we addressed the comments accordingly.

We use across the text (below) the following highlights:

In red, reviewer's comments

In black, authors' answers

In green, text from manuscript submitted

In blue, proposed changes to revised manuscript

In magenta, some citations from different papers.

Before answering to the reviewer's questions/comments etc, we would like to indicate a few changes that we have done to the initial manuscript. Those changes are not related with the data processing and analysing or its scientific content but rather with text cosmetics.

- minor correction to author names or affiliations

- based on suggestions for Part I, we changed NE, SE, SW, CE to North-East, South-East, South-West, Central throughout the text (except figures and tables)

-in Funding we added:

The research was partially funded by the European Regional Development Fund through the Competitiveness 613 Operational Programme 2014-2020, POC-A.1-A.1.1.1- F- 2015, project Research Centre for Environment and Earth 614 Observation CEO-Terra, SMIS code 108109, contract No. 152/2016. Juan Antonio Bravo-Aranda received funding from the Marie Sklodowska-Curie Action Cofund 2016 EU project – Athenea3i under grant agreement no. 754446

-Figures

The figures in the main manuscript and most of the figures in Supplement (except former Figs. S3, S5, S6), were replaced with better quality ones (as suggested). Figures in Supplement were renamed after introducing Fig. S1 with stations location. The two panels in Figure S2 (former S1) were glued in a single panel.

-date/time

The dates were changed from e.g. 20150710 or 10th of July 2015 to 10 July 2015 (as replaced during proofreading in Part I).

**Anonymous Referee #1**

This paper presents observations of biomass burning observations, specifically aerosol intensive parameters, from a regional lidar network, EARLINET. The paper then discusses their origins as determined by back trajectory analysis. This paper is a companion to an earlier paper (acp-2020-320) which described the methodology for QC on this network of instruments of disparate data quality. This is certainly an important topic; however, I found the paper to be difficult to follow and review, first because the paper seems to have several different opinions about what its topic is, and second because the limited data available makes broader interpretation challenging. I would recommend the authors consider what exactly they want to convey from this paper, and make sure that the paper accomplishes this.

I have some further concerns about the analysis presented in the paper, especially much of Section 4 (4.2 and 4.3), again largely due to the limited data available for analysis. Of course this doesn't mean analysis can't be performed, or that the results are meaningless, but only that they must be more carefully considered, and I would like to see much more to that effect (maybe some of this is in there; if so, I couldn't easily follow it). NB: I have not reviewed the companion paper, but I have skimmed it to try to see whether I've missed some information which would address these concerns, but I wasn't able to find much that was relevant. The topic presented here is interesting, but the paper as written does not address it in a satisfactory manner. At the very least I think major revisions are in order before it can be considered for publication.

Specific comments, structure and focus:

–The title and the abstract seem to be describing two different papers. The title talks about biomass burning events as a whole, but the abstract focuses heavily on BB due to long-range transport, and specifically LRT from North America. Why this framing? And then according to the first sentence of Section 3.2 (and the first page of 4.1), N America isn't even half of the

LRT cases? Although this sentence says "We encountered 168 measurements over the 24 LRT periods... from these measurements, 77 have a North American origin and 91 have different BB origin (local)." Are the 168 measurements then all long-range, or also local? In other parts of the paper it seems that this "local" transport is also included under the "LRT" category. I would have thought these would be two distinct classifications. There's also a discussion of transport from Asian and African sources, so why is the abstract solely focused on N Am sources?

Thank you, indeed, the abstract was not explicitly written – we apology for our poor English and lack of clarity. Below is the rewritten abstract, written in a more generic way. As for the first sentence of Section 3.2 (and the first page of 4.1): you are right, the North America LRT isn't even half of the total cases. We apologize for not being clear about LRT events. We only analysed LRT events from N America. We corrected the title for section 3.

3 Long range transport biomass burning events originating in North America

We revised the sentence in question as follows:

Initial:

We encountered 168 measurements over the 24 LRT periods, as follows: one period in 2009 and 2011, two periods in 2012, 2014, three periods in 2015, and five periods in 2013, 2016 and 2017. From these measurements, 77 have a North American origin and 91 have a different BB origin (local).

Changed:

There were in total 168 measurements over the 24 periods identified as related to LRT of smoke from North America, i.e. one period in 2009 and 2011, two periods in 2012 and 2014, three periods in 2015, and five periods in 2013, 2016 and 2017. From those measurements, 77 are strictly originated from North America while 91 have different origin (e.g. fires in Eastern Europe, Iberian Peninsula, North Africa).

From the statistics mentioned in 4.1. we see that overall, we quantify 4699 fires in N America and 13635 elsewhere. The total number of smoke layers analysed in these papers is 795. From those, 77 smoke layers are originating from N America. Moreover, 27 out of those 77 layers record a mixed smoke (N American + local). The above facts, (77/168 measurements have origin in N America) draws our attention to the fact that when we perform measurements over forecasted LRT smoke we may measure as well local smoke or smoke from other directions than the forecasted ones. Thus, our quantification over the contributing fires brings a new perspective over data analysis. At this stage we focused only on LRT from N America simply because it was easier to delimitate. We did not intend at this stage to quantify LRT from Africa or Asia in the same manner as for N America. This analysis implies more criteria for selection of the LRT as compared with N America. Hopefully, in the future we will tackle LRT from all regions and analyse the potential difference but it is out of the scope of this work. Indeed, quantifying LRT from Africa or Asia can be a stand-alone paper. In the present study we analyse the BB events originating in Africa and Asia through the statistical analysis in section 4.

Initial abstract:

Biomass burning events are analysed using the European Aerosol Research Lidar Network database for atmospheric profiling of aerosols by lidars. Atmospheric profiles containing forest fires layers were identified in data collected by fourteen stations during 2008–2017. The data ranged from complete data sets (particle backscatter coefficient, extinction coefficient and linear depolarization ratio) to single profiles (particle backscatter coefficient). The data analysis methodology was described in Part I (Biomass burning events measured by lidars in EARLINET. Part I. Data analysis methodology, under discussions to ACP, the EARLINET special issue). The results are analysed by means of intensive parameters in the following directions: I) long range transport of smoke particles from North America (here, we divided the events into 'pure North America' and 'mixed'-North America and local) smoke groups, and II) analysis of smoke particles over four geographical regions (SE Europe, NE Europe, Central Europe and SW Europe). 24 events were determined for case I). A statistical analysis over the four geographical regions considered revealed that smoke originated from different regions. The smoke detected in the Central Europe region (Cabauw, Leipzig, and Hohenpeißenberg) was mostly brought over from North America (87 % of the fires), by long range transport. The smoke in the South West region (Barcelona, Evora, and Granada) came mostly from the Iberian Peninsula and North Africa, the long-range transport from North America accounting for only 9 % here. The smoke in the North Europe region (Belsk, Minsk, and Warsaw) originated mostly in East Europe (Ukraine and Russia), and had a 31 % contribution from smoke by long-range transport from North America. For the South East region (Athens, Bucharest, Potenza, Sofia, Thessaloniki) the origin of the smoke was mostly located in SE Europe (only 3 % from North America). Specific features for the lidar-derived intensive parameters based on smoke continental origin were determined for each region. Based on the whole dataset, the following signatures were observed: i) the colour ratio of the lidar ratio and the backscatter Ångström exponent increase with travel time, while the extinction Ångström exponent and the colour ratio of the particle depolarization ratio

decrease; ii) an increase of the colour ratio of the particle depolarization ratio corresponds to both a decrease of the colour ratio of the lidar ratios and an increase of the extinction Ångström exponent; iii) the measured smoke originating from all continental regions is characterized in average as aged smoke, except for a few cases; iv) in general, the local smoke shows a smaller lidar ratio while the long range transported smoke shows a higher lidar ratio; and v) the depolarization is smaller for long range transported smoke. A complete characterization of the smoke particles type (either fresh or aged) is presented for each of the four geographical regions versus different continental source regions.

Changed abstract:

Biomass burning episodes measured at 14 stations of the European Aerosol Research Lidar Network (EARLINET) over 2008-2017 were analysed using the methodology described in "Biomass burning events measured by lidars in EARLINET. Part I. Data analysis methodology" (ACP, this issue). Fire smoke layers were identified in lidar profiles of particle backscatter and extinction coefficients and particle linear depolarization ratio. A number of 795 layers for which we identified at least one intensive parameter were analysed. The layers were distributed as following: 399 layers for South East Europe, 119 layers for South West Europe, 243 layers for North East Europe and 34 layers for Central Europe. The intensive parameters obtained within those layers were analysed comparatively and interpreted following three research directions: (I) case study of long range transport from North America recorded by three stations, (II) analyses of all long range transport of smoke particles from North America to assess the mean properties in two classes: pure North America smoke (50 cases) and mixed with local smoke (27 cases), and (III) statistical analysis of the smoke properties (fresh versus aged) over smoke occurrences in four European regions (Central, North-East, South-West and South-East Europe) separating the smoke events into continental source regions (European, North American, African, Asian or a mixture of two), based on trajectory analysis. The smoke detected in the Central Europe (stations: Cabauw, Leipzig, and Hohenpeißenberg) was mostly brought form North America (87 % of fires). In the North-East Europe (Belsk, Minsk, Warsaw) smoke advected mostly from Eastern Europe (Ukraine and Russia) but there was a significant contribution (31 %) of smoke from North America. In the South-West Europe (Barcelona, Evora, Granada) smoke originated mainly form Iberian Peninsula and North Africa (while 9 % were originating in North America). In the South-East Europe (Athens, Bucharest, Potenza, Sofia, Thessaloniki) the origin of the smoke was mostly local (while 3 % represented smoke from North America). The following features, correlated with the increased smoke travel time (corresponding to aging) were found: the colour ratio of the lidar ratio and the colour ratio of the backscatter Ångström exponent increase, while the extinction Ångström exponent and the colour ratio of the particle depolarization ratio decrease. The smoke originating from all continental regions can be characterized on average as aged smoke, with a very few exceptions. In general, the long range transported smoke shows higher lidar ratio and lower depolarization ratio compared to the local smoke.

–The abstract also doesn't make any mention of the case study [edit: studies] which is discussed in Section 3.1. Why not? Isn't this a big part of this paper?

Yes, the reviewer is right. The event of 8-10 July 2013 (subsection 3.1) is exceptional due to its temporal duration. We included in the revised Abstract.

– I found it very difficult to follow all the multiple sets of acronyms used (site names, groups of sites, plume origins, parameter names…). A list of the acronyms is provided, but it's in the last page of a separate supplement file, rather than in an appendix or even in the main text. This really inhibited my ability to understand the paper. I would suggest at the very least moving the acronym list to an appendix within the paper, if not to the main text itself. I think the authors rely on these abbreviations enough that it would belong in the main text.

We are sorry for this. We agree with the reviewer. Thus, as suggested, we moved the full list of acronyms and symbols to a dedicated Appendix in the main manuscript (after Conclusions). Moreover, we provide explanations and give the full name to every abbreviation and symbol, first time introduced in the revised paper. As mentioned by other comments, we added in the list the acronym "SCC" (not explained previously) and deleted UTLS (not used here).

– Does Figure 3 use one color to indicate two completely different things (e.g. am I reading this right that WAW and AS are both marigold, but the marigold asterisks are not what's used to determine the marigold circle-whisker?)? This is unnecessarily confusing. Pick different colors or consider e.g. using shape distinctions for one or the other. (this comment refers to the final pattern of Fig 3… I guess the previous page is also Fig 3).

Yes, we used the same colour for two different things. However, the markers are different. The asterisks are related with the stations (e.g. *waw represents data from Warsaw station) while the circles and error bars are related with the smoke source (e.g. oAS means smoke originating from Asia). For consistency with Part I we preferred to keep the same coding. However, a better explanation is given in the figure caption as:

Figure 3. Scatter plots between various two intensive parameters for North-East (NE) region. The colour code of the asterisks is station related (as labelled in the title). The colour code for the mean values (shown by circles) and their STD values (shown by error bars) are related with the smoke source origin (stated as text on the plots). Thus, we have the following source origins: Europe (EU - blue), Asia (AS - yellow), North America (NA – magenta), Europe – Africa (EUAF – green), Europe – Asia (EUAS – cyan) and Europe – North America (EUNA – dark red).

– On that subject, a lot of the figures are simply too large, and the captions are insufficient to describe them. There are also a whole lot of supplementary figures too. Consider splitting them into smaller sets of figures, or reassess whether the information really needs to be presented this way. Figures that span 3 pages with a single (small) caption are very confusing to follow. Especially e.g. cases such as Figure 4: you conclude there is no correlation between the variables presented in panels d-f. Maybe this isn't necessary to show at all, then?

We thank the reviewer for this appreciation. We agree with the reviewer that there is always a compromise in the information to show to the reader (neither much nor little). In this particular case, we think that this kind of information is useful as readers may like to analyse the features (even not correlated) and eventually compare with their findings. Thus, we follow the advice of the reviewer by improving the figures captions as follows:

Fig 2. We added: There are 27 measurements of mixed smoke.

Figure 2. All of the 77 measurements recorded during LRT from North America. Measurements are divided into North America origin ('pure NA') and 'mixed' (North America and local) origin. There are 27 measurements of mixed smoke. Along with the intensive parameters, the layers altitude and thickness (marked as error bar) are shown on the upper plot. Mean, minimum and maximum values from literature are shown in red. 'pure NA' stands for 'pure North America'. Symbols (* and o) are shown in panels title.

Fig. 3: All the panels were shrunk and grouped on the same page. Caption (changed, as described above) was moved on the same page in a text box.

Fig. 4: both figures are brought on the same page. See changes below (in blue).

Figure 4. Scatter plots between CRBAE and CRPDR (a), EAE and CRPDR (b), CRLR and CRPDR (c), EAE and CRLR (d), CRBAE and CRLR (e), CRBAE and EAE (f). The mean values found in literature are added (see red and blue circles). a)-c) plots are obtained only for North-East (NE) region where two PDR are available (Warsaw). The regression equation and coefficient of correlation are shown for each plot.

Fig. 5: the six figures were grouped two by two.

Figure 5. Intensive parameters and corresponding colour ratio (CR) versus smoke source origin based on Fig. 4. Plots a)–f) correspond to plots a)–f) in Fig. 4.

We reprocessed the figures (with better quality) and replaced the initial ones.

Regarding Fig. 4, yes, we concluded that there is no correlation for panels e and f.

Regarding the number of plots in the Supplement, we want to provide a clear view over the data availability for each station (Sect. 3 and Sect. 5) and their range of variation (Sect. 4 and Sect. 6) for those readers that may be interested in the particularities of individual stations. Since this information is not essential, we locate these figure in the Supplement where only readers interested in details will access.

Specific comments, scientific analysis:

– perhaps this was addressed in Part I, although I found no mention of it: it seems this classification is entirely determined based on one run of the HYSPLIT trajectory model run over ten days. What have you done to assess whether these trajectories are robust? Did you run it in ensemble mode to verify whether the trajectories were consistent? Or did you run the model forward from a fire location to see if the airmasses ended up in roughly the same location? Perhaps the authors have done this, but I would like to see much more information regarding how they addressed the uncertainties inherent in HYSPLIT or any trajectory model. Especially since many of the conclusions presented here are based on very few (sometimes just one) points, I think this is a necessary exercise.

As the reviewer indicates, the classification is based on one run of the 10-days HYSPLIT trajectory model runs. The trajectories were not assessed for their robustness and consistence by runs of ensemble mode (either backward or forward). Therefore, the uncertainties inherent in HYSPLIT trajectory model were not addressed explicitly. The main reason was the lack of human and technical resources to manage the huge quantity of the data to be analysed (we run Hysplit for 1901 layers).

In part I we address it (see below) and we add the following sentence in section 2:

As mentioned in Part I, we did not consider the uncertainties related with Hysplit backtrajectories or FIRMS database. In future related studies, we will consider the estimation of the injection height and thus asses if the smoke reaches the air mass altitude.

Text from Part I:

We performed 1036 Hysplit runs for 1901 layers corresponding to 960 time stamps…

…It is worth mentioning that the Hysplit model does not provide the uncertainty. In order to get a possible uncertainty of an individual trajectory, a trajectory ensemble is suggested (Rolph et al., 2017). We may assume that high uncertainties in the air-mass location may occur particularly over long periods of time (e.g. 10 d), which in conjunction with a fire's location may mean a missed fire or a fire detection that was not contributing to the measurement. Drexler (https://www.arl.noaa.gov/hysplit/hysplit-frequently-asked-questions-faqs/faq-hg11/, last access: 26 November 2019) mentions that the uncertainty is between 15% and 30 %. On the other hand, FIRMS may miss some fires (especially in a cloudy atmosphere). According to Giglio et al. (2016), the collection 6 MODIS has a smaller commission error (false alarm) as compared with Collection 5 (1.2% versus 2.4% respectively). The probability of fire detection (regionally) increased by 3% in boreal North America while staying almost the same in regions such as Europe or northern Africa. We have been using fires with a confidence level larger than 70 %. We did not investigate the injection height based on FRP in order to estimate whether the smoke of a particular fire indeed reached the altitude of the back trajectory. We would like to emphasize that, due to the satellite's polar orbit, the same geographical location can be seen four times a day at the Equator and more times as the latitude increases (due to orbit overlap). Thus, we may miss a certain number of fires (which burn less than a few hours, between the two orbits). However, we may consider those short-lived fires to be insignificant in smoke production. The FIRMS database was used in several studies to identify the BB origin. However, all fires occurring over certain periods (for which the back trajectories were calculated) are typically accounted for. Thus, fires occurring over the whole day (e.g. Nicolae et al., 2013; Stachlewska et al., 2018; Janicka et al., 2017) or several days (e.g. Mylonaki et al., 2017; Heese andWiegner, 2008; Tesche et al., 2011) were reported. By contrast, our novel approach accounts only for those fires which were occurring around a back trajectory (100 km radius) at the time of air-mass passage (±1 h).

In conclusions of Part I we mentioned:

The possible sources of uncertainty during such an analysis may be the following. A small change in the input to Hysplit may give a different output (e.g. use of altitude a.g.l. versus a.s.l., use of GDAS0.5 versus GDAS1). See for example Su et al. (2015). Various algorithms employed to estimate the layer geometry may give slightly different values over the mean values in the layers. Thus, the direct comparison with other reports over the same event should be carefully performed. **Uncertainties in Hysplit back trajectories as well as in the FIRMS database are not considered**. Last but not least, the imperfect data quality control (including the present methodology) may contribute as well….

…Future investigations envision several important features to be accounted for. A more detailed analysis of grouping the source locations using cluster analysis is envisioned, where a larger number of clusters should be chosen in order to pack more homogeneous regions with a similar vegetation type. Thus, a more accurate correlation between the source type and the measurements is envisioned. The time travel should be considered in some way. FRP will be considered to estimate the injection height and thus have more confidence that the smoke will reach the back-trajectory altitude. The biggest challenge remains the quantification of the contribution of different fires.

Related, p.6 Line 11: "Moreover, the ensemble backtrajectory for 3000 m a.g.l. at 19:00 using GDAS 0.5 showed more backtrajectories towards North America." More appropriate would probably be something like "the ensemble back trajectory corroborated that the air mass originated in N America," if that's what the ensemble trajectory showed. Consistent ensemble paths back to NA would give extra confidence to the result that this air mass originated there, but doesn't necessarily mean that more air came from there. Conversely, if the ensemble shows e.g., half from NA and half from Asia, that would indicate the HYSPLIT trajectory is not particularly robust in this case and we may need to reassess that confidence. I don't see much other text indicating whether these types of tests have been performed.

We agree with the reviewer on that we performed just a few ensemble backtrajectories in comparison with other studies but we emphasize in the manuscript that most of the trajectories go to N America (not all). As aforementioned, the full ensemble study was unapproachable due to the lack of human and technical resources. We rewrote in the manuscript as follows and will consider this comment for future studies:

Moreover, the ensemble backtrajectory calculated at the site location for 3000 m a.g.l. at 19:00 using GDAS 0.5 (not shown for brevity) corroborated that most of the air mass originated in North America. Note that, consistent ensemble paths back to certain area-region give extra confidence to the result that this air mass originated there, but it does not necessarily mean that more air masses came from there. Conversely, if the ensemble indicates e.g., two or more sources of origin separated from each other, the trajectory is not robust and one needs to reassess its confidence.

I do notice that p.6L14-15 seems to say that "the results are not in agreement" [for two different configurations of HYSPLIT trajectories]? Is that to what this sentence is referring? Then how can you believe them? Maybe I'm misunderstanding this section…

The reviewer refers to a part of the manuscript where authors indicates that the results from GDAS1 and GDAS0.5 presents slightly differences. From my point of view, a "slightly different" should not be understood as 'not in agreement' but as a possible source of uncertainty as we mentioned in Conclusions of Part I. Besides, we think that the comparison is still valid. I guess an optimal solution is to apply one of the options for processing.

– I don't follow the logic behind Section 4.3. Examining means can be a decent way to analyze two different populations, but in this case (with very small sample sizes and large ranges between them; e.g. the error/range points/whiskers presented in Figs 3, 4, 5) I'm not convinced studying just the means is meaningful. For that to be the case, you'd want to have populations where the properties were known to be 1) normally distributed and 2) fairly uniform, but as the authors themselves state on p. 17 (Lines 14-17), these episodes are a mixture of several fires already. So this doesn't seem a good approach to me. Perhaps medians/percentiles, or simply present the ranges (rather than stdev) especially when the points are so few?

Plus, if you're trying to draw conclusions based on 2 points of one case and 1 of another, this doesn't seem statistically robust or generalizable. That's not to say that minimal data can't be useful, but its limitations need to be acknowledged. I think the data presented in Figure 5 are actually quite useful to that end, but again, I found the presentation quite confusing (log-y axis on one side, lin-y axis on the other, with multiple colors and a not very informative caption). Perhaps focusing on additional case studies would be a clearer presentation for the cases outside EU-origin.

In 4.3 we stated:

As a general statement, we consider that the cases where we have only one or two measurements are not statistically significant, a good confidence being given by at least five measurements available. Therefore, the results discussed below should be regarded with care in such situations.

We rephrase as:

As a general statement, we consider that the cases where we have only one or two measurements are not statistically significant, a good confidence being given when at least five measurements are available. Therefore, the results discussed below should be carefully treated  in such situations.

We stated in Conclusions that in most of the cases we measure mixed smoke. There were three ways we determined the mixture. 1) In the examples for common fire measured by two stations, discussed in Part I., Sect 5.2.1, Fig. 7, we quantified the layers which strictly measured smoke from the common fire and those layers which also had contributing smoke from other fires (the number of fires were counted). 2) in the examples for LRT from N America, (Fig. 1 here as well as in Part I, Fig. 9), we determined the smoke layers which were originating strictly in N America and the mixed layers which had also a contribution from local fires. The number of N American and local fires were counted. 3) In the statistics performed in 4.3, the quantification was performed based on continental location of the fires. Thus, a EUNA (Europe-N America) classification means a mixed smoke with contributing fires from both N America and Europe. We think that the classification based on continental source origin is a good start for such approach. However, as seen in Fig. 5, we obtained statistically significant samples only for Europe source region and SE and NE receptor regions. Moreover, CRPDR is available only for NE region (Warsaw station).

Regarding Fig. 5, we chose the log scale on the right y-axis because the number of cases is quite different and using the log scale is more readable. We wanted to point out the available number of cases for each of the scatter plots in Fig. 4. We agree with the reviewer that statistics is significant when we have a large number of events. Unfortunately, this was not the case so we indicate in the manuscript that the discussion should be regarded with care and we avoid statistical tools such as normal distribution or the percentiles. We propose to leave the analysis as presented, for consistency with Part I. Your fruitful suggestions will be taken into account in future studies.

– in Figure 4 you present values of R=1. This is because you are fitting through only 3 (2?) points. This R is meaningless in this context, especially since one has very large uncertainty bars and the other has only one point.

You are right, R is based on three mean values (panels b-c). Moreover, the mean over EUNA is based on 1 event while the mean for EUAS is based on 2 events (as seen in Fig. 5 b-c). Only EU region is statistically significant. We will add the following statement in the text of 4.3.1.:

Note that even the high correlation (R=1) for the regressions shown in panels b-c, it is not statistically significant taking into account that the regression is based on three mean values. Moreover, two of the mean values were based on one and two values respectively (see Fig. 5b-c). However, we believe that future similar studies will strengthen this correlation.

– Section 4.2: the authors draw conclusions that e.g., EAE values of 1.4 indicates a mixture of fresh and aged smoke, and 1.2 is only aged smoke. With such a small difference between the two, I'm not convinced this is robust; e.g., even the STD values in Table 3 are uniformly greater than this. What are the ranges?

We appreciate this reviewer's comment. We revised the sentence and gave the ranges of EAE values instead of the mean. Note that these are mean and STD values based on all available values and thus they may not correspond to figs. 3 and 4-5 or table 4. The sentences referring to CRLR and EAE are revised as following. Note it was an error on EAE mean value (1.2. instead of 1.1).

A mean value of $0.95 \pm 0.2$ ($1.2 \pm 0.1$) is observed for $CR_{LR}$ for EU (EUAS) source region, corresponding to fresh (aged smoke). In average, the values of EAE for EU and EUAS regions are $1.4 \pm 0.4$ and $1.1 \pm 0.1$ respectively, suggesting a mixture of fresh and aged smoke for the EU source region and aged smoke for the EUAS source region.

Further, (and assuming the above issue can be resolved) in Table 4 all except for three categories are classified as "aged," with two more "fresh/aged" and one "fresh." This is one of the conclusions in the abstract as well, but I'm not seeing any definition of what "aged" means in this case. And if this is truly the case, the classification of "aged" corresponding to different regions and origins is not very useful at all. Aerosol aging especially of BB can mean different things on different timescales (e.g. Haywood et al. https://doi.org/10.1029/2002JD002226 defined "fresh" as only a few minutes after emission), and a few days old likely won't be the same as a week old, so what exactly is meant by this? It may be more instructive to only focus on the cases which are *not* aged and examine what distinguishes them from the other cases. Or, conversely, if you're using back trajectories, can these determine exactly how "aged" (= time from emission) each population is? That might actually be more instructive than classifying observations as "aged" based on (as I understand it based on p.5L26) the EAE and CR, and could potentially allow for discussion of the property of BB different ages, rather than the regional Europe division.

We thank the reviewer the proposed methodology to classify the BB types. However, we followed Nicolae et al. (2013) methodology based on CRLR and EAE to differentiate fresh and aged BB (Table 4 shows 2 categories with fresh smoke and 2 categories with mixed fresh/aged). Indeed, there are other ways to classify the smoke in fresh and aged, based on different approaches. We used the one based on optical properties (intensive parameters and/or colour ratio) and partially the travel time. We were not able to use the microphysical properties (e.g. Haywood, reference provided by the reviewer) as there were very few datasets with 3+2+1 retrievals. We mentioned in Conclusions about further investigations which should consider the travel time as well. In Part I it was mentioned:

EAE decreases with time (distance) (e.g. Müller et al., 2005, 2007, 2016).

Lidar ratio: it was shown that for aged BB particles (big travel time), LR@532 > LR@355 (e.g. Wandinger et al.,

2002, Murayama et al., 2004; Müller et al., 2005; Sugimoto et al., 2010, Nicolae et al., 2013), hence the colour ratio (CR) for LR, i.e. CRLR > 1 for aged smoke (LR@532 increases more with aging smoke).

As aforementioned by the reviewer, a lot of information is already provided in the manuscript (also in the Supplement) and thus, we think we should not include further investigations here for the sake of clarity. Further studies may consider the travel time as well as other criteria and smaller dataset (e.g. SE Europe receptor). A classification of smoke based on travel time was shown by Janicka and Stachlewska (2019) who analyses 116 sub-layers over 2 days of measurements in Warsaw.

– In the abstract, findings i) and ii) are saying basically the same thing, but inverted.

We removed ii).

And (related to the previous comment), "travel time" might more appropriately be called "aerosol age" (="time from emission"). But, I don't really see any discussion of this beyond the case studies in Section 3 (and maybe a statement p.17L1-2, that it was aged based on high RH… I don't know that this is always the case, e.g. African biomass burning in particular can see high humidity very near emission).

We agree with the reviewer that the smoke travel time is actually the smoke age. However, for consistency with Part I, we will prefer to keep the term 'travel time'. Regarding RH, apart from that specific case, we did not analyse RH along back trajectories.

This study has allowed us to discover other future ways to study the BB but, unfortunately, the influence of the RH was out of the scope of this study. Hopefully we will address other aspects in future studies.

And regarding the conclusion in the abstract and in Section 4.3.1, "A slight decrease of the CRPDR with travel time was observed, while the CRBAE maintained similar values for all the source regions," I'm not clear whether this can be concluded from the data as presented. Was the same plume observed from multiple stations along its trajectory? Otherwise it's hard to say whether the initial aerosol properties were the same, or whether they were different to begin with. I'd like to see more clarification as to how this conclusion was reached.

No, there were not the same plumes. Indeed, we cannot conclude about their initial properties. In figs. 4 and 5 we basically show the averages over all events measured in different regions based on different continental sources. In Fig. 4a) we see a larger variation for CRPDR while CRBAE varies less. In Fig. 5 a, we see a decrease for CRPDR from European source to Asian source and then N American source. On the other hand, CRBAE for European source is similar with N American source and larger for Asian source.

As mentioned in Part I, there were only five cases where we had the same BB as source while the plume was measured by two stations. Unfortunately, only for two events we could compare the same intensive parameter (BAE). Thus, for this type of study, more data are necessary in the Earlinet database.

–p.6,L19-20: "IPs STD represents _ 15, 38, 27, 17 and 37 % of the mean, respectively, and thus we claim a relatively small variability over the whole three days of measurement." I don't follow this. First it will depend what the mean value and what dynamic range is typically expected from a particular parameter, and second, with only 8-13 measurements for several of these parameters, I'm not sure stdev is the best metric for variability. What's the range, or maybe (for the 31-case BAE) percentiles?

We agree that the number of cases is not sufficient and thus the statistics might not be very relevant. We are sorry we did not consider the percentiles from the beginning. We added the range for each variable:

In our study, the mean and standard deviation (STD) for all IPs (except LR@532 = 91 ± 3 sr and EAE = 0.3 ± 0.1, where we have only single values) are as following: LR@355 = 38 ± 6 sr (average over based on nine cases over the range [30, 46]), BAE@355/532 = 1.4 ± 0.5 (average over based on 13 cases over the range [0.5, 1.9]), BAE@532/1064 = 1.1 ± 0.3 (average over based on 31 cases over the range [0.3 2]), PDR@355 = 2.5 ± 0.4 % (average over based on nine cases over the range [1.7, 3.1]), PDR@532 = 2.4 ± 0.9 % (average over based on eight cases over the range [1.1, 3.9]).

The 95[th] percentile for BAE@532/1064 is 1.49 while the 5[th] percentile is 0.575.

In Part I we wrote:

The imposed limits (data filtering) for IPs are the following: LR@355=[20, 150] sr, LR@532=[20, 150] sr, EAE=[-1, 3], BAE@355/532=[-1, 3], BAE@532/1064=[-1, 3], PDR@355=[0, 0.3] and PDR@532=[0, 0.3] (following closely Burton et al., 2012; Nicolae et al., 2018).

–p.8,L10+: As above, what is the confidence on the back trajectories from NAm vs elsewhere? Further, I think this paragraph ("we noted several IP values… outside the range reported") refers to both NA and NA+local mixed aerosol? What fraction of each is included under "mixed" conditions? Is it relatively constant, or does the percentage vary for different cases? Was there a threshold (e.g., only 5% local) below which EUNA->just NA? I didn't see this in the paper… and while certainly it's possible to report values outside what's been previously reported, it seems prudent to discuss how much mixing and how much confidence goes into the present estimates, where they disagree. (Relatedly, what are the asterisk vs circle for the literature numbers in the Fig 2 panels? I don't see this described anywhere.)

In Part I, we wrote:

We may assume that high uncertainties in the air-mass location may occur particularly over long periods of time (e.g. 10 d), which in conjunction with a fire's location may mean a missed fire or a fire detection that was not contributing to the measurement. Drexler (https://www.arl.noaa.gov/hysplit/hysplit-frequently-asked-questions-faqs/faq-hg11/, last access: 26 November 2019) mentions that the uncertainty is between 15% and 30 %.

As seen in Fig. 2, for BAE@355/532 there are 8 cases where the values are larger than the maximum reported (with 6 cases for mixed smoke) and one case where the value is lower than the minimum reported (pure N America smoke). For

BAE@532/1064 there are 6 cases where the values are lower than the minimum reported (with 2 cases for mixed smoke). There is one vent for which both BAE are smaller than the minima reported.

At this stage we just quantified the number of fires (N America and local) which contributed to each smoke measurement. We considered the smoke originating from N America (NA) when all the fires were detected there (0% local). We did not go into detail to assess how large was the contribution of N America or local (besides the number). For this, FRP should be taken into account. We only performed this analysis for a few cases. The fraction of local versus N America varies from one measurement to another.

The asterisks and the circles are shown in the title of each plot. Thus, we have LR and PDR @355 shown by * and LR and PDR @532 by o. BAE@355/532 is shown by * and BAE@532/1064 is shown by o. Figure 2 caption was improved, as mentioned above.

We added the following sentence related to the large value for EAE:

The large value for the mixed case EAE may be due to the contribution of the local, fresh smoke.  At a closer look, the large 'pure N America' EAE value, recorded on 4 July 2013 in Thessaloniki in a layer at ~ 3.6 km altitude, correspond to air masses reaching ~ 9 – 11 km over the fires in North America. It is possible that the fires did not reach that altitude and thus the measurements for that layer may come from other sources.

–conclusions, p. 16,L23-25: with the limited amount of data available, I think this is too strong of a statement.

Thank you, we rewrote as:

Based on the dataset analysed over 2008-2017, the smoke detected in the Central Europe (stations: Cabauw, Leipzig, and Hohenpeißenberg) was mostly brought form North America (87 % of fires). In the North-East Europe (Belsk, Minsk, Warsaw) smoke advected mostly from Eastern Europe (Ukraine and Russia) but there was a significant contribution (31 %) of smoke from North America. In the South-West Europe (Barcelona, Evora, Granada) smoke originated mainly over Iberian Peninsula and North Africa (while 9 % were originating in North America). In the South-East Europe (Athens, Bucharest, Potenza, Sofia, Thessaloniki) the origin of the smoke was mostly local (while 3 % represented smoke from North America).

–another couple points of clarification: when the authors say "fire" is identified by being within 100km, +/-1h from the trajectory (p.3L23), were altitude thresholds applied or not; i.e., if HYSPLIT trajectories were at 8km, was it still considered to be a smoke source even for surface-confined, non-pyrocb fires?

We did not apply at this stage an altitude threshold (as shown in part I). This will be taken into account in additional studies where several ideas discussed here will be implemented, starting with a smaller dataset (for one station or one region). Please see the answer on the same topic as for RC3, general comment 3).

Also, are the "total fires" on p.5L_18 distinct fires, or just MODIS fire detections at a given time? (may be semantics, but surely a single fire will often be detected multiple times during multiple overpasses; are these considered distinct fires in these summaries?

Thank you, this was clarified in the text. In this study, the fires denote MODIS fire detections at a given time. We mentioned that a fire can be detected more than once over a certain period. In Part I we wrote:

Please note that as a function of the air trajectory, one fire can be seen more than once (e.g. during a cyclone or anticyclone) or due to slow air motion (spatial–temporal stationarity over the 100 km area and 1 h).

Over the 7879211 fires provided by FIRMS over the canvas chosen (including N America, Europe and parts of Africa and Asia) during the time period 2008-2017, there were 10 fires which were provided twice (i.e. by the same time, longitude and latitude) and not used in our analysis. Indeed, a fire (denoted by longitude and latitude) can be seen more than once by MODIS as it depends on its burning time (sometimes a fire can burn few days). When we mention for example 961 fires, detected 1664 times that means that the 961 fires have different locations (in terms of latitude and longitude) while 1664 detections means that some of the 961 fires were seen more than once (at different times). We did not intend to extract the number of distinct fire spots. We consider it as a difficult task to be performed.

We add the following text in 3.1:

As mentioned in Adam et al. (2020), function of the air trajectory, one fire can be seen more than once (e.g. during a cyclone or anticyclone) or due to slow air motion (spatial–temporal stationarity over the 100 km area and 1 h).

However, only some of them contain smoke originating from North America (according to our criteria) as marked by squares on panel e) and further shown  on panels j–n (the mixed cases are marked by diamonds). Thus, from 21/8/25 layers measured at Belsk/Cabauw/Warsaw, only 12/8/11 layers were identified as having smoke origin in North America. Further, 2 and 4 smoke layers were identified as mixed (North America and local) for Belsk and Warsaw respectively.

Other comments:

– Q: What is red in Table 2? A: local fire; this is buried in the text, add it to the caption.

Yes, with red are the local fires. We added the following to the caption:

The local fires contributing to the mixed smoke are shown in red.

– p.4L18: "same graphics as in Section 3" but this is Section 3?

We are sorry for this mistake (due to re-organizing) pointed out by the reviewer. We deleted the sentence. Additionally, we change the section title to:

3 Long range transport biomass burning events originating in North America

– p.6,L21: "better sphericity" => "greater sphericity"?

Corrected as: "greater sphericity".

– p.6,L23: "increase in LR" relative to what? Not clear.

Corrected as: "relative increase in LR".

– p.6,L27: If this event occurred on 14July, why is it within a section labeled 0708-0710? Expand the 3.1 title or make a new subsection. Same for the following paragraph, now you're talking about the 2017-2018 event in this same section?

We added the subsection 3.2 (while 3.2 becomes 3.3.) as:

3.2. Other long range transport events

–p.7L11: suggest move the Peterson reference up to L4.

Suggestion accepted. We  also added the reference in References (we forgot to add it).

Peterson, D. A., Campbell, J. R., Hyer, E. J., Fromm, M. D., Kablick, G. P., Cossuth, J. H. and DeLand, M. T.: Wildfire-driven thunderstorms cause a volcano-like stratospheric injection of smoke, NPJ Clim. Atmos. Sci., 1:30, doi:10.1038/s41612-018-0039-3, 2018.

–p.11,L8-13: this paragraph in particular was pretty incomprehensible to me. There are some other phrasings throughout which are difficult to parse as well; I haven't listed them all.

We apologize for this. We rephrased as following:

The analysis based on the mean IP values can be performed in various ways. Here we chose to analyse the function of continental source region. One can look at the mean values computed as the average over all available measurements for each IP. Recall that the number of events for each IP may vary for different measurements. Thus, the synergetic interpretation based on all IPs is challenging. Alternately, one may consider analysing the scatter plots between the different CRs and EAE, where, for each scatter plot, the mean values correspond to the same measurements. However, different scatter plots can be based on slightly different sets of measurements. The latter approach was chosen for our investigations.

We perform the analysis based on the mean IP values as a function of continental source region. We consider analysing the scatter plots between the different CRs and EAE, where, for each scatter plot, the mean values correspond to the same measurements. However, different scatter plots can be based on slightly different sets of measurements.

**Anonymous Referee #2**

The manuscript is the second part of a broader series where long range transport and local biomass burning events are detected and characterized through EARLINET - ACTRIS lidar network observations in Europe. Despite the importance of the subject under discussion, the paper is not introducing anything new at this stage compared with the other manuscript already published.

Biomass burning events have been extensively characterized by lidar observations over the past two decades. This manuscript, at present, reads as a dull and sometimes hard-to-follow laundry list of individual biomass burning events distinguished by some ambiguous set of common characteristics. Instrument networks are of fundamental importance to monitoring aerosol optical, geometrical and microphysical characteristics, and thus measurements and results cannot be reduced to such trivialization.

The paper is further missing compulsory context, as in who is going to benefit from these observations and how the article improves our knowledge on the subject?

We would like to clarify to the reviewer that this study is novel and original since it is the first study of such degree ever conducted based on ACTRIS-EARLINET datasets. As mentioned, previous (numerous) studies focused on specific case studies and most of the time they refer to individual stations. A few studies considered two or three stations. The current study takes into consideration all the available measurements (3589 files) at all locations (14 stations) over the period analysed (2008-2017). We mention again that as a result of the quality cheeks imposed on the data, we did not obtain many cases with complete optical properties (3 backscatter, 2 extinction and 1 or 2 depolarization). Thus, we were not able to pursue to a study on microphysical properties of the smoke. Please keep in mind also that many data used on various papers are not yet available in Earlinet database. Additionally, the following aspects haven been established in this work:

a) we are able to quantify the origin of BB in different locations and thus have a clearer picture of the type of smoke recorded (short range, long range transport smoke)

b) we tried to characterize the IPs measured in different geographical regions function of different smoke origin (here considered roughly as continental origin)

c) the analysis also revealed that, in spite of the big period used for analysis (2008-2017), a larger number of smoke measurements is needed while an improved quality control of the data provided to the database is required or a homogeneous data processing among the different stations should be applied. In the current study we dismissed around half of the input data.

d) The retrieved intensive optical properties and the errors of the biomass burning aerosols from the different stations can be used as a reference for future automatic aerosol classification.

Taken as a whole, the paper is more of a technical report that important contribution to the literature. The paper does not, therefore, clear the bar for advocacy of publication and need major revisions before publication.

In the manuscript, it is often cited that the increase in lidar ratio is linked to a higher absorption of the aerosols. The authors cannot assume that the size distribution is unchanged?

We are not investigating a process of dynamical changes within a certain time period, but we work on a certain set of properties obtained per time, thus the assumption of negligible change in size distribution per measurement is correct. Even if we consider changes in the particle size distribution caused by dry deposition, this process would affect similarly to all the cases and thus, the increase in the lidar ratio could only be explained by an increase the absorption.

It would be very interesting to pair lidar data with AERONET observations for a case study. The synergy among the two instruments could help to better characterize the microphysical elements in these events.

We agree with the reviewer that the synergistic combination of lidar and sun-photometer measurements may help to perform an analysis form the microphysical point of view, but this is out of the scope of this work. An example of the lidar derived microphysical properties compared with the sunphotometer retrievals is shown by Ortiz-Amezcua et al. (2017) for Warsaw station for 9 July 2013 (00:00-01:00 UTC). The sunphotometer data were from 04:23 UTC. The paper by Ortiz-Amezcua et al. presents data from the event of 8-10 July 2013 shown here.

The manuscript even if "Part II", should be able to stand alone. The majority of the acronyms are not defined and left to reader interpretation. Specific comments are found in the attached file.

We agree with the statement of the reviewer. We double check the acronyms. We added the definition of SCC and deleted UTLS (not used in this manuscript). Also, we added the description along the text when first used. The list of acronyms is now as Appendix of the main manuscript, as suggested by RC1 (the list of acronyms was added initially in Supplement).

Please also note the supplement to this comment: https://acp.copernicus.org/preprints/acp-2020-647/acp-2020-647-RC2-supplement.pdf

Specific comments:

Page 1 Line 30-31: not clear, please rephrase: "The data ranged from complete data sets (particle backscatter coefficient, extinction coefficient and linear depolarization ratio) to single profiles (particle backscatter coefficient)".

The abstract was rewritten as shown above (see answers to RC1, page 3).

Page 3 line 18: please write a correct data format, e.g. 20 June 2020.

Done throughout the entire manuscript.

Page 3 line 27: "SNR ≥ 2" why this threshold? It seems that also that the acronym is not defined before.

Thank you: it has been rephrased to:

The mean optical properties within the layers were calculated following a few criteria; only the optical properties for which signal to noise ratio SNR ≥ 2 were selected while the number of available data in the layer is ≥ 90% (Adam et al., 2020).

SNR ≥ 2 was chosen as a minimum for being able to discriminate between the noise and the real signal level. SNR is defined as the ratio between the signal and its uncertainty.

Page 4 line 6: Table 1 is difficult to read as the format is wrong in this pdf.

We changed the format increasing the font to 10.

Page 4 line 12: "SCC" not defined.

Thank you, we added the definition (Single Calculus Chain).

Page 4 line 13: "AOD" not defined.

Thank you, we added the definition (aerosol optical depth).

Page 4 line 27: "CRLR > 1 and EAE ~ 1" not defined.

Thank you, we added the definitions: colour ratio of lidar ratio (CRLR) and extinction-related Angstrom Exponent (EAE). (see Acronyms in Appendix 1).

Page 6 line 25-26: "In general, the change in LR can be linked to a change in particle size and/or a change in the light absorption capability of the particles (Müller et al., 2007)." then the authors are assuming that the particles didn't change their size distribution?

Yes, we rewrote this as:

In general, the change in LR can be linked to either a change in the particle size or a change in the light absorption capability of the particles, or both (Müller et al., 2007)." In the current analysed data, we assume that the change on the particles size distribution is negligible per set of measured data at a site.

Page 10 line 14-15: Again the assumption that the size distribution didn't change should hold.

Indeed, this is the case. We clarify by adding in the revised version of the paper: Here, again holds the assumption that the size distribution does not change.

Page 11 line 15: "... five measurements available." is this arbitrary? Statistics is significant depending on representativity of the process.

Yes, it is arbitrary.

Page 15 line 29: "by EARLINET stations" - by some EARLINET stations.

Corrected, thank you.

Page 15 line 32: "fires" - fire sources.

Corrected, thank you.

Page 16 line 13: "a moderate absorption at 355 nm (46 sr overall) and a high absorption at 532 nm (71 sr overall)" is it not possible to use ancillary observations to validate this statement?

Thank you, we clarified by adding: Unfortunately, it is not possible to use ancillary observations to validate this statement.

**Anonymous Referee #3

The paper by Adam et al. is part of a 2-part set of papers describing EARLINET observations of biomass burning transport to EARLINET sites in Europe. This second paper specifically covers the attribution of smoke events to longe range transport and regional smoke emissions, and provides an analysis of smoke property changes during transport. The attribution is specifically supported through HYSPLIT trajectory analyses and MODIS fire detections.

The subject matter is highly relevant to ACP and the evaluation of smoke transport dynamics is a significant contribution to the field. Arguably, the vertically resolved information on smoke properties after long range transport is one of the most important contributions by EARLINET as a whole. The quality of the EARLINET data set has been the subject of a long list of previous publications and as such is beyond reproach. The validity of the analysis and attribution methods using the trajectory and satellite fire detection algorithms is less obviously appropriate as indicated by my specific comments and questions below. The presentation quality in the form of figures and text is not at the required high level for ACP – specific suggestions for improvements are made below.

Overall, I would suggest that the paper is not acceptable for publication in its current form. However, because of the importance of the subject matter, the authors should be encouraged to resubmit or address all general and detailed comments included below.

General comments:

1) There is no section of text that describes how the analysis in this paper relates to the contents of paper 1. The authors seem to frequently assume that the reader must have read part 1 prior to reading part 2. This is always the challenge with multi-part papers. While it is appropriate to leave technical details to the other paper, it is not appropriate to require the reader to read the first paper before this one. Please add a section that describes the connectivity of the two papers and why they were divided the way they were.

The methodology used for the data analyses in this paper was detailed in the Part I (Adam et al. 2020, this issue). Here, in Section 2 (Review of methodology) we highlight the most important steps of the analyses (omitting the technical details), shown in Fig. S2.

In Section 2, we added the following when referring to the rejection of outliers:

The imposed limits (data filtering) for IPs are the following: LR@355 = [20 150] sr, LR@532 = [20 150] sr, EAE = [-1 3], BAE@355/532 = [-1 3], BAE@532/1064 = [-1 3], PDR@355 = [0 0.3] and PDR@532 15 = [0 0.3] (Adam et al., 2020).

We added the info about the number of layers analysed.

We analysed a number of 795 layers for which we identified at least one intensive parameter.

As a first step, the lidar profiles allocated to the Forest Fire category in the EARLINET/ACTRIS database were selected. The so-called backscatter (b) and extinction (e) files were used. They contain profiles of particle backscatter coefficient, particle extinction coefficient, and particle linear depolarization ratio. The quality of these profiles was ensured by internal EARLINET Quality Check (QC) procedures and external check to make sure only high-quality profiles are analysed. As a second step, the smoke layers were identified following an in-house developed method based on selecting layers with peak in the backscattered profiles. For each such potential layer, the 10-days backward trajectory was computed with the Hybrid Single-Particle Lagrangian Integrated Trajectory model (HYSPLIT) (Stein et al., 2015; Rolph et al., 2017). For these calculations the Global Data Assimilation System (GDAS) meteorological model was applied with 0.5° resolution. The identification of the smoke layers was based on the hypothesis of an existing fire within 100 km and ± 1 h from the time and location of the airmass, respectively. The location of the fires was provided by the Fire Information for Resource Management System (FIRMS) (https://firms.modaps.eosdis.nasa.gov/, last access 20191126) based on the Moderate Resolution Imaging Spectroradiometer (MODIS) sensor observations onboard the Aqua and Terra satellites (Davies et al., 2009). We regard that short-lived fires missed by satellite overpass do not significantly contributed to the amount of transported smoke. Fires not detected due to cloud occurrence were not taken into account. As a third step, the mean extensive and intensive optical properties were calculated within the identified smoke layers. Note that the mean properties within the smoke layers were calculated only for the properties for which SNR ≥ 2, while the number of available data in the layer is ≥ 90%. Moreover, data filtering for outliers was based on the following criteria: LR@355 = [20 150] sr, LR@532 = [20 150] sr, EAE = [-1 3], BAE@355/532 = [-1 3], BAE@532/1064 = [-1 3], PDR@355 = [0 0.3] and PDR@532 15 = [0 0.3]. We derived 795 smoke layers in total for which we have at least one intensive parameter available for analyses. In the Supplement, in Fig. S3 we show the number of layers and extensive optical properties (beta and alpha, collectively called EPs) for each time stamp for all stations, in Fig. S4 the mean optical properties values in the smoke layer, in Fig. S5 the number of layers and intensive optical parameters (EAE, PDR, LR,

collectively called IPs) for each time stamp for all stations, and in Fig. S6 the corresponding mean IPs values for each station. All IP values (including the outliers) are shown, along with the extreme values (marked by lines) reported in literature (based on the literature review in Adam et al. 2020, this issue).

The mean, median, minimum and maximum values of the intensive parameters for all of the stations providing at least one parameter (all stations but Sofia) are shown in Table 1. The number of available values for each variable is shown in Table 1 (# lines). As mentioned in Part I, there was a small number of IPs dismissed (outliers) based on predefined ranges of acceptable values (3.7 % overall). Regarding LR@355 obtained for the Thessaloniki station, we observed lower values than the ones reported by Amiridis et al. (2009) and Siomos et al. (2018a). The current mean value (over 20 cases) lies within the standard 10 deviation reported by Amiridis et al. (2009) but it is lower than the Free Troposphere minimum and maximum values of 61 ± 5 sr and 71 ± 7 sr, respectively (monthly averages over 15 years with filtered extreme values), reported by Siomos et al (2018a). Note that the current dataset for Thessaloniki was processed with SCC (Single Calculus Chain, version 4), while the dataset used by Siomos et al. (2018a) was processed with a station algorithm. Siomos et al. (2018b) showed that the AOD (aerosol optical depth) values (based on the particle extinction coefficient retrieved from the Raman channel) computed with the SCC algorithm are underestimated, as compared with those processed with the in-house algorithm.

2) Regarding the focus of the paper, there really is none. It is a confusing mix of single event discussions, source-receptor links, and particle property evolution during long range transport. The Abstract and Conclusion sections are accordingly confusing regarding the paper's main motivation.

We rewrote the abstract to clearly state the focus of the paper (see page 3 this document). Also, we partly modified the Conclusions as follows:

The present study shows results based on biomass burning events as recorded by some EARLINET stations over the 2008–2017 period, according to a methodology described in Part I (Adam et al., 2020). The main features of the methodology are: aerosol layers were labelled as smoke layers based on their Hysplit backtrajectory and the fire locations (provided by FIRMS), along the airmass backtrajectory according to established criteria. The smoke is labelled as 'mixed' if multiple fire sources contributed to the smoke measurement. For LRT smoke from North America, the smoke is labelled as 'pure North America' or 'mixed' (with contribution from both North America and Europe fires). Based on the methodology presented in Part I, we demonstrated that in most of the cases we record mixed smoke. Moreover, the number of fires and detections contributing to a smoke measurement was quantified. The quantification (based on number of fires and detections) of the contributing fires to the mixture explains the various values obtained for the intensive parameters and colour ratios.

The LRT event described here was recorded at three stations (Belsk, Cabauw and Warsaw) in July 2013, and captured some of the strongest fires that occurred in North America in June–July 2013. Our analysis revealed the presence of only one local fire (Sweden), which was weak compared to the North America ones and, thus, did not significantly contributed to the mixture. The 2462 fires identified in North America were detected 3970 times. The IPs values for 'mixed' and 'pure North America' cases are very similar (weak fire in Europe). The particles depolarization was low and small particles had low absorption.

The statistics over all the LRT events from North America revealed the following. The mean values of all IPs, except the LR@355 and EAE for 'pure North America' smoke, are closer to the mean values reported in literature for LRT smoke coming from North America. However, the relative differences between 'pure North America' and 'mixed' cases are not significant. For the LRT smoke, a moderate absorption at 355 nm (46 sr overall) and a high absorption at 532 nm (71 sr overall) were observed. Unfortunately, it is not possible to use ancillary observations to validate this statement. The mean $CR_{LR}$ and EAE suggest aged smoke, while the PDR values indicate a low depolarization and the BAE reveal more backscatter at smaller wavelengths.

~~The trajectory analysis based on four geographical regions revealed specific features. The histogram of the fires detected by each region along with the histogram of the backtrajectories revealed the following: the Central Europe stations detected mainly LRT smoke from North America, while the SW Europe region mostly smoke from fires occurring in the Iberian Peninsula and North Africa; in the NE and SE regions was measured mostly smoke from fires occurring in East Europe (especially Ukraine and West Russia). However, sporadic measurements were taken during the presence of smoke coming by LRT from North America. For each region, the IPs and, further, the colour ratio (CR) of various IPs are analysed based on their continental source origin. Most of the measurements were confined locally, within the SE, SW and NE Europe regions. The present methodology results revealed that North American fires contributed by 87 % to the smoke detected in Central Europe, 31 % to the smoke in the NE region, 9 % in the SW region and 4 % in the SE region.~~

The statistical analysis of the smoke properties (fresh versus aged) over smoke occurrences in four European regions (Central, North-East, South-West and South-East Europe) separating the smoke events into continental source regions

(European, North American, African, Asian or a mixture of two), based on trajectory analysis revealed the following. Based on the dataset analysed over 2008-2017, the smoke detected in the Central Europe (stations: Cabauw, Leipzig, and Hohenpeißenberg) was mostly brought form North America (87 % of fires). In the North-East Europe (Belsk, Minsk, Warsaw) smoke advected mostly from Eastern Europe (Ukraine and Russia) but there was a significant contribution (31 %) of smoke from North America. In the South-West Europe (Barcelona, Evora, Granada) smoke originated mainly over Iberian Peninsula and North Africa (while 9 % were originating in North America). In the South-East Europe (Athens, Bucharest, Potenza, Sofia, Thessaloniki) the origin of the smoke was mostly local (while 3 % represented smoke from North America). Most of the measurements were confined locally, within the South-East, South-West and North-East Europe regions.

For each region, the IPs and, further, the colour ratio (CR) of various IPs are analysed based on their continental source origin. The signature analysis of the scatter plots revealed the following features for the current dataset. Correlated with the increase of smoke travel time (corresponding to aging), $CR_{LR}$ and $CR_{BAE}$ increases while EAE and CRPDR  For the current dataset, the variability of the mean values (STD) is large in general and, thus, the individual values for different source regions overlap. Based on data from Warsaw (North-East region), the depolarization at 532 nm decreases for LRT (while $CR_{PDR} < 1$).

When the smoke originated from a single continental source, we noticed that the smoke is aged for all receptor regions except NE,  (where we have a mixture of fresh and aged smoke when the source is located in Europe. When the smoke originates from two continental sources (mixtures), the regions can measure either aged, fresh or a mixture of aged and fresh smoke, based on the smaller or higher contribution of the European (local) sources. Thus, in the South-East measurement region it was measured fresh smoke for the EUNA source region and a mixture of fresh and aged smoke originating from the EUAS. In the North-East region fresh smoke originating from EUAF was measured. For the South-West region with European or African source regions we obtained a $CR_{LR}$ of 0.8 and an EAE of 1. We assumed that the smoke measured was aged based on the high RH (in agreement with Veselovskii et al., 2020).

The lowest absorption was determined for the Central region (LRs < 36 sr). The South-West region displayed a highly absorbing smoke (61 sr < LR@355 < 79 sr and 64 < LR@532 < 91 sr). The South-East region displayed smoke with a medium/relatively high absorption at 532 nm (50–72 sr) and a low/medium absorption at 355 nm (31–48 sr). The smoke measured in the North-East region has a medium to very high absorption at 532 nm (57–91 sr) and a medium to high absorption at 355 nm (46–78 sr). The quite diverse absorption determined for the different measurement's regions, even for smoke from the same continental source region, may be related with different RH conditions (e.g. Veselovskii et al, 2020). We did not investigate the RH field.

The current study showed (in line with previous studies) that BAE and further $CR_{BAE}$ do not show specific values based on sources and no trend is observed. Thus, they cannot be used to identify the smoke type. In order to easily quantify the aerosol type, information about LR ($CR_{LR}$) and EAE is essential. Based on the implementation of ACTRIS Research Infrastructure in the next few years, the presented methodology will be applied on a larger dataset (more automatic lidar systems expected) providing more and more complete 3 backscatter + 2 extinction + depolarization datasets with enhanced quality control procedures.

~~One of the most important features observed on this study is that most of the smoke represents a mixture of several fires, which can be located very far from each other, and have (most probably) different characteristics. The quantification (based on number of fires and detections) of the contributing fires to the mixture explains the various values obtained for the intensive parameters and colour ratios.~~

The present methodology used to analyse the biomass burning events shows new approaches for smoke characterization (smoke type along with information about absorption and depolarization in the context of different continental sources) and can provide valuable information for various scientific communities (modelling, satellites).

For further investigations we envisage a more detailed analysis on grouping the sources' locations using cluster analysis, where a larger number of clusters should be chosen, to identify more homogeneous regions with similar vegetation type. Thus, a more accurate correlation between the source type and the measurements is envisaged. Moreover, the smoke time travel will be integrated. The challenge that remains is the quantification of the contribution of different fires in the mixed smoke (besides their number and detections).

3) Pertaining to the quality of the analysis, I fundamentally question the notion that a fire affects an airmass, just because the back-trajectory is located within a certain horizontal distance (100km) and time (1hr) of a satellite fire detection. It seems

necessary to detect whether (i) the trajectory is low enough in the atmosphere to be affected or (ii) the fire injection height is likely to have reached the trajectory altitude. This could be done on the basis of simple stability arguments from a reanalysis model. Without such an analysis, the attribution of smoke layers to given fires is highly questionable.

As a general statement, we should keep in mind that the principal investigators of each station do several initial checks of the obtained profiles per station to classify them into several categories, e.g. forest fire, dust, ash, Cirrus. In addition, through the Earlinet internal communication, emails are sent when models forecast events of Saharan dust or biomass burning. For the categories that are related to aerosol occurrence this is commonly assessed based on the backward trajectory analyses, which is usually constrained by the altitude at which the backward trajectory was persistent over the aerosol source area. For instance, for the mineral dust source over Africa this altitude is typically limited to the lower and middle troposphere, as these are the heights up to which the thermals can easily lift dust particles. In the case of the smoke particles, the injection height depends on the type of the burned area. For instance, the backtrajectory altitude over the source area can be constrained to much higher values over the forest fire areas (lifting smoke particles up to the high troposphere and even the stratosphere, e.g. Baars et al. 2019), and to much lower altitudes over the peatland and grass fire areas (confined to the lower troposphere, e.g. Stachlewska et al 2017). Therefore, the assessment on the likeliness of the smoke occurrence was done in the first place by the station PIs. Our task was to constrain it further. One of the reasons is that during a specific measurement, several layers can be observed where they might have different origin (e.g. Janicka et al., 2017, Osborne et al., 2019). In the example of LRT from N America, as recorded by Belsk, Warsaw and Cabauw (July 2013), if we look at the backtrajectories shown in Supplement, we see the trajectory altitudes (a.s.l.) above the fire location ranging from 0.5 km to 10 km. This strong forest fire event (July 2013) was reported by Ortiz-Amezcua et al. (2017) and Janicka et al. (2017) showing measurements in Warsaw, Leipzig and Granada. High trajectory altitudes over fires area were reported as well. The most recent strong forest fire event occurred in August 2017 in N America and it was already reported in several papers while the smoke reached the stratosphere (e.g. Peterson et al., 2018; Ansmann et al., 2018; Baars et al, 2020). Finally, a number of fires are likely not to be injected high enough to reach the air mass altitude. Indeed, the current approach may be improved with FRP addition, still - comparing with previous studies interpreting smoke layers with the fires from FIRMS - we introduced solid constrains. However, in the present study, we analysed 795 layers for which we identified 18555 fires, detected 34585 times.

In part I (section 4.3) we mentioned:

"We did not investigate the injection height based on fire radiative power (FRP) in order to estimate if the smoke of a particular fire reached indeed the altitude of the backtrajectory."

We also mentioned the common use of Hysplit and FIRMS in previous studies:

"FIRMS database was used in several studies to identify the BB origin. However, all fires occurring over certain periods (for which the backtrajectories were calculated) are typically accounted for. Thus, there were reported fires occurring over the whole day (e.g. Nicolae et al., 2013; Stachlewska et al., 2018; Janicka et al., 2017), or several days (e.g. Mylonaki et al., 2017; Heese and Wiegner, 2007; Tesche et al., 2011). By contrast, our novel approach accounts only for those fires which were occurring around backtrajectory (100 km radius) at the time of air masses passage (± 1 h)."

In Conclusions, when mentioning the future investigations, we added among others:

"FRP will be considered to estimate the injection height and thus have more confidence that the smoke reaches the backtrajectory' altitude."

In section 4.1 from part I, we have the following information:

"Most of the layers detected are situated between 1000 and 5000 m altitude (typically above PBL). However, the minimum layer bottom was found at 257.5 m while the highest layer top was found at 19,8 km. Minimum, maximum and the mean layer thickness were 300, 6862.5 and 1337.5 m. Please note that not all the layers shown here have BB origin (as this check is not performed yet)."

4) Except for stations explicitly mentioned in the text, there is no way to translate the station acronyms into geographic locations. Please eliminate all use of station acronyms, and provide a map and table with station information.

We add the map with the location (Fig. S1) and a table with geographical locations (Table S1) in Supplement. We eliminated the use of acronyms as suggested by editor before submitting the manuscript. We only left the acronyms in tables and figures (as the full name is not appropriate due to space constrains). The acronyms list is now in the main manuscript as appendix. They are also given in Table S1.

5) A long list of acronyms are not defined upon first use or not at all: LR, CRLR, SCC.

Thank you, we apologise for this mistake. We add the description when acronyms are first time used in the text, as suggested.

6) Quality of figures and figure captions is low for many figures – there are many instances of text on top of text (e.g., Fig 1, Fig S6), figure titles that are difficult to understand (e.g., in FigS2 - "evo:# times=13,# times with layers and optical properties=13. # total layers=16"), legend text is too small and/or blurry to read at 100% magnification; these occurrences are too frequent to list in detail. Some figures appear to run on for several pages (e.g., Fig S3), have lots of white space, no identification/letters for subplots, etc. Other figures have different aspect ratios for different subplots (e.g., Fig S4) because the number of events are different for the different stations, but that gives the reader the impression that the stations with fewer events are more important than others, because the former are taller. As a matter of fact, the list of required improvements to the figures is so long and the text is so difficult to follow, that I am breaking off the review in section 4.2. The authors should go through a general effort of making figures mode legible and titles, legends and captions more understandable, rather than relying on this version of figures that was clearly intended for QA purposes, first and foremost.

We apologize for figures being not straightforward to read. We reproduced the figures and replaced them in the revised version.

In Fig. 1, the title on each panel gives information about the type of IP (at different wavelength or different wavelength ratio, using * and o) while the colour is associated with the station (cog-blue, cbw-red and waw-yellow). Fig. S6 (now S7) are replaced with better quality plots.

Regarding Fig. S3, the white panels means no data. The plots are produced automatically in the same manner. We added the following statement in the caption of Figs 3 and 5:

The empty panels signify no data.

We thought that the labelling for the 13 figures (one for each station) does not bring value. The station acronym is mentioned in the title while the panels for intensive parameters are easy to read.

Regarding Fig. S4, you are right, the panels for Athens ("atz"), Bucharest ('ino') and Warsaw ("waw") have different aspect ratio. We reproduce the figures and replaced the old ones.

7) Predominantly in the Abstract and Introduction, there is a confusing lack of distinction in language between receptor regions and smoke origin regions. This makes it difficult to understand the primary purpose of the paper at hand. No map is provided that shows the four receptor regions and the 14 stations therein. Specifically, understanding the meaning of the sentence in the abstract "The results are analysed by means of intensive parameters in the following directions: I) long range transport of smoke particles from North America (here, we divided the events into 'pure North America'and 'mixed'-North America and local) smoke groups, and II) analysis of smoke particles over four geographical regions (SE Europe, NE Europe, Central Europe and SW Europe)." is very difficult. Given the focus on transport, the word "directions" is highly unfortunate. In fact, considering the entire manuscript, this is a rather poor description of the approach. A simpler description would have been that you analyzed the identified smoke occurrences in four European regions and separated the smoke events into source regions based on trajectory analyses (long-range transport from NA, mixed NA, and local).

We added the map with the stations as well as a table with geographical coordinates (Fig. S1 and Table S1 in Supplement). We replace the term "direction" with "research directions". See the abstract rewritten shown above. We rephrased as:

(III) statistical analysis of the smoke properties (fresh versus aged) over smoke occurrences in four European regions (Central, North-East, South-West and South-East Europe) separating the smoke events into continental source regions (European, North American, African, Asian or a mixture of two), based on trajectory analysis.

8) There is significant room for improvement in the use of English grammar. In particular, the use of definite and indefinite articles and the use of proper verb forms needs to be improved throughout the entire manuscript.

We did our best, including a professional scientific editor. Moreover, there will be the proofreading at the end.

We apologize for the quality of the English grammar which is due to the fact the most of the authors are not native. We used a professional scientific editor to correct the text. We also hope that the journal proofreading helps to perfect it at the end.

Detailed comments:

1) Page 1, line 36: Define what constitutes an "event".

An event represents a period of measurements. We had 23 periods lasting over a day and one period lasting over 3 days. After rephrasing the Abstract, the reference to 24 events was deleted. However, it is mentioned in section 3 as:

24 events (periods of measurements) of LRT from North America were identified, for which at least one intensive parameter was retrieved. The events occurred during 2009 and 2012–2017.

2) Abstract contains no statistics of the number of analyzed layers at all stations or in the stations grouped into geographic regions. Please add.

We acknowledge the reviewer this suggestion. The information about individual stations are given in Figs. S2 and S4. However, the total number was not mentioned (as given in Part I). We introduce the following sentence in the abstract.

A number of 795 layers for which we identified at least one intensive parameter was analysed. The layers were distributed as following: 399 layers for South East Europe, 119 layers for South West Europe, 243 layers in North East Europe and 34 layers for Central Europe.

3) Page 2, line 24: I am not familiar with the Keywood study, but based on a google search and my own understanding I can attest that an "inverse effect of BB impact on climate" is not commonly used terminology nor that such an impact is well recognized. Please expand.

According to the suggestion, we rephrase as follows:

Initial:

Keywood et al. (2013) report that the inverse effect of BB impact on climate is well recognized but not fully understood.

Changed:

Besides the biomass burning impacts on climate change, Keywood et al. (2013) describe the impacts of climate change on biomass burning (e.g. fire severity, increase of fuel consumption). The authors discuss the influence of short- and long-term climate changes on fire occurrence and severity and then they tackle the limitations in our ability to predict fire ignition and behaviour.

4) Page 3, line 22: "The identification of the smoke layers was assessed based on the hypothesis of an existing fire within 100 km and +/- 1 h from the time and location of the airmass, respectively". Was the altitude of the back-trajectory near the identified location of fires or an estimated Plume Injection Height not considered in the identification of likely smoke layers? (see general comment above).

See answer for general comment 3).

5) Page 3, line 26: Davies reference is missing.

We thank the reviewer. The corresponding reference was added.

6) Page 3: "The mean optical properties within the layers were calculated following a few criteria;…" This statement contains no information. Either add the criteria or remove.

We changed as:

The mean optical properties within the layers were calculated following a few criteria; only the optical properties for which SNR ≥ 2 were selected while the number of available data in the layer is ≥ 90% (Adam et al., 2020).

7) Page 3, line 32: Explain why there are often fewer numbers of parameters retrieved than layers detected in Fig S5.

Indeed, Comparing Figs S4 and S2, one can see that the number of optical properties profiles and the derived intensive parameters are in general smaller than the number of layers. The number of retrieved parameters depends on provision of sets of profiles by particular station as well as on the passing quality criteria listed in methodology. For a specific layer, not all the optical properties could be determined, mainly due to the following reasons: a) a specific profile was not provided in the database, e.g. beta profile provided by no kappa profile – thus no LR=kappa/beta; b) the profiles provided to the database covered different range (e.g. beta profile provided up to 6 km altitude while alpha only to 2 km, whereas the layer is observed at 5 km altitude), c) the mean value of the optical property in the layer was calculated only when the SNR of optical properties was greater than 2 (SNR ≥ 2 for ≥ 90% of data in the layer), or d) values filtered when being outside the realistic range.

We introduce the following statement for explanation:

In general, the number of the optical properties is smaller than the number of layers (Figs. S2) due to the following reasons: a) some profiles of the optical properties are not available in the database, b) some profiles provided in the database do not cover the entire attitude range, c) the mean values are calculated only if 90% of the data are available while the SNR ≥ 2 (see Fig.

S2). Further, the smaller number of IPs compared with the number of layers (Figs. S4) is due to the following reasons: a) an IP could not be determined if one optical property was missing, b) SNR should be $\geq 2$, c) filtering criteria.

8) Page 4, line 1: Terra/Aqua do not have four observations per day everywhere. Specify latitude range for which this is true or revise statement to properly reflect frequency of observations relevant to this investigation.

According to the reviewer's suggestion, we corrected for consistency with Part I ("… the same geographical location can be seen four times a day at the equator and more times as the latitude increases (due to orbits overlap)." as:

…overpasses (four observations a day at the equator and more times as the latitude increases)

9) Page 4, line 4: provide reference for MODIS fire detection.

The reference by Davies et al., 2009 was added.

Further, some fires might have not been detected due to clouds (Davies et al., 2009).

10) Page 4, line 18: "same graphics as in Section 3" – this is section 3. Do you mean section 3 of the part 1 paper?

We apologise for this confusing statement, it has been deleted, as the description of figures is given in section 3.1.

11) Page 4, lines 24-31: I do not understand the purpose of describing an event here that was described in detail in part 1. Without figure, this discussion is difficult to follow. Please describe the relevance for bringing this event up here.

As required by the reviewer, we rephrased the paragraph as following:

In the Part I paper we presented the event recorded on 13 July 2017 in Athens. Three layers were identified related to the LRT of smoke: the first two layers detected, as mixed with much larger contribution from local fires, while the third one as 'pure North America' smoke. The quantification of contributing smoke (LRT versus local) explains the differences in the properties (e.g. $CR_{LR}$ and EAE) derived in these layers. Although no colour ratio of the lidar ratio ($CR_{LR}$) was derived for the first layer, for the second it was $CR_{LR} < 1$, suggesting presence of some fresh smoke, in agreement with the large contribution of the local fires. In the contrary, for the third layer, values of $CR_{LR} > 1$ (and EAE ~1) suggest the presence of relatively large, aged particles. Moreover, the BAE@532/1064 and BAE@355/532 was the largest and lowest, respectively, i.e. colour ratio of the backscatter Angstrom exponent $CR_{BAE}$ showed a larger value for this 'pure North America' smoke.

12) Page 5, line 6 and Fig 1: I find panels c and d superfluous – they do not add to my understanding, as this info is contained in the map. The locations of Warsaw and Belsk on the map do not look right – the markers are too close to the border with Belarus.

Panels c) and d) show the location of the fires versus occurrence time and measurement time. Thus, one can observe the travel time by comparing the two times. It is true that for this particular case, where we encounter many events, it is not easy to make the link between the two locations. We followed for consistency the same graphics as used in Part I. Yes, you are right about markers. For maps with low resolution, the markers for the stations are shown somehow at the right side of the location given (by longitude and latitude). When zooming in the matlab figure, the locations look correct. We adjusted the location of the stations and the plot is updated.

13) Page 5, line 8: is this supposed to say Figs. 1c, d?

Yes, thanks you. We corrected the figure.

14) Page 5, line 16: "As a first remark…". By virtue of placement in the text, this is not a first remark, but it should be. Please move general description to the front of this section.

We moved the following text after the first sentence of the section:

As a first remark, we observed a very large number of fires occurring in North America (hundreds) from 30 June to 5 July 2013. Specifically, the smoke from 961 fires (detected 1664 times) was measured by Belsk, the smoke from 855 fires (detected 1241 times) was measured by Cabauw, and the smoke from 646 fires (detected 1065 times) was measured by Warsaw. This amounts to a total of 2462 fires, detected 3970 times.

We modified the following sentence:

Most of  the North America fires were quite strong, as shown by the colour and size of the markers (Figs. S6).

15) Page 5 and onward: Section 3.1 is a dense and unacceptably difficult to follow enumeration of facts and statistics that are not linked by any common thought or thread.

The event shown in 3.1 represents three days of measurements of strong N American fires recorded by three stations. According to our calculations, the fires contributing to each measurement were different (as stated by longitude and latitude). Over three days, there were 54 layers identified as smoke layer from which only 31 had origin in N America. Six of them represented mixed smoke where the local smoke was identified as being weak and thus without a large contribution to the mixture. LR@355 were below 50sr. Only one value for LR@532 (91sr) and for EAE (0.3) could be calculated. PDR was small. BAE showed a large variation over [0.3, 2]. Unfortunately, besides BAE there were not many common IPs determined for the three stations.

16) Page 9, line 10: Figure S7, after significant improvement needs to go into the introduction section and frame the entire paper.

We did not understand the comment (Fig. S7 is mentioned on line 4).

17) Figure 3 caption: explain the meaning of the markers with large error bar symbols.

In the caption of Fig. 3 we have:

The colour code for the mean (circle) and STD values is related with the source origin (stated as text on the plots).

We add the following to improve the plots' interpretation:

The colour code for the mean values (shown by circles) and their STD values (shown by error bars) is related to the source origin (stated as text on the plots). Thus, we have the following source origins: Europe (EU - blue), Asia (AS - yellow), North America (NA – magenta), Europe – Africa (EUAF – green), Europe – Asia (EUAS – cyan) and Europe – North America (EUNA – dark red).